# Source-free adaptation to measurement shift via bottom-up feature restoration

**Cian Eastwood**[*†§]    **Ian Mason**[*†]    **Christopher K. I. Williams**[†‡]    **Bernhard Schölkopf**[§]

† School of Informatics, University of Edinburgh
‡ Alan Turing Institute, London
§ MPI for Intelligent Systems, Tübingen

## Abstract

Source-free domain adaptation (SFDA) aims to adapt a model trained on labelled data in a source domain to unlabelled data in a target domain *without access to the source-domain data during adaptation*. Existing methods for SFDA leverage entropy-minimization techniques which: (i) apply only to classification; (ii) destroy model calibration; and (iii) rely on the source model achieving a good level of feature-space class-separation in the target domain. We address these issues for a particularly pervasive type of domain shift called *measurement shift* which can be resolved by *restoring* the source features rather than extracting new ones. In particular, we propose *Feature Restoration* (FR) wherein we: (i) store a lightweight and flexible approximation of the feature distribution under the source data; and (ii) adapt the feature-extractor such that the approximate feature distribution under the target data realigns with that saved on the source. We additionally propose a bottom-up training scheme which boosts performance, which we call *Bottom-Up Feature Restoration* (BUFR). On real and synthetic data, we demonstrate that BUFR outperforms existing SFDA methods in terms of accuracy, calibration, and data efficiency, while being less reliant on the performance of the source model in the target domain.

## 1 Introduction

In the real world, the conditions under which a system is developed often differ from those in which it is deployed—a concept known as *dataset shift* (Quiñonero-Candela et al., 2009). In contrast, conventional machine learning methods work by ignoring such differences, assuming that the development and deployment domains match or that it makes no difference if they do not match (Storkey, 2009). As a result, machine learning systems often fail in spectacular ways upon deployment in the test or *target* domain (Torralba & Efros, 2011; Hendrycks & Dietterich, 2019)

One strategy might be to re-collect and annotate enough examples in the target domain to re-train or fine-tune the model (Yosinski et al., 2014). However, manual annotation can be extremely expensive. Another strategy is that of *unsupervised domain adaptation* (UDA), where unlabelled data in the target domain is incorporated into the development process. A common approach is to minimize the domain 'gap' by aligning statistics of the source and target distributions in feature space (Long et al., 2015; 2018; Ganin & Lempitsky, 2015). However, these methods require simultaneous access to the source and target datasets—an often impractical requirement due to privacy regulations or transmission constraints, e.g. in deploying healthcare models (trained on private data) to hospitals with different scanners, or deploying image-processing models (trained on huge datasets) to mobile devices with different cameras. Thus, UDA *without access to the source data at deployment time* has high practical value.

Recently, there has been increasing interest in methods to address this setting of *source-free domain adaptation* (SFDA, Kundu et al. 2020; Liang et al. 2020; Li et al. 2020; Morerio et al. 2020) where the source dataset is unavailable during adaptation in the deployment phase. However, to adapt to the target domain, most of these methods employ entropy-minimization techniques which: (i) apply only to classification (discrete labels); (ii) destroy model calibration—minimizing prediction-entropy causes every sample to be classified (correctly or incorrectly) with extreme confidence; and (iii) assume that, in the target domain, the feature space of the unadapted source model contains reasonably well-separated data clusters, where samples within a cluster tend to share the same class label. As

---

*Equal contribution. Correspondence to c.eastwood@ed.ac.uk or ianxmason@gmail.com.

demonstrated in Section 5, even the most innocuous of shifts can destroy this *initial feature-space class-separation* in the target domain, and with it, the performance of these techniques.

We address these issues for a specific type of domain shift which we call *measurement shift* (MS). Measurement shift is characterized by a change in measurement system and is particularly pervasive in real-world deployed machine learning systems. For example, medical imaging systems often fail when deployed to hospitals with different scanners (Zech et al., 2018; AlBadawy et al., 2018; Beede et al., 2020) or different staining techniques (Tellez et al., 2019), while self-driving cars often struggle under "shifted" deployment conditions like natural variations in lighting (Dai & Van Gool, 2018) or weather conditions (Volk et al., 2019). Importantly, in contrast to many other types of domain shift, measurement shifts can be resolved by simply *restoring* the source features in the target domain—we do not need to learn *new* features in the target domain to discriminate well between the classes. Building on this observation, we propose Feature Restoration (FR)—a method which seeks to extract features with the same semantics from the target domain as were previously extracted from the source domain, under the assumption that this is sufficient to restore model performance. At development time, we train a source model and then use softly-binned histograms to save a lightweight and flexible approximation of the feature distribution under the source data. At deployment time, we adapt the source model's feature-extractor such that the approximate feature distribution under the target data aligns with that saved on the source. We additionally propose Bottom-Up Feature Restoration (BUFR)—a bottom-up training scheme for FR which significantly improves the degree to which features are restored by preserving learnt structure in the later layers of a network. While the assumption of measurement shift does reduce the generality of our methods—they do not apply to all domain shifts, but rather a subset thereof—our experiments demonstrate that, in exchange, we get improved performance on this important real-world problem. To summarize our main contributions, we:

- Identify a subset of domain shifts, which we call *measurement shifts*, for which restoring the source features in the target domain is sufficient to restore performance (Sec. 2);

- Introduce a *lightweight* and *flexible* distribution-alignment method for the source-free setting in which softly-binned histograms approximate the marginal feature distributions (Sec. 3);

- Create & release EMNIST-DA, a simple but challenging dataset for studying MS (Sec. 5.1);

- Demonstrate that BUFR generally outperforms existing SFDA methods in terms of accuracy, calibration, and data efficiency, while making less assumptions about the performance of the source model in the target domain (i.e. the initial feature-space class-separation) (Sec. 5.2–5.5);

- Highlight & analyse issues with entropy-minimization in existing SFDA methods (Sec. 5.5).

## 2 SETTING: SOURCE-FREE ADAPTATION TO MEASUREMENT SHIFT

We now describe the two phases of source-free domain adaptation (SFDA), development and deployment, before exploring measurement shift. For concreteness, we work with discrete outputs (i.e. classification) but FR can easily be applied to continuous outputs (i.e. regression).

**Source-free adaptation.** At **development time**, a source model is trained *with the expectation that an unknown domain shift will occur upon deployment in the target domain*. Thus, the primary objective is to equip the model for source-free adaptation at deployment time. For previous work, this meant storing per-class means in feature space (Chidlovskii et al., 2016), generating artificial negative datasets (Kundu et al., 2020), or introducing special training techniques (Liang et al., 2020). For us, this means storing lightweight approximate parameterizations of the marginal feature distributions, as detailed in the next section. More formally, a source model $f_s : \mathcal{X}_s \to \mathcal{Y}_s$ is trained on $n_s$ labelled examples from the source domain $\mathcal{D}_s = \{(\mathbf{x}_s^{(i)}, y_s^{(i)})\}_{i=1}^{n_s}$, with $\mathbf{x}_s^{(i)} \in \mathcal{X}_s$ and $y_s^{(i)} \in \mathcal{Y}_s$, before saving any lightweight statistics of the source data $\mathcal{S}_s$. At **deployment time**, we are given a pre-trained source model $f_s$, lightweight statistics of the source data $\mathcal{S}_s$, and $n_t$ unlabelled examples from the target domain $\mathcal{D}_t = \{\mathbf{x}_t^{(i)}\}_{i=1}^{n_t}$, with $\mathbf{x}_t^{(i)} \in \mathcal{X}_t$. The goal is to learn a target model $f_t : \mathcal{X}_t \to \mathcal{Y}_t$ which accurately predicts the unseen target labels $\{y_t^{(i)}\}_{i=1}^{n_t}$, with $y_t^{(i)} \in \mathcal{Y}_t$. Importantly, the source dataset $D_s$ is not accessible during adaptation in the deployment phase.

**Domain shift.** As depicted in Figure 1a, *domain shift* (Storkey, 2009, Section 9) can be understood by supposing some underlying, domain-invariant latent representation $L$ of a sample $(X, Y)$. This combines with the domain (or environment) variable $E$ to produce the observed covariates $X = m_E(L)$, where $m_E$ is some domain-dependent mapping. For example, $L$ could describe the shape,

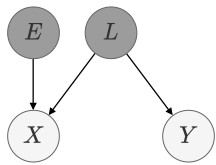 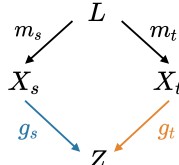 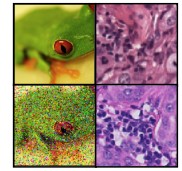 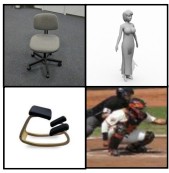

(a) Domain shift     (b) Feature restoration     (c) Measurement shifts     (d) *Non* measurement shifts

Figure 1: Domain shift, feature restoration and measurement shift. *(c,d):* Top=source, bottom=target. *(c):* CIFAR-10-C 'frog' & CAMELYON17 'tumor'. *(d):* Office-31 'desk chair' & VisDA-C 'person'.

appearance and pose parameters of scene objects, with $X$ obtained by "rendering" the scene $L$, taking into account parameters in $E$ that prescribe e.g. lighting, camera properties, background etc.

**Feature restoration.** In the source domain we learn a feature space $Z = g_s(X_s) = g_s(m_s(L))$, where our source model $f_s$ decomposes into a feature-extractor $g_s$ and a classifier $h$, with $f_s = h \circ g_s$ (left path of Figure 1b). For our source model $f_s$ to achieve good predictive accuracy, the features $Z$ *must* capture the information in $L$ about $Y$ and ignore the variables in $E = s$ that act as "nuisance variables" for obtaining this information from $X_s$ (e.g. lighting or camera properties). In the target domain ($E = t$), we often cannot extract the same features $Z$ due to a change in nuisance variables. This hurts predictive accuracy as it reduces the information about $L$ in $Z = g_s(X_t)$ (and thus about $Y$). We can *restore* the source features in the target domain by learning a target feature-extractor $g_t$ such that the target feature distribution aligns with that of the source (right path of Figure 1b), i.e. $p(g_t(X_t)) \approx p(g_s(X_s))$. Ultimately, we desire that for any $L$ we will have $g_s(m_s(L)) = g_t(m_t(L))$, i.e. that for source $X_s = m_s(L)$ and target $X_t = m_t(L)$ images generated from the same $L$, their corresponding $Z$'s will match. We can use synthetic data, where we have source and target images generated from the same $L$, to quantify the degree to which the source features are *restored* in the target domain with $|g_s(m_s(L)) - g_t(m_t(L))|$. In Section 5.5, we use this to compare quantitatively the degree of restoration achieved by different methods.

**Measurement shifts.** For many real-world domain shifts, restoring the source features in the target domain is sufficient to restore performance—we do not need to learn new features in order to discriminate well between the classes in the target domain. We call these *measurement shifts* as they generally arise from a change in measurement system (see Figure 1c). For such shifts, it is preferable to restore the same features rather than learn new ones via e.g. entropy minimization as the latter usually comes at the cost of model calibration—as we demonstrate in Section 5.

**Common UDA benchmarks are *not* measurement shifts.** For many other real-world domain shifts, restoring the source features in the target domain is *not* sufficient to restore performance—we need *new* features to discriminate well between the classes in the target domain. This can be caused by *concept shift* (Moreno-Torres et al., 2012, Sec. 4.3), where the features that define a concept change across source and target domains, or by the source model exploiting spurious correlations or *"shortcuts"* (Arjovsky et al., 2019; Geirhos et al., 2020) in the source domain which are not discriminative—or do not even exist—in the target domain. Common UDA benchmark datasets like Office-31 (Saenko et al., 2010) and VisDA-C (Peng et al., 2018) fall into this category of domain shifts. In particular, Office-31 is an example concept shift—'desk chair' has very different meanings (and thus features) in the source and target domains (left column of Fig. 1d)—while VisDA-C is an example of source models tending to exploit shortcuts. More specifically, in the synthetic-to-real task of VisDA-C (right column of Fig. 1d), source models tend not to learn general geometric aspects of the synthetic classes. Instead, they exploit peculiarities of the e.g. person-class which contains only 2 synthetic "people" rendered from different viewpoints with different lighting. Similarly, if we consider the real-to-synthetic task, models tend to exploit textural cues in the real domain that do not exist in the synthetic domain (Geirhos et al., 2019). As a result, the standard approach is to first pretrain on ImageNet to gain more "general" visual features and then carefully[1] fine-tune these features on (i) the source domain and then (ii) the target domain, effectively making the adaptation task ImageNet → synthetic → real. In Appendix D we illustrate that existing methods actually fail without this ImageNet pretraining as successful discrimination in the target domain *requires* learning new combinations of the general base ImageNet features. In summary, common UDA benchmarks like Office and VisDA-C *do not contain measurement shift* and thus are not suitable for evaluating our methods. We nonetheless report and analyse results on VisDA-C in Appendix D.

---

[1]Many works lower the learning rate of early layers in source and target domains, e.g. Liang et al. (2020).

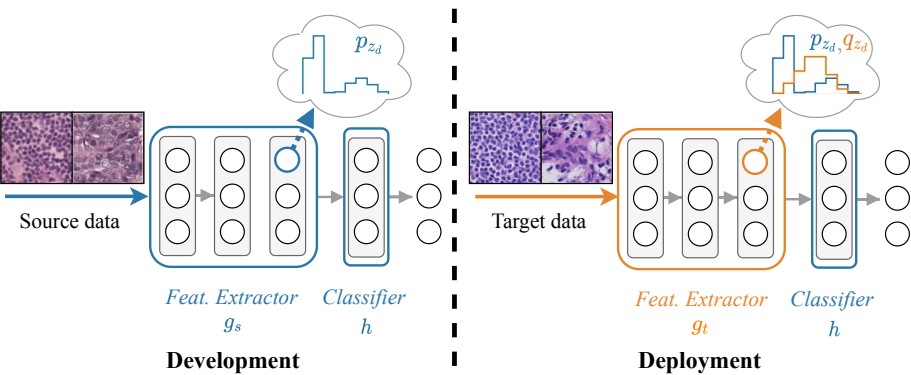

Figure 2: The Feature Restoration framework. *Left:* At development time, a source model is trained before saving approximations of the $D$ marginal feature distributions under the source data $\{p_{z_d}\}_{i=1}^{D}$. *Right:* At deployment time, the feature-extractor is adapted such that the approximations of the marginal feature distributions on the target data $\{q_{z_d}\}_{i=1}^{D}$ align with those saved on the source.

## 3 FEATURE RESTORATION

Below we detail the *Feature Restoration* (FR) framework. During development we train a model and then save a lightweight approximation of the feature distribution under the source data. At deployment time, we adapt the model's feature-extractor such that the approximate feature distribution under the target data aligns with that saved on the source. Figure 2 gives an overview of the FR framework.

### 3.1 DEVELOPMENT

**Setup.** The source model $f_s$ is first trained using some loss, e.g. cross-entropy. Unlike most existing SFDA methods (Chidlovskii et al., 2016; Liang et al., 2020; Kundu et al., 2020), we make no modification to the standard training process, allowing pretrained source models to be utilized. We decompose the source model $f_s$ into a feature-extractor $g_s : \mathcal{X}_s \to \mathbb{R}^D$ and a classifier $h : \mathbb{R}^D \to \mathcal{Y}_s$, where $D$ is the dimensionality of the feature space. So $\mathbf{z}_s^{(i)} = g_s(\mathbf{x}_s^{(i)})$ denotes the features extracted for source sample $i$, and $\hat{y}_s^{(i)} = f_s(\mathbf{x}_s^{(i)}) = h(g_s(\mathbf{x}_s^{(i)}))$ denotes the model's output for source sample $i$. Under the assumption of measurement shift, the feature extractor should be adapted to unlabelled target data to give $\mathbf{z}_t^{(i)} = g_t(\mathbf{x}_t^{(i)})$, but the classifier $h$ should remain unchanged, so that $\hat{y}_t^{(i)} = f_t(\mathbf{x}_t^{(i)}) = h(g_t(\mathbf{x}_t^{(i)}))$.

**Choosing an approximation of the feature distribution.** For high-dimensional feature spaces, storing the full joint distribution can be prohibitively expensive[2]. Thus, we choose to store only the marginal feature distributions. To accurately capture these marginal distributions, we opt to use soft binning (Dougherty et al., 1995) for its (i) *flexibility*—bins/histograms make few assumptions about distributional form, allowing us to accurately capture marginal feature distributions which we observe empirically to be heavily-skewed and bi-modal (see Appendix I); (ii) *scalability*—storage size does not scale with dataset size (Appendix A, Table 5), permitting very large source datasets (for a fixed number of bins $B$ and features $D$, soft binning requires constant $O(BD)$ storage and simple matrix-multiplication to compute soft counts); and (iii) *differentiability*—the use of soft (rather than "hard") binning, detailed in the next section, makes our approximation differentiable.

**Estimating the parameters of our approximation on the source data.** We now use the soft binning function of Yang et al. (2018, Sec. 3.1) to approximately parameterize the $D$ marginal feature distributions on the source data $\{p_{z_d}\}_{i=1}^{D}$, where $p_{z_d}$ denotes the marginal distribution of the $d$-th feature $z_d$. Specifically, we approximately parameterize $p_{z_d}$ using $B$ normalized bin counts $\pi_{z_d}^s = [\pi_{z_d,1}^s, \ldots, \pi_{z_d,B}^s]$, where $\pi_{z_d,b}^s$ represents the probability that a sample $z_d^{(i)}$ falls into bin $b$ under the source data and $\sum_{b=1}^{B} \pi_{z_d,b}^s = 1$. $\pi_{z_d}^s$ is calculated using

$$\pi_{z_d}^s = \sum_{i=1}^{n_s} \frac{\mathbf{u}(z_d^{(i)})}{n_s} = \sum_{i=1}^{n_s} \frac{\mathbf{u}(g(\mathbf{x}^{(i)})_d \, ; z_d^{min}, z_d^{max})}{n_s}, \tag{1}$$

where $z_d^{(i)} = g(\mathbf{x}^{(i)})_d$ denotes the $d$-th dimension of the $i$-th sample in feature space, $\mathbf{u}$ is the vector-

---

[2]If we assume features are jointly Normal, computational complexity is $O(ND^2)$ per update, where $N$ is the batch size. If we bin the feature space into histograms ($B$ bins per dimension), memory complexity is $O(B^D)$.

valued soft binning function (see Appendix A), $z_d^{min} = \min_{i=1}^{n_s} z_d^{(i)}$, and $z_d^{max}$ is defined analogously to $z_d^{min}$. Repeating this for all $D$ features, we get $\pi_{\mathbf{z}}^s = [\pi_{z_1}^s, \pi_{z_2}^s, \ldots, \pi_{z_D}^s]$. In the left-hand "cloud" of Figure 2, the blue curve depicts one such approximate marginal feature distribution $\pi_{z_d}^s$. We find it useful to additionally store approximate parameterizations of the marginal logit distributions on the source data $\pi_{\mathbf{a}}^s$, where the logit (i.e. pre-softmax) activations $\mathbf{a}^{(i)}$ are a linear combination of the feature activations $\mathbf{z}^{(i)}$, and $\pi_{\mathbf{a}}^s$ is defined analogously to $\pi_{\mathbf{z}}^s$. Note that we can parameterize a similar distribution for regression. Intuitively, aligning the marginal logit distributions further constrains the ways in which the marginal feature distributions can be aligned. We validate this intuition in the ablation study of Appendix J.2. Finally, we equip the model for source-free adaptation at deployment time by saving the parameters/statistics of the source data $\mathcal{S}_s = \{\pi_{\mathbf{z}}^s, \pi_{\mathbf{a}}^s, \mathbf{z}^{min}, \mathbf{z}^{max}, \mathbf{a}^{min}, \mathbf{a}^{max}\}$, where $\mathbf{z}^{min} = [z_1^{min}, z_2^{min}, \ldots, z_D^{min}]$ and $\mathbf{z}^{max}$, $\mathbf{a}^{min}$, and $\mathbf{a}^{max}$ are defined analogously.

## 3.2 DEPLOYMENT

At deployment time, we adapt the feature-extractor such that the approximate marginal distributions on the target data $(\pi_{\mathbf{z}}^t, \pi_{\mathbf{a}}^t)$ align with those saved on the source $(\pi_{\mathbf{z}}^s, \pi_{\mathbf{a}}^s)$. More specifically, we learn the target feature-extractor $g_t$ by minimizing the following loss on the target data,

$$\mathcal{L}_{tgt}(\pi_{\mathbf{z}}^s, \pi_{\mathbf{z}}^t, \pi_{\mathbf{a}}^s, \pi_{\mathbf{a}}^t) = \sum_{d=1}^{D} D_{SKL}(\pi_{z_d}^s || \pi_{z_d}^t) + \sum_{k=1}^{K} D_{SKL}(\pi_{a_k}^s || \pi_{a_k}^t), \tag{2}$$

where $D_{SKL}(p||q) = \frac{1}{2} D_{KL}(p||q) + \frac{1}{2} D_{KL}(q||p)$ is the symmetric KL divergence, and $D_{KL}(\pi_{z_d}^s || \pi_{z_d}^t)$ is the KL divergence between the *distributions* parameterized by normalized bin counts $\pi_{z_d}^s$ and $\pi_{z_d}^t$, which is calculated using

$$D_{KL}(\pi_{z_d}^s || \pi_{z_d}^t) = \sum_{b=1}^{B} \pi_{z_d,b}^s \log \frac{\pi_{z_d,b}^s}{\pi_{z_d,b}^t}, \tag{3}$$

with $\pi_{z_d,b}^s$ representing the probability of a sample from feature $d$ falling into bin $b$ under the source data, and $\pi_{z_d,b}^t$ under the target data. Practically, to update on a batch of target samples, we first approximate $\pi_{\mathbf{z}}^t$ and $\pi_{\mathbf{a}}^t$ on that batch using Eq. 1, and then compute the loss. Appendix B details the FR algorithm at development and deployment time, while Appendix L summarizes the notations.

## 3.3 BOTTOM-UP FEATURE RESTORATION

A simple gradient-based adaptation of $g_t$ would adapt the weights of all layers at the same time. Intuitively, however, we expect that many measurement shifts like brightness or blurring can be resolved by only updating the weights of early layers. If the early layers can learn to extract the same features from the target data as they did from the source (e.g. the same edges from brighter or blurrier images of digits), then the subsequent layers shouldn't need to update. Building on this intuition, we argue that adapting all layers simultaneously unnecessarily destroys learnt structure in the later layers of a network, and propose a bottom-up training strategy to alleviate the issue. Specifically, we adapt $g_t$ in a bottom-up manner, training for several epochs on one "block" before "unfreezing" the next. Here, a block can represent a single layer or group of layers (e.g. a residual block, He et al. 2016), and "unfreezing" simply means that we allow the block's weights to be updated. We call this method *Bottom-Up Feature Restoration* (BUFR). In Section 5 we illustrate that BU training *significantly* improves accuracy, calibration, and data efficiency by preserving learnt structure in later layers of $g_t$.

## 4 RELATED WORK

**Fine-tuning.** A well-established paradigm in deep learning is to first pretrain a model on large-scale "source" data (e.g. ImageNet) and then fine-tune the final layer(s) on "target" data of interest (Girshick et al., 2014; Zeiler & Fergus, 2014). This implicitly assumes that new high-level concepts should be learned by recombining old (i.e. fixed) low-level features. In contrast, under the assumption of measurement shift, we fix the final layer and fine-tune the rest. This assumes that the same high-level concepts should be *restored* by learning new low-level features. Royer & Lampert (2020) fine-tune each layer of a network individually and select the one that yields the best performance. For many domain shifts, they find it best to fine-tune an early or intermediate layer rather than the final one. This supports the idea that *which layer(s)* should update depends on *what* should be transferred.

**Unsupervised DA.** Inspired by the theory of Ben-David et al. (2007; 2010), many UDA methods seek to align source and target domains by matching their distributions in feature space (Long et al., 2015; 2018; Ganin & Lempitsky, 2015; Ganin et al., 2016; Tzeng et al., 2017; Shu et al., 2018).

However, as most of these methods are nonparametric (i.e. make no assumptions about distributional form), they require the source data during adaptation to align the distributions. In addition, parametric methods like Deep CORAL (Sun & Saenko, 2016) are not designed for the source-free setup—they prevent degenerate solutions during alignment with a classification loss on the source data and have storage requirements that are at least quadratic in the number of features. In contrast, our method works without the source data and its storage is linear in the number of features.

**Source-free DA.** Recently, Liang et al. (2020) achieved compelling results by re-purposing the semi-supervised information-maximization loss (Krause et al., 2010) and combining it with a pseudo-labelling loss (Lee et al., 2013). However, their entropy-minimizing losses are classification-specific, destroy model calibration, and rely on good initial source-model performance in the target domain (as demonstrated in the next section). Other works have trained expensive generative models so that the source data-distribution can be leveraged in the target domain (Li et al., 2020; Morerio et al., 2020; Kundu et al., 2020; Kurmi et al., 2021; Yeh et al., 2021; Stan & Rostami, 2021). However, these methods are still classification-specific and rely on good initial feature-space class-separation for entropy minimization (Li et al., 2020; Kundu et al., 2020), pseudo-labelling (Morerio et al., 2020; Stan & Rostami, 2021), and aligning the predictions of the source and target models (Kurmi et al., 2021; Yeh et al., 2021). Another approach is to focus on the role of batch-normalization (BN). Li et al. (2017) propose Adaptive BN (AdaBN) where the source data BN-statistics are replaced with those of the target data. This simple parameter-free method is often competitive with more complex techniques. Wang et al. (2021) also use the target data BN-statistics but additionally train the BN-parameters on the target data via entropy minimization, while Ishii & Sugiyama (2021) retrain the feature-extractor to align BN-statistics. Our method also attempts to match statistics of the marginal feature distributions, but is not limited to matching only the first two moments—hence can better handle non-Gaussian distributions.

## 5 EXPERIMENTS

In this section we evaluate our methods on multiple datasets (shown in Appendix F), compare to various baselines, and provide insights into *why* our method works through a detailed analysis.

### 5.1 SETUP

**Datasets and implementation.** Early experiments on MNIST-M (Ganin et al., 2016) and MNIST-C (Mu & Gilmer, 2019) could be well-resolved by a number of methods due to the small number of classes and relatively mild corruptions. Thus, to better facilitate model comparison, we additionally create and release EMNIST-DA—a domain adaptation (DA) dataset based on the 47-class Extended MNIST (EMNIST) character-recognition dataset (Cohen et al., 2017). We also evaluate on object recognition with CIFAR-10-C and CIFAR-100-C (Hendrycks & Dietterich, 2019), and on real-world measurement shifts with CAMELYON17 (Bandi et al., 2018). We use a simple 5-layer convolutional neural network (CNN) for digit and character datasets and a ResNet-18 (He et al., 2016) for the rest. Full dataset details are provided in Appendix F and implementation details in Appendix G. Code is available at `https://github.com/cianeastwood/bufr`.

**Baselines and their relation.** We show the performance of the source model on the source data as *No corruption*, and the performance of the source model on the target data (before adapting) as *Source-only*. We also implement the following baselines for comparison: *AdaBN* (Li et al., 2017) replaces the source BN-statistics with the target BN-statistics; *PL* is a basic pseudo-labelling approach (Lee et al., 2013); *SHOT-IM* is the information-maximization loss from Liang et al. (2020) which consists of a prediction-entropy term and a prediction-diversity term; and *target-supervised* is an upper-bound that uses labelled target data (we use a 80-10-10 training-validation-test split, reporting accuracy on the test set). For digit and character datasets we additionally implement *SHOT* (Liang et al., 2020), which uses the SHOT-IM loss along with special pre-training techniques (e.g. label smoothing) and a self-supervised PL loss; and *BNM-IM* (Ishii & Sugiyama, 2021), which combines the SHOT-IM loss from Liang et al. with a BN-matching (BNM) loss that aligns feature mean and variances on the target data with BN-statistics of the source. We additionally explore simple alternative parameterizations to match the source and target feature distributions: *Marg. Gauss.* is the BNM loss from Ishii & Sugiyama which is equivalent to aligning 1D Gaussian marginals; and *Full Gauss.* matches the mean and full covariance matrix. For object datasets we additionally implement *TENT* (Wang et al., 2021), which updates only the BN-parameters to minimize prediction-entropy, and also compare to some UDA methods. For all methods we report the classification accuracy and Expected Calibration Error (ECE, Naeini et al. 2015) which measures the difference in expectation between confidence and accuracy.

Table 1: Digit and character results. Shown are the mean and 1 standard deviation.

| Model | EMNIST-DA | | EMNIST-DA-SEVERE | | EMNIST-DA-MILD | |
|---|---|---|---|---|---|---|
| | ACC ↑ | ECE ↓ | ACC ↑ | ECE ↓ | ACC ↑ | ECE ↓ |
| No corruption | $89.4 \pm 0.1$ | $2.3 \pm 0.1$ | $89.4 \pm 0.1$ | $2.3 \pm 0.1$ | $89.4 \pm 0.1$ | $2.3 \pm 0.1$ |
| Source-only | $29.5 \pm 0.5$ | $30.8 \pm 1.6$ | $3.8 \pm 0.4$ | $42.6 \pm 3.5$ | $78.5 \pm 0.7$ | $4.8 \pm 0.5$ |
| AdaBN (Li et al., 2017) | $46.2 \pm 1.1$ | $30.3 \pm 1.1$ | $3.7 \pm 0.7$ | $52.4 \pm 4.9$ | $84.9 \pm 0.2$ | $4.9 \pm 0.3$ |
| Marg. Gauss. (Ishii & Sugiyama, 2021) | $51.8 \pm 1.1$ | $26.7 \pm 1.1$ | $4.8 \pm 0.5$ | $51.6 \pm 6.4$ | $85.8 \pm 0.3$ | $4.5 \pm 0.3$ |
| Full Gauss. | $67.9 \pm 0.7$ | $17.4 \pm 0.7$ | $29.8 \pm 9.8$ | $45.8 \pm 8.4$ | $85.7 \pm 0.2$ | $4.9 \pm 0.2$ |
| PL (Lee et al., 2013) | $50.0 \pm 0.6$ | $49.9 \pm 0.6$ | $2.7 \pm 0.4$ | $97.2 \pm 0.4$ | $83.5 \pm 0.1$ | $16.4 \pm 0.1$ |
| BNM-IM (Ishii & Sugiyama, 2021) | $63.7 \pm 2.2$ | $35.6 \pm 2.2$ | $8.3 \pm 1.3$ | $90.2 \pm 1.1$ | $86.5 \pm 0.1$ | $13.0 \pm 0.1$ |
| SHOT-IM (Liang et al., 2020) | $70.3 \pm 3.7$ | $29.6 \pm 3.7$ | $24.0 \pm 7.5$ | $76.0 \pm 7.5$ | $86.3 \pm 0.1$ | $13.7 \pm 0.1$ |
| SHOT (Liang et al., 2020) | $80.0 \pm 4.4$ | $19.7 \pm 4.4$ | $55.1 \pm 23.5$ | $42.7 \pm 23.0$ | $86.1 \pm 0.1$ | $14.8 \pm 0.1$ |
| FR (ours) | $74.4 \pm 0.8$ | $12.9 \pm 0.9$ | $15.3 \pm 6.8$ | $58.0 \pm 6.8$ | $86.4 \pm 0.1$ | $4.6 \pm 0.3$ |
| BUFR (ours) | $\mathbf{86.1 \pm 0.1}$ | $\mathbf{4.7 \pm 0.2}$ | $\mathbf{84.6 \pm 0.2}$ | $\mathbf{5.6 \pm 0.3}$ | $\mathbf{87.0 \pm 0.2}$ | $\mathbf{4.2 \pm 0.2}$ |
| Target-supervised | $86.8 \pm 0.6$ | $7.3 \pm 0.7$ | $85.7 \pm 0.6$ | $7.0 \pm 0.5$ | $87.3 \pm 0.7$ | $8.4 \pm 1.1$ |

## 5.2 CHARACTER-RECOGNITION RESULTS

Table 1 reports classification accuracies and ECEs for EMNIST-DA, with Appendix K reporting results for MNIST datasets (K.1) and full, per-shift results (K.4 and K.5). The severe and mild columns represent the most and least "severe" shifts respectively, where a shift is more severe if it has lower AdaBN performance (see Appendix K.5). On EMNIST-DA, BUFR convincingly outperforms all other methods—particularly on severe shifts where the initial feature-space class-separation is likely poor. Note the large deviation in performance across random runs for SHOT-IM and SHOT, suggesting that initial feature-space clustering has a big impact on how well these entropy-minimization methods can separate the target data. This is particularly true for the severe shift, where only BUFR achieves high accuracy across random runs. For the mild shift, where all methods perform well, we still see that: (i) BUFR performs the best; and (ii) PL, BNM-IM, SHOT-IM and SHOT are poorly calibrated due to their entropy-minimizing (i.e. confidence-maximizing) objectives. In fact, these methods are only reasonably calibrated if accuracy is very high. In contrast, our methods, and other methods that lack entropy terms (AdaBN, Marg. Gauss., Full Gauss.), maintain reasonable calibration as they do not work by making predictions more confident. This point is elucidated in the reliability diagrams of Appendix H.

## 5.3 OBJECT-RECOGNITION RESULTS

Table 2 reports classification accuracies and ECEs for CIFAR-10-C and CIFAR-100-C. Here we observe that FR is competitive with existing SFDA methods, while BUFR outperforms them on almost all fronts (except for ECE on CIFAR-100-C). We also observe the same three trends as on EMNIST-DA: (i) while the entropy-minimizing methods (PL, SHOT-IM, TENT) do well in terms of accuracy, their confidence-maximizing objectives lead to higher ECE—particularly on CIFAR-100-C where their ECE is even higher than that of the unadapted source-only model; (ii) the addition of bottom-up training significantly boosts performance; (iii) BUFR gets the largest boost on the most severe shifts—for example, as shown in the full per-shift results of Appendix K.6, BUFR achieves 89% accuracy on the impulse-noise shift of CIFAR-10-C, with the next best SFDA method achieving just 75%. Surprisingly, BUFR even outperforms target-supervised fine-tuning on both CIFAR-10-C and CIFAR-100-C in terms of accuracy. We attribute this to the regularization effect of bottom-up training, which we explore further in the next section.

We also report results for the "online" setting of Wang et al. (2021), where we may only use a single pass through the target data, applying mini-batch updates along the way. As shown in Table 13 of Appendix K.2, FR outperforms existing SFDA methods on CIFAR-10-C and is competitive on CIFAR-100-C. This includes TENT (Wang et al., 2021)—a method designed specifically for this online setting.

## 5.4 REAL-WORLD RESULTS

Table 4 reports results on CAMELYON17—a dataset containing real-world (i.e. naturally occurring) measurement shift. Here we report the average classification accuracy over 4 target hospitals. Note that the accuracy on the source hospital (i.e. no corruption) was 99.3%. Also note that this particular dataset is an ideal candidate for entropy-minimization techniques due to: (i) high AdaBN accuracy on the target data (most pseudo-labels are correct since updating only the BN-statistics gives ∼84%); (ii) a low number of classes (random pseudo-labels have a 50% chance of being correct); and (iii) a large target dataset. Despite this, our methods achieve competitive accuracy and show greater data efficiency—with 50 examples-per-class or less, only our methods meaningfully improve upon the simple AdaBN baseline which uses the target-data BN-statistics. These results illustrate that: (i) our method performs

Table 2: Object-recognition results. *: result adopted from Wang et al. (2021).

| Model | CIFAR-10-C | | CIFAR-100-C | |
|---|---|---|---|---|
| | ACC ↑ | ECE ↓ | ACC ↑ | ECE ↓ |
| No corruption | 95.3 ± 0.2 | 2.4 ± 0.1 | 76.4 ± 0.2 | 4.8 ± 0.1 |
| DANN* (Ganin et al., 2016) | 81.7 | - | 61.1 | - |
| UDA-SS.* (Sun et al., 2019) | 83.3 | - | 53 | - |
| Source-only | 57.8 ± 0.7 | 28.2 ± 0.4 | 36.4 ± 0.5 | 19.4 ± 0.9 |
| AdaBN (Li et al., 2018) | 80.4 ± 0.1 | 11.2 ± 0.1 | 56.6 ± 0.3 | **12.5 ± 0.1** |
| PL (Lee et al., 2013) | 82.5 ± 0.3 | 17.5 ± 0.3 | 62.1 ± 0.2 | 37.7 ± 0.2 |
| SHOT-IM (Liang et al., 2020) | 85.4 ± 0.2 | 14.6 ± 0.2 | 67.0 ± 0.2 | 32.9 ± 0.2 |
| TENT (Wang et al., 2021) | 86.6 ± 0.3 | 12.8 ± 0.3 | 66.0 ± 0.4 | 25.7 ± 0.4 |
| FR (ours) | 87.2 ± 0.7 | 11.3 ± 0.3 | 65.5 ± 0.2 | 15.7 ± 0.1 |
| BUFR (ours) | **89.4 ± 0.2** | **10.0 ± 0.2** | **68.5 ± 0.2** | 14.5 ± 0.3 |
| Target-supervised | 88.4 ± 0.9 | 6.4 ± 0.6 | 68.1 ± 1.2 | 9.6 ± 0.7 |

Table 3: EMNIST-DA degree of restoration.

| Model | D |
|---|---|
| Source-only. | 3.2 ± 0.0 |
| AdaBN | 3.1 ± 0.1 |
| Marg. Gauss. | 2.9 ± 0.0 |
| Full Gauss. | 2.0 ± 0.0 |
| PL | 2.6 ± 0.0 |
| BNM-IM | 2.5 ± 0.1 |
| SHOT-IM | 2.9 ± 0.1 |
| FR (ours) | 1.8 ± 0.0 |
| BUFR (ours) | **1.2 ± 0.0** |

Table 4: CAMELYON17 accuracies for a varying number of examples-per-class in the target domain.

| Model | 5 | 10 | 50 | 500 | All(> 15k) |
|---|---|---|---|---|---|
| Source-only | 55.8 ± 1.6 | 55.8 ± 1.6 | 55.8 ± 1.6 | 55.8 ± 1.6 | 55.8 ± 1.6 |
| AdaBN (Li et al., 2018) | 82.6 ± 2.2 | 83.3 ± 2.3 | 83.7 ± 1.0 | 83.9 ± 0.8 | 84.0 ± 0.5 |
| PL (Lee et al., 2013) | 82.5 ± 2.0 | 83.7 ± 1.7 | 83.6 ± 1.2 | 85.0 ± 0.8 | **90.6 ± 0.9** |
| SHOT-IM (Liang et al., 2020) | 82.6 ± 2.2 | 83.4 ± 2.5 | 83.7 ± 1.2 | 86.4 ± 0.7 | 89.9 ± 0.2 |
| FR (ours) | **84.6 ± 0.6** | 86.0 ± 0.7 | 86.0 ± 1.1 | 89.0 ± 0.6 | 89.5 ± 0.4 |
| BUFR (ours) | 84.5 ± 0.8 | **86.1 ± 0.2** | **87.0 ± 1.2** | **89.1 ± 0.8** | 89.7 ± 0.5 |

well in practice; (ii) measurement shift is an important real-world problem; and (iii) source-free methods are important to address such measurement shifts as, e.g., medical data is often kept private.

## 5.5 ANALYSIS

**Feature-space class-separation.** Measurement shifts can cause the target data to be poorly-separated in feature space. This point is illustrated in Figure 3 where we provide t-SNE visualizations of the feature-space class-separation on the EMNIST-DA crystals shift. Here, Figure 3a shows the initial class-separation *before* adapting the source model. We see that the source data is well separated in feature space (dark colours) but the target data is not (light colours). Figure 3b shows the performance of an entropy-minimization method when applied to such a "degraded" feature space where initial class-separation is poor on the target data. While accuracy and class-separation improve, the target-data clusters are not yet (i) fully homogeneous and (ii) returned to their original location (that of the source-data clusters). As shown in Figure 3(c,d), our methods of FR and BUFR better restore class-separation on the target data with more homogeneous clusters returned to their previous location.

**Quantifying the degree of restoration.** We quantify the degree to which the EMNIST source features are *restored* in each of the EMNIST-DA target domains by calculating the average pairwise distance: $D = \frac{1}{T} \sum_{t=1}^{T} \frac{1}{N} \sum_{i=1}^{N} |g_s(m_s(X^{(i)})) - g_t(m_t(X^{(i)}))|$, where $T$ is the number of EMNIST-DA target domains, $N$ is the number of EMNIST images, $X^{(i)}$ is a clean or uncorrupted EMNIST image, $m_s$ is the identity transform, and $m_t$ is the shift of target domain $t$ (e.g. Gaussian blur). Table 3 shows that the purely alignment-based methods (Marg. Gauss., Joint Gauss., FR, BUFR) tend to better restore the features than the entropy-based methods (PL, BNM-IM, SHOT-IM), with our alignment-based methods doing it best. The only exception is Marg. Gauss.—the weakest form of alignment. Finally, it is worth noting the strong rank correlation (0.6) between the degree of restoration in Table 3 and the ECE in Table 1. This confirms that, for measurement shifts, it is preferable to restore the same features rather than learn new ones as the latter usually comes at the cost of model calibration.

**Restoring the semantic meaning of features.** The left column of Figure 4a shows the activation distribution (bottom) and maximally-activating image patches (top) for a specific filter in the first layer of a CNN trained on the standard EMNIST dataset (white digit, black background). The centre column shows that, when presented with shifted target data (pink digit, green background), the filter detects similar patterns of light and dark colours but no longer carries the same semantic meaning of detecting a horizontal edge. Finally, the right column shows that, when our BUFR method aligns the marginal feature distributions on the target data (orange curve, bottom) with those saved on the source data (blue curve, bottom), this restores a sense of semantic meaning to the filters (image patches, top). Note that we explicitly align the *first-layer* feature/filter distributions in this illustrative experiment.

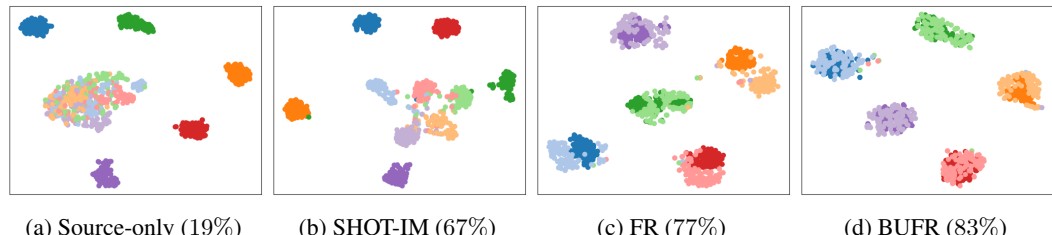

(a) Source-only (19%)      (b) SHOT-IM (67%)      (c) FR (77%)      (d) BUFR (83%)

Figure 3: t-SNE (Van der Maaten & Hinton, 2008) visualization of features for 5 classes of the EMNIST-DA crystals shift. Dark colours show the source data, light the target. Model accuracies are shown in parentheses.

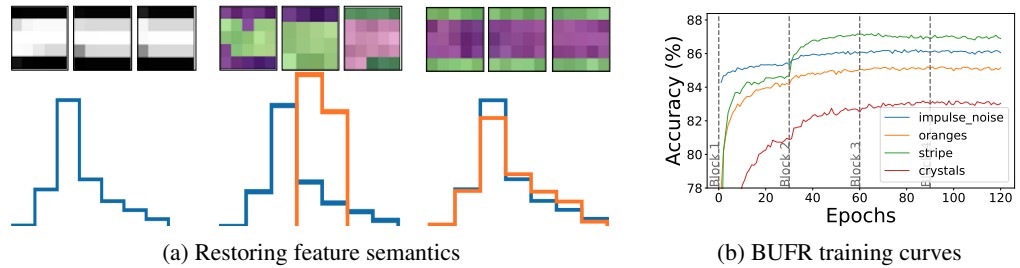

(a) Restoring feature semantics          (b) BUFR training curves

Figure 4: (a) Activation distributions (bottom) and maximally-activating image patches (top) for a specific filter in the first layer of a CNN. *Left:* Source model, source data (white digit, black backgr.). *Centre:* Source model, target data (pink digit, green backgr.). *Right:* Target model (adapted with BUFR), target data. (b) BUFR training curves on selected EMNIST-DA corruptions. Dashed-grey lines indicate when the next block is unfrozen.

**Efficacy of BU training.** Figure 4b shows that, when training in a bottom-up manner, updating only the first two blocks is sufficient to resolve many measurement shifts. This confirms the previous intuition that updating only the early layers should be sufficient for many measurement shifts. BUFR exploits this by primarily updating early layers, thus preserving learnt structure in later layers (see Appendix J.3–J.4). To examine the regularization benefits of this structure preservation, we compare the accuracy of BUFR to other SFDA methods as the number of available target examples reduces. As shown in Table 9 of Appendix J.1, the performance of all competing methods drops sharply as we reduce the number of target examples. In contrast, BUFR maintains strong performance. With only 5 examples-per-class, it surpasses the performance of many methods using all 400 examples-per-class.

**Ablation study.** We also conduct an ablation study on the components of our loss from Equation 2. Table 10 of Appendix J.2 shows that, for easier tasks like CIFAR-10-C, aligning the logit distributions and using the symmetric KL divergence (over a more commonly-used asymmetric one) make little difference to performance. However, for harder tasks like CIFAR-100-C, both improve performance.

## 6   DISCUSSIONS

**Aligning the marginals may be insufficient.** Our method seeks to restore the joint feature distribution by aligning (approximations of) the marginals. While we found that this is often sufficient, it cannot be guaranteed unless the features are independent. One potential remedy is to encourage feature independence in the source domain using "disentanglement" (Bengio et al., 2013; Eastwood & Williams, 2018) methods, allowing the marginals to better capture the joint.

**Model selection.** Like most UDA & SFDA works, we use a target-domain validation set (Gulrajani & Lopez-Paz, 2021) for model selection. However, such labelled target data is rarely available in real-world setups. Potential solutions include developing benchmarks (Gulrajani & Lopez-Paz, 2021) and validation procedures (You et al., 2019) that allow more realistic model selection and comparison.

**Conclusion.** We have proposed BUFR, a method for source-free adaptation to measurement shifts. BUFR works by aligning histogram-based approximations of the marginal feature distributions on the target data with those saved on the source. We showed that, by focusing on measurement shifts, BUFR can outperform existing methods in terms of accuracy, calibration and data efficiency, while making less assumptions about the behaviour of the source model on the target data. We also highlighted issues with the entropy-minimization techniques on which existing SFDA-methods rely, namely their classification-specificity, tendency to be poorly calibrated, and vulnerability to simple but severe shifts.

ACKNOWLEDGEMENTS

We thank Tim Hospadales, Amos Storkey, Oisin Mac Aodha, Luigi Gresele and Julius von Kügelgen for helpful discussions and comments. CE acknowledges support from The National University of Ireland via his Travelling Studentship in the Sciences. IM is supported by the Engineering and Physical Sciences Research Council (EPSRC).

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

# Appendix

## Table of Contents

## A  SOFT BINNING

**Function.**  Let $z \sim p_z$ be a continuous 1D variable for which we have $n$ samples $\{z^{(i)}\}_{i=1}^n$. The goal is approximately parameterize $p_z$ using $B$ normalized bin counts $\pi_z = [\pi_{z,1}, \dots, \pi_{z,B}]$, where $\pi_{z,b}$ represents the probability that $z$ falls into bin $b$ and $\sum_{b=1}^B \pi_{z,b} = 1$. We achieve this using the soft binning function of Yang et al. (2018, Section 3.1). The first step is to find the range of $z$, i.e. the minimum and maximum denoted $z^{min} = \min_i z^{(i)}$ and $z^{max} = \max_i z^{(i)}$ respectively. This will allow us to normalize the range of our samples $z^{(i)}$ to be $[0, 1]$ and thus ensure that binning "softness", i.e. the degree to which mass is distributed into nearby bins, is comparable across variables with different ranges. The second step is to define $B - 1$ uniformly-spaced and monotonically-increasing cut points (i.e. bin edges) over this normalized range $[0, 1]$, denoted $\mathbf{c} = [c_1, c_2, \dots, c_{B-1}] = \frac{1}{B-2}[0, 1, 2, \dots, B-3, B-2]$. The third step is to compute the $B$-dimensional vector of soft counts for a sample $z^{(i)}$, denoted $\mathbf{u}(z^{(i)})$, using soft binning vector-valued function $\mathbf{u}$,

$$\mathbf{u}(z^{(i)}; z^{min}, z^{max}) = \sigma\left(\left(\mathbf{w}\left(\frac{z^{(i)} - z^{min}}{z^{max} - z^{min}}\right) + \mathbf{w}_0\right)/\tau\right), \tag{4}$$

where $\mathbf{w} = [1, 2, \dots, B]$, $\mathbf{w}_0 = [0, -c_1, -c_1 - c_2, \dots, -\sum_{j=1}^{B-1} c_j]$, $\tau > 0$ is a temperature factor, $\sigma$ is the softmax function, $\mathbf{u}(z^{(i)})_b$ is the mass assigned to bin $b$, and $\sum_{b=1}^B \mathbf{u}(z^{(i)})_b = 1$. Note that: (i) both $\mathbf{w}$ and $\mathbf{w}_0$ are constant vectors for a pre-specified number of bins $B$; (ii) as $\tau \to 0$, $\mathbf{u}(z^{(i)})$ tends to a one-hot vector; and (iii) the $B - 1$ cut points $\mathbf{c}$ result in $B$ bins, where values $z^{(i)} < 0$ or $z^{(i)} > 1$ are handled sensibly by the soft binning function in order to catch new samples that lie outside the range of our original $n$ samples (as $\tau \to 0$, they will appear in the leftmost or rightmost bin respectively). Finally, we get the total counts per bin by summing over the per-sample soft counts $\mathbf{u}(z^{(i)})$, before normalizing by the total number of samples $n$ to get the normalized bin counts $\pi_z$, i.e., $\pi_z = \sum_{i=1}^n \frac{\mathbf{u}(z^{(i)}; z^{min}, z^{max})}{n}$.

**Memory cost.**  When using 32-bit floating point numbers for each (soft) bin count, the memory cost of soft binning is $32 \times B \times D$ bits—depending only on the number bins $B$ and the number of features $D$, and *not* on the dataset size. For concreteness, Table 5 compares the cost of storing bin counts to that of: (i) storing the whole source dataset; and (ii) storing the (weights of the) source model. As in our experiments, we assume 8 bins per feature and the following network architectures: a variation of LeNet (LeCun et al., 1998) for MNIST; ResNet-18 (He et al., 2016) for CIFAR-100; and ResNet-101 (He et al., 2016) for both VisDA-C (Peng et al., 2018) and ImageNet (Russakovsky et al., 2015).

Table 5: Storage size for different datasets and their corresponding source models.

| Storage size (MB) | MNIST | CFR-100 | VisDA-C | ImageNet |
|---|---|---|---|---|
| Source dataset | 33 | 150 | 7885 | 138000 |
| Source model | 0.9 | 49 | 173 | 173 |
| Source bin-counts | 0.004 | 0.02 | 0.5 | 0.5 |

## B  FR ALGORITHM

Algorithm 1 gives the algorithm for FR at *development time*, where a source model is trained before saving approximations of the feature and logit distributions under the source data. Algorithm 2 gives the algorithm for FR at *deployment time*, where the feature-extractor is adapted such that the approximate feature and logit distributions under the target data realign with those saved on the source.

---

**Algorithm 1:** FR at *development* time.

**Input:** Source model $f_s$, labelled source data $D_s = (X_s, Y_s)$, number of bins $B$, number of training iterations $I$.

```
/* Train src model f_s = h ∘ g_s    */
for i in range(I) do
    L_i ← 𝓛_src(f_s, D_s) ;
    f_s ← SGD(f_s, L_i) ;
```

```
/* Calc. feat.&logit ranges    */
```
$\mathbf{z}^{min}, \mathbf{z}^{max} \leftarrow$ CALC_RANGE$(f_s, X_s)$ ;
$\mathbf{a}^{min}, \mathbf{a}^{max} \leftarrow$ CALC_RANGE$(f_s, X_s)$ ;

```
/* Calc. feat.&logit bin cnts    */
```
$\pi_{\mathbf{z}}^s \leftarrow$ CALC_BC$(f_s, X_s; \mathbf{z}^{min}, \mathbf{z}^{max}, B)$ ;
$\pi_{\mathbf{a}}^s \leftarrow$ CALC_BC$(f_s, X_s; \mathbf{a}^{min}, \mathbf{a}^{max}, B)$ ;

```
/* Gather source stats 𝓢_s    */
```
$\mathcal{S}_s \leftarrow \{\pi_{\mathbf{z}}^s, \pi_{\mathbf{a}}^s, \mathbf{z}^{min}, \mathbf{z}^{max}, \mathbf{a}^{min}, \mathbf{a}^{max}\}$ ;

**Output:** $f_s, \mathcal{S}_s$

---

**Algorithm 2:** FR at *deployment* time.

**Input:** Source model $f_s$, unlabelled target data $X_t$, source data statistics $\mathcal{S}_s$, number of adaptation iterations $I$.

```
/* Init trgt model f_t = h ∘ g_t    */
f_t ← f_s ;
```

```
/* Adapt trgt feat.-extractr g_t   */
for i in range(I) do
```
$\quad \pi_{\mathbf{z}}^t \leftarrow$ CALC_BC$(f_t, X_t; \mathbf{z}^{min}, \mathbf{z}^{max}, B)$ ;
$\quad \pi_{\mathbf{a}}^t \leftarrow$ CALC_BC$(f_t, X_t; \mathbf{a}^{min}, \mathbf{a}^{max}, B)$ ;

$\quad L_i \leftarrow \mathcal{L}_{tgt}(\pi_{\mathbf{z}}^s, \pi_{\mathbf{z}}^t, \pi_{\mathbf{a}}^s, \pi_{\mathbf{a}}^t)$ ;
$\quad g_t \leftarrow$ SGD$(g_t, L_i)$ ;

**Output:** $g_t$

---

## C  WHEN MIGHT FR WORK?

**Toy example where FR *will* work.**  Let $L$ take two values $\{-1, 1\}$, and let

$$Y = L \tag{5}$$
$$X = U[L - 0.5, L + 0.5] + E, \tag{6}$$

where $U$ denotes a uniform distribution and $E$ a domain-specific offset (this setup is depicted in Figure 1a). Then the optimal classifier $f : X \rightarrow Y$ can be written as $f(X) = sign(X - E)$. Imagine the source domain has $E = 0$, and the target domain has $E = 2$. Then all points will be initially classified as positive in the target domain, but FR will restore optimal performance by essentially "re-normalizing" $X$ to achieve an intermediate feature representation $Z$ with the same distribution as before (in the source domain).

**Toy example where FR *will not* work.**  Let $L$ be a rotationally-symmetric multivariate distribution (e.g. a standard multivariate Gaussian), and let $X$ be a rotated version of $L$ where the rotation depends on $E$. Now let $Y = L_1$, the first component of $L$. Then any projection of $X$ will have the correct marginal distribution, hence FR will not work here as matching the marginal distributions of the intermediate feature representation $Z$ will not be enough to yield the desired invariant representation.

**How to know if FR is suitable.**  We believe it reasonable to assume that one has knowledge of the type of shifts that are likely to occur upon deployment. For example, if deploying a medical imaging system to a new hospital, one may know that the imaging and staining techniques may differ but the catchment populations are similar in e.g. cancer rate. In such cases, we can deduce that measurement shift is likely and thus FR is suitable.

## D COMMON UDA BENCHMARKS ARE NOT MEASUREMENT SHIFTS

**Overview.** The standard approach for common UDA benchmarks like VisDA-C (Peng et al., 2018) is to first pretrain on ImageNet to gain more "general" visual features and then carefully fine-tune these features on (i) the source domain, and then (ii) the target domain, effectively making the adaptation task ImageNet $\rightarrow$ synthetic $\rightarrow$ real. Here, we use VisDA-C to: (i) investigate the reliance of existing methods on ImageNet pretraining; (ii) evaluate our FR and BUFR methods on domain shifts that *require* learning new features (i.e. *non* measurement shifts); and (iii) investigate the effect of label shift on our methods (which violates the assumption of measurement shift and indeed even domain shift).

**Reducing label shift.** For (iii), we first note that VisDA-C contains significant label shift. For example, 8% of examples are labelled 'car' in the source domain, while 19% of examples are labelled 'car' in the target domain. To correct for this while retaining as many examples as possible, we randomly drop examples from some classes and oversample examples from others so that all classes have 11000 examples in the source domain and 3500 examples in the target domain—this is labelled as "No label shift" in Table 6.

**Results.** In Table 6 we see that: (i) without ImageNet pre-training, all (tested) methods fail—despite similar accuracy being achieved in the source domain with or without ImageNet pre-training (compare ✗✗ vs. ✓✗); (ii) with the standard VisDA-C setup (i.e. ✓✗), AdaBN < FR << SHOT, as SHOT learns *new* discriminative features in the target domain; and (iii) correcting for label shift boosts the performance of FR and closes the gap with SHOT (compare ✓✗ vs. ✓✓), but some gap remains as *VisDA-C is not a measurement shift but rather a more general domain shift*. Finally, we note that ImageNet pretraining makes the features in early layers quite robust, reducing the advantage of bottom-up training.

**Implementation details.** These results were achieved using a standard VisDA-C implentation/setup: we train a ResNet-101 (He et al., 2016) (optionally pre-trained on ImageNet) for 15 epochs using SGD, a learning rate of 0.001, and a batch size of 64. We additionally adopt the learning rate scheduling of (Ganin & Lempitsky, 2015; Long et al., 2018; Liang et al., 2020) in the source domain, and reduce the learning rate to 0.0001 in the target domain.

Table 6: VisDA-C results (ResNet-101). *No label shift*: examples were dropped or oversampled to correct for label shift.

| Model | ImageNet pretrain | No label shift | Avg. Acc. |
|---|---|---|---|
| No corruption | ✗ | ✗ | 99.8 |
| Source-only | ✗ | ✗ | 10.4 |
| AdaBN (Li et al., 2017) | ✗ | ✗ | **15.9** |
| SHOT (Liang et al., 2020) | ✗ | ✗ | **17.1** |
| FR | ✗ | ✗ | **16.8** |
| BUFR | ✗ | ✗ | **16.2** |
| No corruption | ✓ | ✗ | 99.6 |
| Source-only | ✓ | ✗ | 47.0 |
| AdaBN (Li et al., 2017) | ✓ | ✗ | 65.2 |
| SHOT (Liang et al., 2020) | ✓ | ✗ | **82.9** |
| FR | ✓ | ✗ | 73.7 |
| BUFR | ✓ | ✗ | 72.9 |
| No corruption | ✓ | ✓ | 99.7 |
| Source-only | ✓ | ✓ | 44.6 |
| AdaBN (Li et al., 2017) | ✓ | ✓ | 68.7 |
| SHOT (Liang et al., 2020) | ✓ | ✓ | **85.0** |
| FR | ✓ | ✓ | 82.8 |
| BUFR | ✓ | ✓ | 83.1 |

# E    FURTHER RELATED WORK

**Domain generalization.** Domain generalization seeks to do well in the target domain *without updating the source model.* The goal is to achieve this through suitable data augmentation, self-supervision, and inductive biases with respect to a perturbation of interest (Simard et al., 1991; Engstrom et al., 2019; Michaelis et al., 2019; Roy et al., 2019; Djolonga et al., 2021). One may view this as specifying the shifts that a model should be robust to *a priori*. Practically, however, we generally do not know what shift will occur upon deployment—there will always be unseen shifts. Furthermore, the condition that our augmented development process be sufficiently diverse is untestable—with the worst-case error still being arbitrarily high (David et al., 2010; Arjovsky et al., 2019). Permitting adaptation in the target domain is one reasonable solution to these problems.

**Common corruptions.** Previous works (Hendrycks & Dietterich, 2019) have used *common corruptions* to study the robustness of neural networks to simple transformations of the input, e.g. Gaussian noise (common in low-lighting conditions), defocus blur (camera is not properly focused or calibrated), brightness (variations in daylight intensity), and impulse noise (colour analogue of salt-and-pepper noise, caused by bit errors). We see common corruptions as one particular type of measurement shift, with all the aforementioned corruptions arising from a change in measurement system. However, not all measurement shifts are common corruptions. For example, the right column of Figure 1c depicts tissue slides from different hospitals. Here, the shift has arisen from changes in slide-staining procedures, patient populations and image acquisition (e.g. different sensing equipment). This measurement shift cannot be described in terms of simple input transformations like Gaussian noise or blurring, and thus we do not consider it a common corruption. In addition, EMNIST-DA shifts like bricks and grass use knowledge of the object type (i.e. a digit) to change the background and foreground separately (see Figure 7). We do not consider these to be common corruptions as common corruptions rarely have knowledge of the image content—e.g. blurring all pixels or adding noise randomly. In summary, we consider measurement shifts to be a superset of common corruptions, thus warranting their own definition.

**SFDA and related settings.** Table 7 compares the setting of SFDA to the related settings of fine-tuning, unsupervised domain adaptation (UDA), and domain generalization (DG).

Table 7: Source-free domain adaptation and related settings. Adapted from Wang et al. (2021).

| Setting | Source data | Target data | Adapt. Loss |
|---|---|---|---|
| Fine-tuning | - | $x^t, y^t$ | $L(x^t, y^t)$ |
| UDA | $x^s, y^s$ | $x^t$ | $L(x^s, y^s) + L(x^s, x^t)$ |
| Domain gen. | $x^s, y^s$ | - | $L(x^s, y^s)$ |
| Source-free DA | - | $x^t$ | $L(x^t)$ |

# F    DATASETS

Figures 5, 6, 7, 8 and 9 below visualize the different datasets we use for evaluation and analysis.

MNIST-M (Ganin et al., 2016) is constructed by combining digits from MNIST with random background colour patches from BSDS500 (Arbelaez et al., 2011). The source domain is standard MNIST and the target domain is the same digits coloured (see Figure 5). MNIST-C (Mu & Gilmer, 2019) contains 15 different corruptions of the MNIST digits. Again, the source domain is standard MNIST and the corruptions of the same digits make up the 15 possible target domains (see Figure 6).

As shown in Appendix K.1 many methods achieve good performance on these MNIST datasets. For this reason we create and release the more challenging EMNIST-DA dataset. EMNIST-DA contains 13 different shifts chosen to give a diverse range of initial accuracies when using a source model trained on standard EMNIST. In particular, a number of shifts result in very low initial performance but are conceptually simple to resolve (see Figure 7). Here, models are trained on the training set of EMNIST (source) before being adapted to a shifted test set of EMNIST-DA (target, unseen examples).

We also use the CIFAR-10-C and CIFAR-100-C corruption datasets (Hendrycks & Dietterich, 2019) to compare methods on object-recognition tasks. These datasets contain 19 different corruptions of the CIFAR-10 and CIFAR-100 test sets (see Figure 8). Here, a model is trained on the training set of CIFAR-10/CIFAR-100 (source, Krizhevsky 2009) before being adapted to a corrupted test set (target).

Finally, we show real-world measurement shift with CAMELYON17 (Bandi et al., 2018), a medical dataset with histopathological images from 5 different hospitals which use different staining and imaging techniques (Figure 9). The goal is to determine whether or not an image contains tumour tissue. We train on examples from a single source hospital (hospital 3) before adapting to one of the 4 remaining target hospitals. We use the WILDS (Koh et al., 2021) implementation of CAMELYON17.

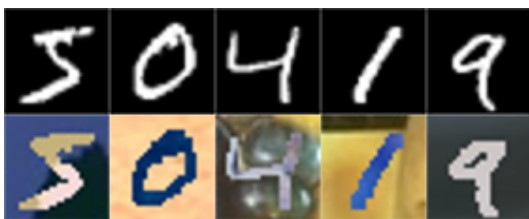

Figure 5: *Top:* samples from MNIST. *Bottom:* samples from MNIST-M.

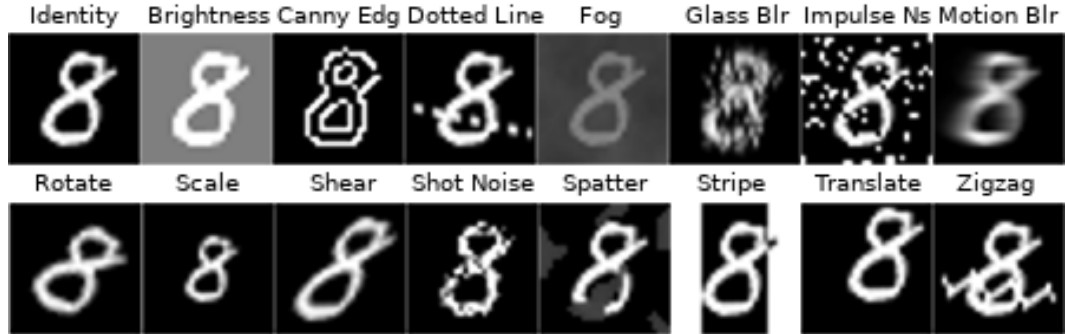

Figure 6: MNIST-C corruptions.

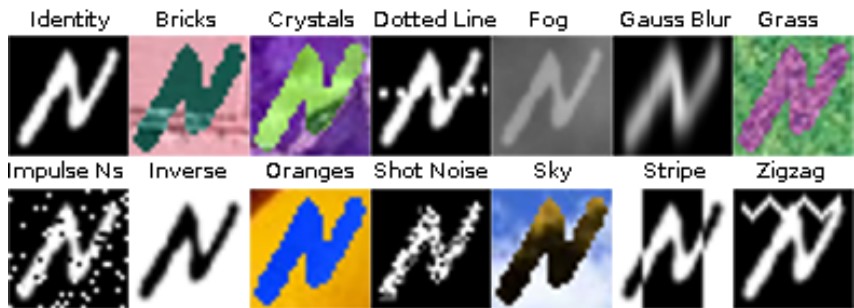

Figure 7: EMNIST-DA shifts.

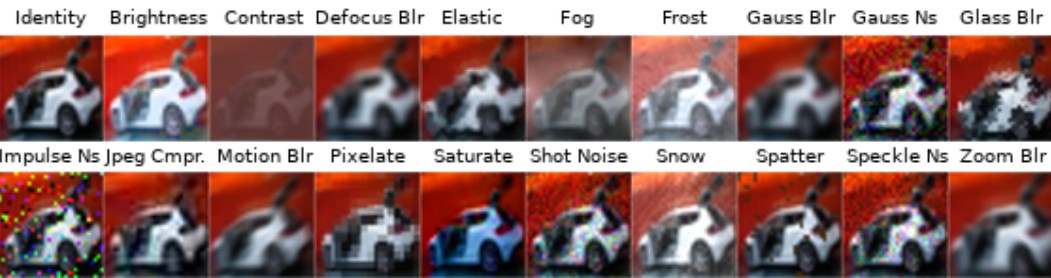

Figure 8: CIFAR corruptions. The same corruptions are used for CIFAR-10-C and CIFAR-100-C.

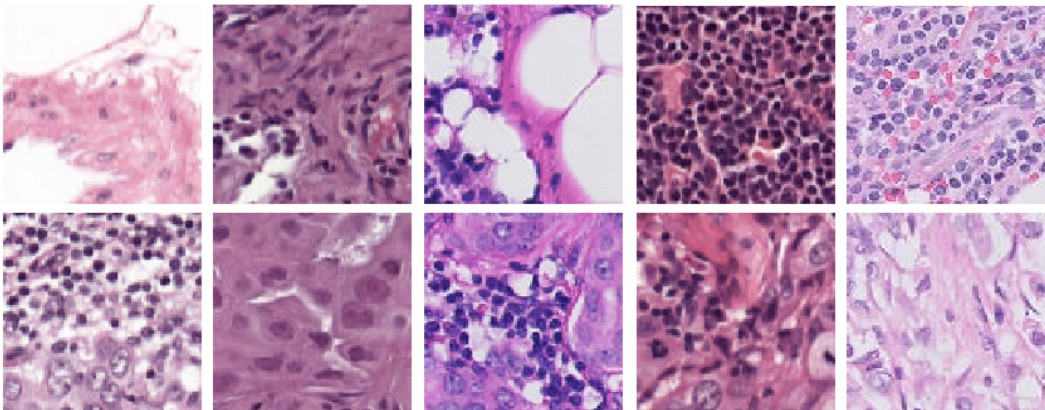

Figure 9: CAMELYON17. Columns show different hospitals. *Top row:* no tumour tissue. *Bottom row:* tumour tissue present.

## G    FURTHER IMPLEMENTATION DETAILS

**Architectures.**    The architecture of the simple 5-layer CNN (a variant of LeNet, LeCun et al. 1998), which we use for digit and character datasets, is provided in Table 8. For the object-recognition and medical datasets, we use a standard ResNet-18 (He et al., 2016).

**Training details.**    For all datasets and methods we train using SGD with momentum set to $0.9$, use a batch size of $256$, and report results over $5$ random seeds. In line with previous UDA & SFDA works (although often not made explicit), we use a test-domain validation set for model selection (Gulrajani & Lopez-Paz, 2021). In particular, we select the best-performing learning rate from $\{0.0001, 0.001, 0.01, 0.1, 1\}$, and for BUFR, we train for 30 epochs per block and decay the learning rate as a function of the number of unfrozen blocks in order to further maintain structure. For all other methods, including FR, we train for $150$ epochs with a constant learning rate. The temperature parameter $\tau$ (see Appendix A, Eq. 4) is set to $0.01$ in all experiments.

**Tracking feature and logit distributions.**    To track the marginal feature and logit distributions, we implement a simple `StatsLayer` class in PyTorch that can be easily inserted into a network just like any other layer. This seamlessly integrates distribution-tracking into standard training processes. In the source domain, we simply: (i) add `StatsLayer`s to our (pre)trained source model; (ii) pass the source data through the model; and (iii) save the model as normal in PyTorch (the tracked statistics, i.e. bin counts, are automatically saved as persistent buffers akin to BN-statistics). In the target domain, the source model can be loaded as normal and the inserted `StatsLayer`s will contain the source-data statistics. Code is available at `https://github.com/cianeastwood/bufr`.

**The Full Gauss. baseline.**    This baseline models the distribution of hidden features as a joint multivariate Gaussian, with dimensionality equal to the number of hidden units. After training a model on the source data, the source data is passed through once more and the empirical mean vector and covariance matrix are calculated and saved. To adapt to the target data the empirical mean and covariances are calculated for each minibatch and the distributions are aligned using the KL divergence $D_{KL}(Q||P)$, where $Q$ is the Gaussian distribution estimated on the target data minibatch and $P$ from the source data. This divergence has an analytic form (Duchi, 2007, Sec. 9) which we use as the loss function. We use this direction for the KL divergence as we only need to invert the covariance matrix once (for saved $P$) rather than the covariance matrix for $Q$ on every batch.

**Online setup.**    In the online setting, where only a single epoch is permitted, we find that all methods are very sensitive to the learning rate (unsurprising, given that most methods will not have converged after a single epoch). For fair comparison, we thus search over learning rates in $\{0.1, 0.01, 0.001, 0.0001\}$ for all methods, choosing the best-performing one. Additionally, when learning speed is of critical importance, we find it beneficial to slightly increase $\tau$. We thus set $\tau = 0.05$ for all online experiments, compared to $0.01$ for all "offline" experiments.

Table 8: Architecture of the CNN used on digit and character datasets. For conv. layers, the weights-shape is: *num. input channels $\times$ num. output channels $\times$ filter height $\times$ filter width*.

| Block | Weights-Shape | Stride | Padding | Activation | Dropout Prob. |
|---|---|---|---|---|---|
| Conv + BN | $3 \times 64 \times 5 \times 5$ | 2 | 2 | ReLU | 0.1 |
| Conv + BN | $64 \times 128 \times 3 \times 3$ | 2 | 2 | ReLU | 0.3 |
| Conv + BN | $128 \times 256 \times 3 \times 3$ | 2 | 2 | ReLU | 0.5 |
| Linear + BN | $6400 \times 128$ | N/A | N/A | ReLU | 0.5 |
| Linear | $128 \times$ Number of Classes | N/A | N/A | Softmax | 0 |

# H    RELIABILITY DIAGRAMS AND CONFIDENCE HISTOGRAMS

This section shows reliability diagrams (DeGroot & Fienberg, 1983; Niculescu-Mizil & Caruana, 2005) and confidence histograms (Zadrozny & Elkan, 2001): (i) over all EMNIST-DA shifts (see Figure 10); (ii) a severe EMNIST-DA shift (see Figure 11); and (iii) a mild shift EMNIST-DA shift (see Figure 12). Reliability diagrams are given along with the corresponding Expected Calibration Error (ECE, Naeini et al. 2015) and Maximum Calibration Error (MCE, Naeini et al. 2015). ECE is calculated by binning predictions into 10 evenly-spaced bins based on confidence, and then taking a weighted average of the absolute difference between average accuracy and average confidence of the samples in each bin. MCE is the maximum absolute difference between average accuracy and average confidence over the bins. In Figures 10–12 below, we pair each reliability diagram with the corresponding confidence histogram, since reliability diagrams do not provide the underlying frequencies of each bin (as in Guo et al. 2017, Figure 1).

In general we see that most models are overconfident, but our models much less so. As seen by the difference in the size of the red 'Gap' bar in the rightmost bins of Figures 10b, 10c, and 10d, when our FR methods predict with high confidence they are much more likely to be correct than IM—a method which works by maximizing prediction confidence. Figure 11 shows that BUFR remains well-calibrated even when the initial shift is severe. Figure 12 shows that, even for a mild shift when all models achieve high accuracy, our methods are better-calibrated. Note that the label 'Original' in Figures 10a and 10e denotes the source model on the *source data*, while 'Source-only' in Figures 11a, 11e, 12a, and 12e denotes the source model on the *target data*.

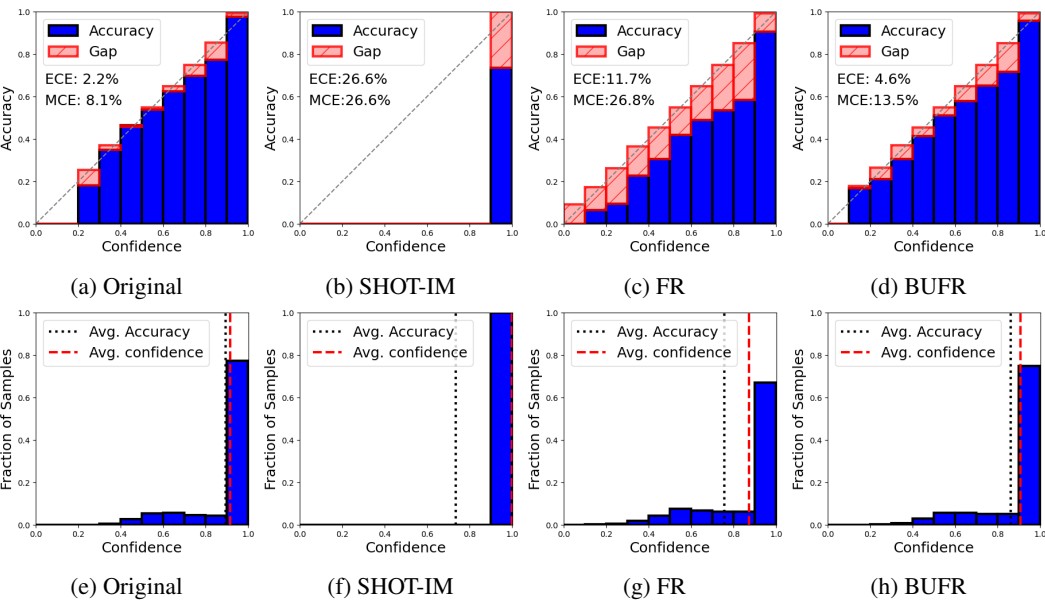

Figure 10: Reliability diagrams and confidence histograms over *all* EMNIST-DA corruptions. *(a–d):* Reliability diagrams showing the difference between average accuracy and average confidence for different methods. *(e–h):* Confidence histograms showing the frequency with which predictions are made with a given confidence. Each confidence histogram corresponds with the reliability diagram above it. *(a & e):* The source model is well-calibrated on the *source data*. *(b & f):* Entropy-minimization leads to extreme overconfidence. *(c & g, d & h):* Our methods, FR and BUFR, are much better-calibrated as they do not work by making predictions more confident.

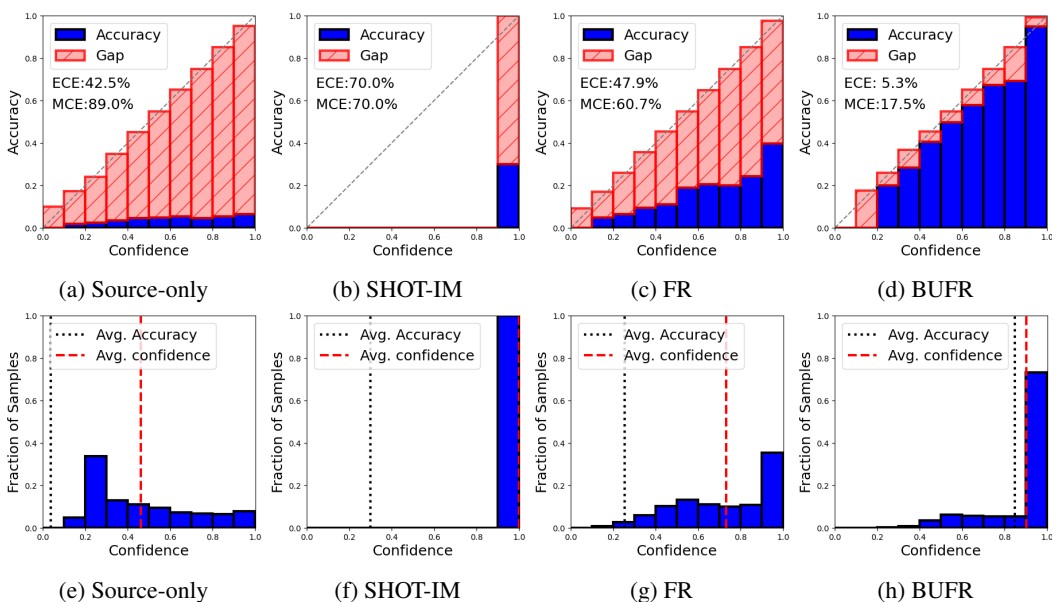

Figure 11: Reliability diagrams and confidence histograms for a severe EMNIST-DA shift (sky) where all methods except BUFR achieve poor accuracy. Each confidence histogram corresponds with the reliability diagram above it. *(a & e)*: Source model on the *target data* achieves poor accuracy and often predicts with low confidence. *(b & f)*: SHOT-IM also achieves poor accuracy but is highly confident. *(d & h)*: Our BUFR method achieves better ECE and MCE than all other methods.

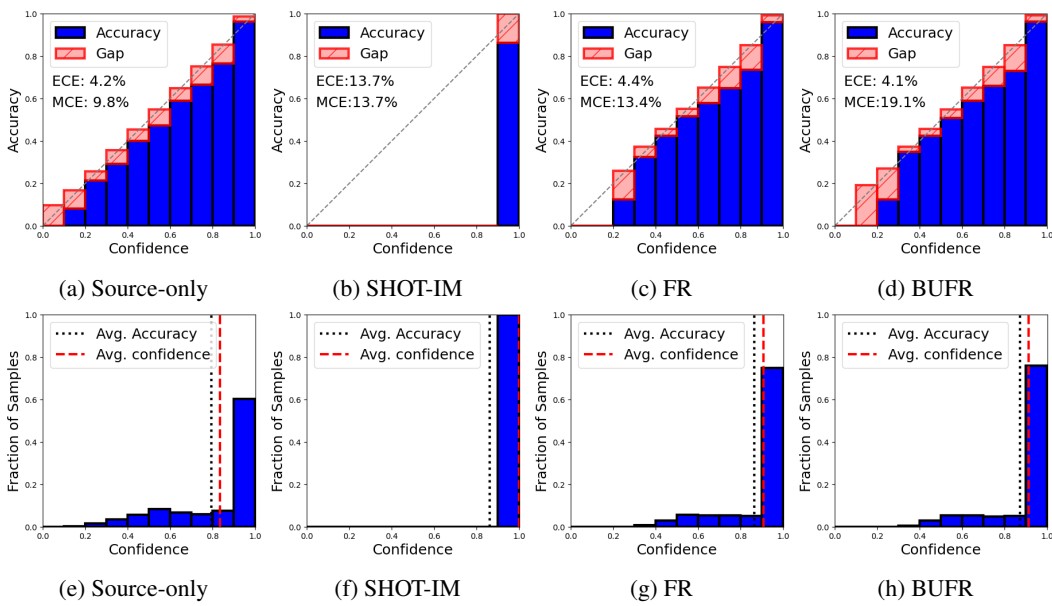

Figure 12: Reliability diagrams and confidence histograms for a mild EMNIST-DA shift (shot noise) where all methods achieve good accuracy. Each confidence histogram corresponds with the reliability diagram above it. When highly confident, our methods *(d & h)* are more often correct than IM *(b & f)*.

# I ACTIVATION DISTRIBUTIONS

**EMNIST-DA (skewed).** Figure 13 depicts histograms of the marginal feature and logit activation-distributions on the EMNIST-DA stripe shift. As shown, the marginal distributions on the source data (blue curve, those we wish to match) may be heavily-skewed. In contrast, the marginal distributions on the target data (*before adapting*, orange curve) tend to be more symmetric but have a similar mean.

**CIFAR-10 (bi-modal).** Figure 14 depicts histograms of the marginal feature and logit activation-distributions on the CIFAR-10-C impulse-noise shift. As shown, the marginal distributions on the source data (blue curve, those we wish to match) tend to be bi-modal. In contrast, the marginal distributions on the target data (*before adapting*, orange curve) tend to be uni-modal but have a similar mean. The two modes can be interpreted intuitively as "detected" and "not detected" or "present" and "not present" for a given feature-detector.

**Alignment after adapting.** Figure 15 shows histograms of the marginal feature activation-distributions on the EMNIST-DA stripe shift. This figure shows curves on the source data (blue curve, same as Figure 13a) and on the target data (*after adapting*, orange curve) for different methods. Evidently, our FR loss causes the marginal distributions to closely align (Figure 15c). In contrast, competing methods (Figures 15a, 15b) do not match the feature activation-distributions, even if they achieve high accuracy. Figure 16 shows the same trend for CIFAR-10-C.

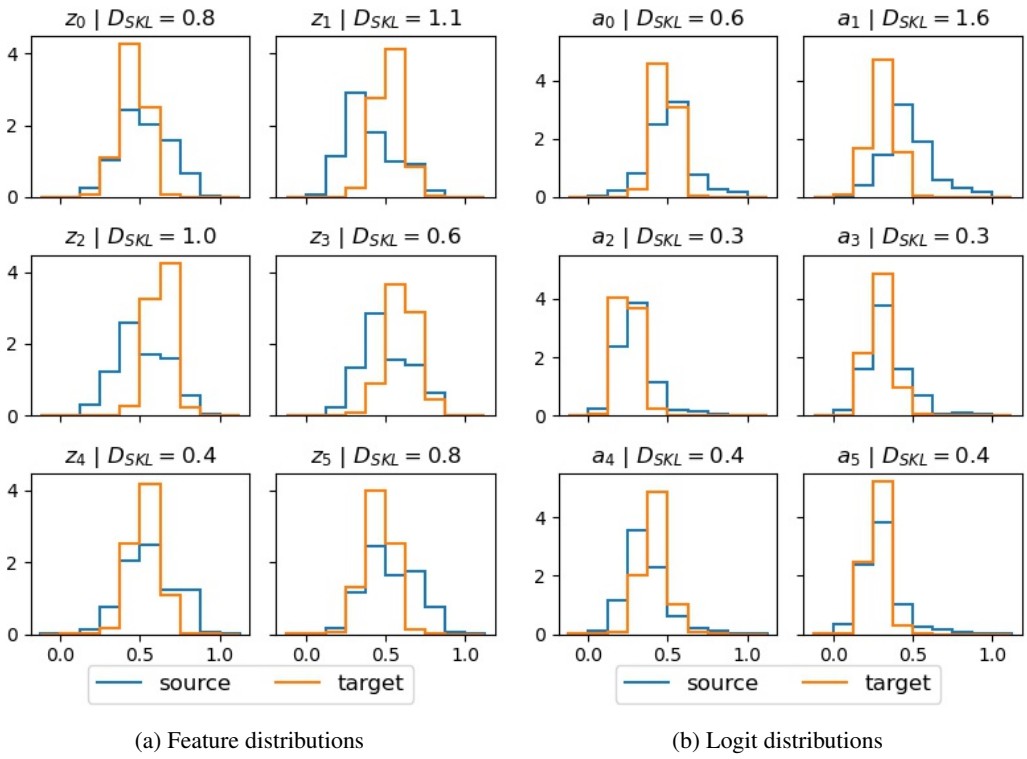

(a) Feature distributions                    (b) Logit distributions

Figure 13: Histograms showing the first 6 marginal activation-distributions on the EMNIST-DA stripe shift. The blue curves are the saved marginal distributions under the source data (i.e. EMNIST). The orange curves are the marginal distributions under the target data *before adaptation* (i.e. the stripe shift). (a) Marginal feature activation-distributions. (b) Marginal logit activation-distributions. $D_{SKL}$ denotes the symmetric KL divergence.

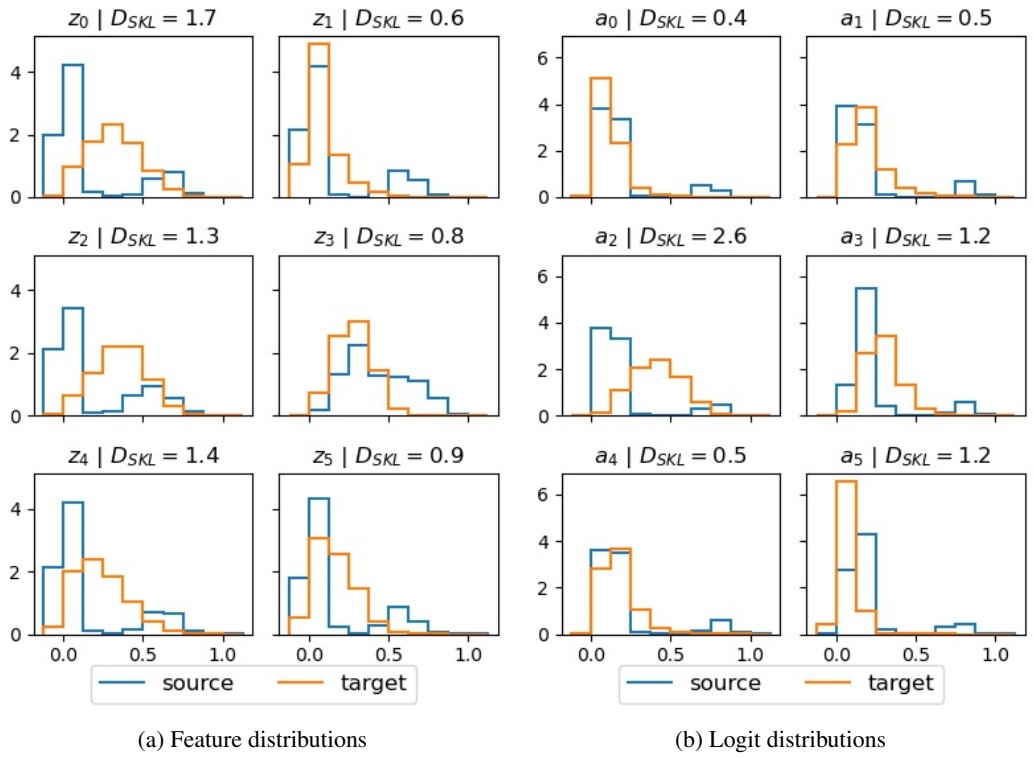

(a) Feature distributions          (b) Logit distributions

Figure 14: Histograms showing the first 6 marginal activation-distributions on the CIFAR-10-C impulse-noise shift. The blue curves are the saved marginal distributions under the source data (i.e. CIFAR-10). The orange curves are the marginal distributions under the target data *before adaptation* (i.e. the impulse-noise shift). (a) Marginal feature activation-distributions and (b) Marginal logit activation-distributions. $D_{SKL}$ denotes the symmetric KL divergence.

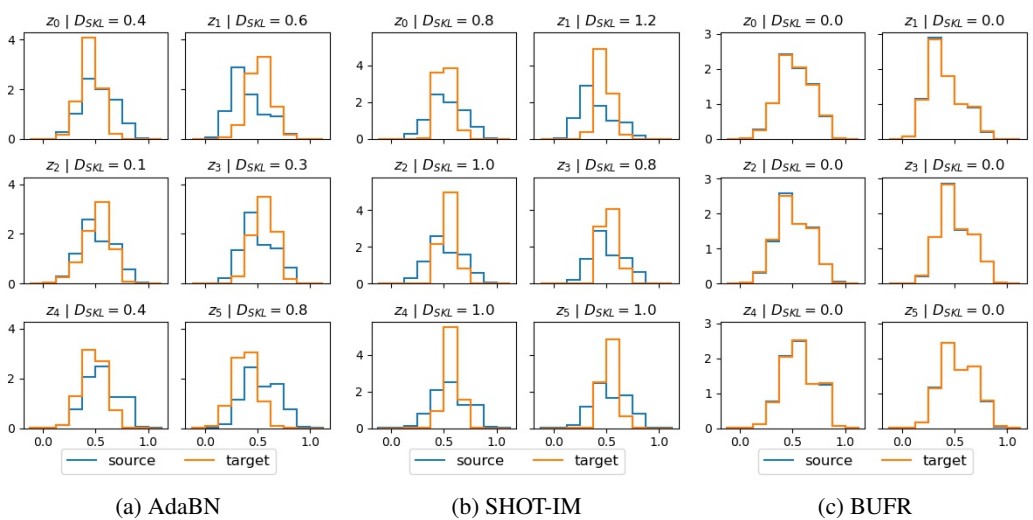

(a) AdaBN          (b) SHOT-IM          (c) BUFR

Figure 15: Histograms showing distribution-alignment on the EMNIST-DA stripe shift. The blue curves are the saved marginal distributions under the source data (i.e. EMNIST). The orange curves are the marginal distributions under the target data *after adaptation* (to the stripe shift). (a,b): AdaBN and SHOT-IM do not align the marginal distributions (despite achieving reasonable accuracy—see Table 17). (c) BUFR matches the activation-distributions very closely, making $D_{SKL}$ very small.

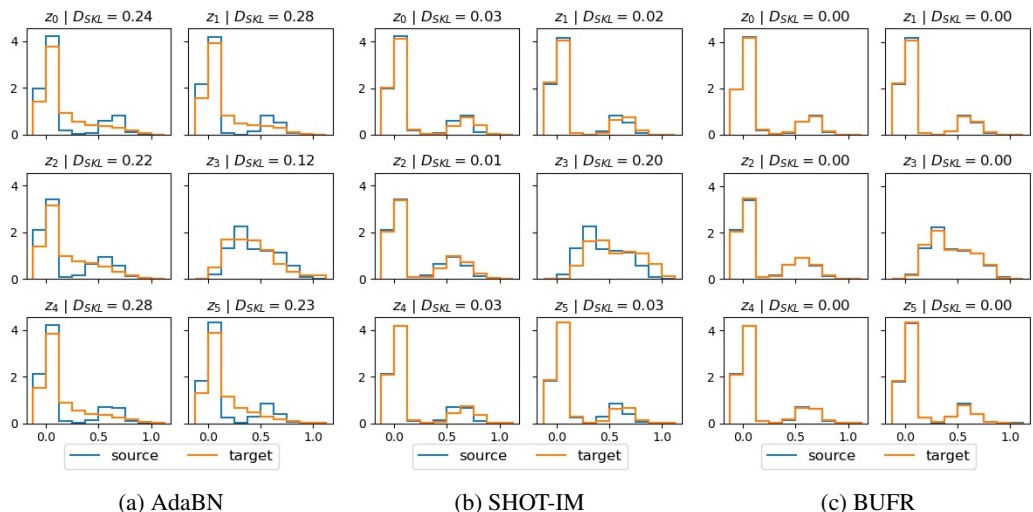

Figure 16: Histograms showing distribution-alignment on the CIFAR-10-C impulse-noise shift. The blue curves are the saved marginal distributions under the source data (i.e. CIFAR-10). The orange curves are the marginal distributions under the target data *after adaptation* (to the impulse-noise shift). (a) AdaBN does not align the marginal distributions. (b) SHOT-IM only partially-aligns the marginal distributions. (c) BUFR matches the activation-distributions very closely, making $D_{SKL}$ very small.

## J   FURTHER ANALYSIS

### J.1   EFFICACY OF BOTTOM-UP TRAINING

Table 9 reports EMNIST-DA accuracy vs. the number of (unlabelled) examples-per-class available in the target domain. BUFR retains strong performance even with only 5 examples-per-class.

Table 9: EMNIST-DA accuracy vs. examples-per-class.

| Model | 5 | 10 | 20 | 50 | 400 |
|---|---|---|---|---|---|
| Marg. Gauss. (Ishii & Sugiyama, 2021) | 49.3 | 49.9 | 50.4 | 50.7 | 50.6 |
| Full Gauss. | 55.4 | 59.7 | 61.0 | 63.3 | 68.3 |
| PL (Lee et al., 2013) | 45.8 | 46.3 | 46.0 | 46.7 | 49.7 |
| BNM-IM (Ishii & Sugiyama, 2021) | 50.5 | 51.5 | 53.0 | 54.7 | 61.4 |
| SHOT-IM (Liang et al., 2020) | 48.3 | 51.7 | 51.2 | 54.7 | 73.4 |
| FR (ours) | 50.8 | 50.5 | 60.1 | 63.1 | 75.6 |
| BUFR (ours) | **78.0** | **82.3** | **83.8** | **84.9** | **86.2** |

### J.2   LOSS ABLATION STUDY

Table 10 reports the performance of our FR loss on CIFAR-10-C and CIFAR-100-C without: (i) aligning the logit distributions; and (ii) using the symmetric KL divergence (we instead use the asymmetric reverse KL). While these components make little difference on the easier task of CIFAR-10-C, they significantly improve performance on the harder task of CIFAR-100-C.

Table 10: Ablation study of $\mathcal{L}_{tgt}$ in Eq. 2.

| Model | CFR-10-C | CFR-100-C |
|---|---|---|
| $\mathcal{L}_{tgt}$ w/o logits | $86.7 \pm 0.2$ | $62.3 \pm 1.3$ |
| $\mathcal{L}_{tgt}$ w/o $D_{KL}(P\|\|Q)$ | $86.5 \pm 0.3$ | $61.5 \pm 0.2$ |
| $\mathcal{L}_{tgt}$ | $\mathbf{87.2 \pm 0.7}$ | $\mathbf{65.5 \pm 0.2}$ |

## J.3 WHO IS AFFECTED

We now analyse which layers are most affected by a measurement shift. Figure 17 shows the (symmetric) KL divergence between the unit-level activation distributions under the source (EMNIST) and target (EMNIST-DA crystals) data *before adapting* (17a) and *after adapting the first layer* (17b). Figure 17a shows that, before adapting, the unit-activation distributions in all layers of the network have changed significantly, as indicated by the large KL divergences. Figure 17b shows that, after updating just the first layer, "normality" is restored in all subsequent layers, with the unit-level activation distributions on the target data realigning with those saved on the source (shown via very low KL divergences). This indicates that measurement shifts primarily affect the first layer/block—since they can be mostly resolved by updating the first layer/block—and also further motivates bottom-up training for measurement shifts.

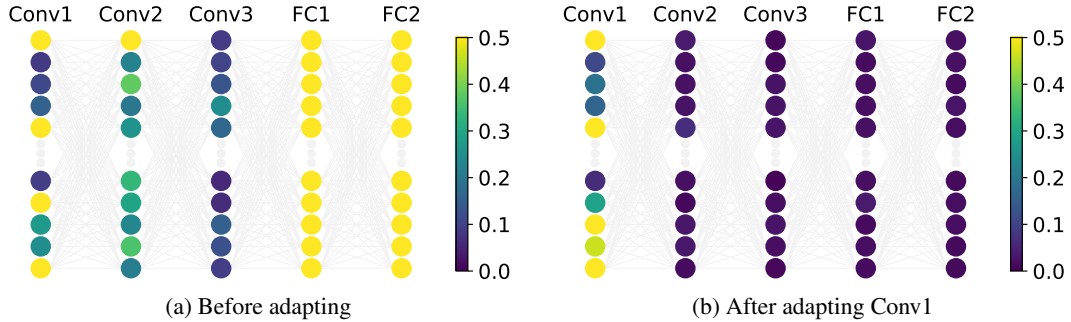

(a) Before adapting        (b) After adapting Conv1

Figure 17: Symmetric KL divergence between the unit-level activation distributions under the source (EMNIST) and target (EMNIST-DA crystals) data: (a) before adapting; and (b) after adapting only the first layer (Conv1). For visual clarity, we show only 10 sample units per layer.

## J.4 WHO MOVES

We now analyse which layers are most updated by BUFR. Figure 18a shows that, on average, FR moves the weights of all layers of $g_t$ a similar distance when adapting to the target data. Figure 18b shows that BUFR primarily updates the early layers, thus preserving learnt structure in later layers.

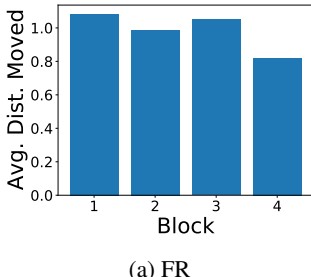 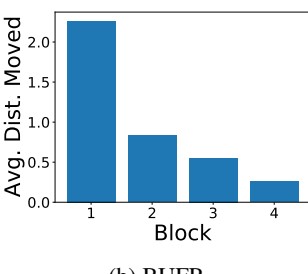

(a) FR                     (b) BUFR

Figure 18: Average distance moved by a unit in each block of $g_t$ on the EMNIST-DA stripe shift when training (a) all layers at once and (b) in a bottom-up manner. Both methods are trained with the same constant learning rate.

# K    FULL RESULTS

In this section we give the full results for all datasets and constituent domains.

## K.1    DIGIT AND CHARACTER SUMMARY RESULTS

The simplest datasets we use are variations of the MNIST dataset (LeCun et al., 1998). Here, a model is trained on MNIST (source domain) before being adapted to MNIST-M (Ganin et al., 2016) or one of the fifteen MNIST-C (Mu & Gilmer, 2019) corruptions (target domain). As mentioned in Section 5, the MNIST-based shifts can be well-resolved by a number of methods.

Tables 11 and 12 summarize the accuracy and ECEs across different models for the digit and character datasets. On MNIST-C, where source-only accuracy is very high, all methods achieve good results (accuracy $\geq 95\%$)—providing limited insight into their relative performances. On MNIST-M, our BUFR method outperforms all baselines, although SHOT is very similar in performance. As discussed in Section 5, our BUFR method outperforms all baseline methods on EMNIST-DA in terms of accuracy *and* ECE as it does not work by making predictions more confident.

Table 11: Digit and character accuracy (%) results. Shown are the mean and 1 standard deviation. EMNIST-DA: mean performance over all 13 EMNIST-DA shifts. EMNIST-DA-SVR & EMNIST-DA-MLD: sample "severe" and "mild" shifts from EMNIST-DA selected based on AdaBN performance.

| Model | MNIST-C | MNIST-M | EMNIST-DA | EMNIST-DA-SVR | EMNIST-DA-MLD |
|---|---|---|---|---|---|
| No corruption | $99.5 \pm 0.1$ | $99.5 \pm 0.1$ | $89.4 \pm 0.1$ | $89.4 \pm 0.1$ | $89.4 \pm 0.1$ |
| Source-only | $86.2 \pm 1.8$ | $42.7 \pm 4.6$ | $29.5 \pm 0.5$ | $3.8 \pm 0.4$ | $78.5 \pm 0.7$ |
| AdaBN (Li et al., 2018) | $94.2 \pm 0.2$ | $59.1 \pm 1.9$ | $46.2 \pm 1.1$ | $3.7 \pm 0.7$ | $84.9 \pm 0.2$ |
| PL (Lee et al., 2013) | $96.4 \pm 0.4$ | $43.1 \pm 2.1$ | $50.0 \pm 0.6$ | $2.7 \pm 0.4$ | $83.5 \pm 0.1$ |
| SHOT-IM (Liang et al., 2020) | $97.3 \pm 0.2$ | $66.9 \pm 9.3$ | $70.3 \pm 3.7$ | $24.0 \pm 7.5$ | $86.3 \pm 0.1$ |
| SHOT (Liang et al., 2020) | $\mathbf{97.7 \pm 0.2}$ | $94.4 \pm 3.1$ | $80.0 \pm 4.4$ | $55.1 \pm 23.5$ | $86.1 \pm 0.1$ |
| FR (ours) | $96.7 \pm 0.1$ | $86.5 \pm 0.6$ | $74.4 \pm 0.8$ | $15.3 \pm 6.8$ | $86.4 \pm 0.1$ |
| BUFR (ours) | $96.4 \pm 0.6$ | $\mathbf{96.2 \pm 1.7}$ | $\mathbf{86.1 \pm 0.1}$ | $\mathbf{84.6 \pm 0.2}$ | $\mathbf{87.0 \pm 0.2}$ |
| Target-supervised | $99.3 \pm 0.0$ | $98.5 \pm 0.0$ | $86.8 \pm 0.6$ | $85.7 \pm 0.6$ | $87.3 \pm 0.7$ |

Table 12: Digit and character ECE (%) results. Shown are the mean and 1 standard deviation. EMNIST-DA: mean performance over all 13 EMNIST-DA shifts. EMNIST-DA-SVR & EMNIST-DA-MLD: sample "severe" and "mild" shifts from EMNIST-DA selected based on AdaBN performance.

| Model | MNIST-C | MNIST-M | EMNIST-DA | EMNIST-DA-SVR | EMNIST-DA-MLD |
|---|---|---|---|---|---|
| No corruption | $0.3 \pm 0.0$ | $0.3 \pm 0.0$ | $2.3 \pm 0.1$ | $2.3 \pm 0.1$ | $2.3 \pm 0.1$ |
| Source-only | $7.2 \pm 1.4$ | $27.0 \pm 7.1$ | $30.8 \pm 1.6$ | $42.6 \pm 3.5$ | $4.8 \pm 0.5$ |
| AdaBN (Li et al., 2018) | $4.1 \pm 0.1$ | $24.4 \pm 2.8$ | $30.3 \pm 1.1$ | $52.4 \pm 4.9$ | $4.9 \pm 0.3$ |
| PL (Lee et al., 2013) | $3.1 \pm 0.4$ | $56.1 \pm 2.2$ | $49.9 \pm 0.6$ | $97.2 \pm 0.4$ | $16.4 \pm 0.1$ |
| SHOT-IM (Liang et al., 2020) | $2.3 \pm 0.2$ | $30.9 \pm 9.0$ | $29.6 \pm 3.7$ | $76.0 \pm 7.5$ | $13.7 \pm 0.1$ |
| SHOT (Liang et al., 2020) | $\mathbf{2.0 \pm 0.2}$ | $\mathbf{2.8 \pm 2.9}$ | $19.7 \pm 4.4$ | $42.7 \pm 23.0$ | $14.8 \pm 0.1$ |
| FR (ours) | $2.5 \pm 0.2$ | $9.8 \pm 0.8$ | $12.9 \pm 0.9$ | $58.0 \pm 6.8$ | $4.6 \pm 0.3$ |
| BUFR (ours) | $3.0 \pm 0.6$ | $2.9 \pm 1.5$ | $\mathbf{4.7 \pm 0.2}$ | $\mathbf{5.6 \pm 0.3}$ | $\mathbf{4.2 \pm 0.2}$ |
| Target-supervised | $0.5 \pm 0.0$ | $1.1 \pm 0.1$ | $7.3 \pm 0.7$ | $7.0 \pm 0.5$ | $8.4 \pm 1.1$ |

## K.2 ONLINE RESULTS

Table 13 reports the online results for CIFAR-10-C and CIFAR-100-C. FR outperforms existing SFDA methods on CIFAR-10-C in terms of both accuracy and ECE. On CIFAR-100-C, our method is competitive with TENT (Wang et al., 2021)—a method designed specifically for this online setting. As in Wang et al. (2021), these results represent the average over batches *during training* (i.e. a single pass through the target data), rather than the average at the end of training, in order to evaluate *online* performance. We omit BUFR from this table as it is not easily applicable to the online setting—it is difficult to set the number of steps per block without information on the total number of steps/batches (generally not available in an online setting). Full per-shift results for this online setting are given in Tables 23 and 24 for CIFAR-10-C, and Tables 25 and 26 for CIFAR-100-C.

Table 13: Online results. Shown are the mean and 1 standard deviation.

| Model | CIFAR-10-C | | CIFAR-100-C | |
|---|---|---|---|---|
| | ACC ↑ | ECE ↓ | ACC ↑ | ECE ↓ |
| AdaBN (Li et al., 2018) | $80.3 \pm 0.0$ | $12.1 \pm 0.0$ | $56.6 \pm 0.3$ | $\mathbf{10.0 \pm 0.1}$ |
| SHOT-IM (Liang et al., 2020) | $83.2 \pm 0.2$ | $10.9 \pm 0.1$ | $62.3 \pm 0.3$ | $13.8 \pm 0.1$ |
| TENT (Wang et al., 2021) | $81.8 \pm 0.2$ | $11.5 \pm 0.1$ | $\mathbf{63.1 \pm 0.3}$ | $14.3 \pm 0.1$ |
| FR (ours) | $\mathbf{85.9 \pm 0.3}$ | $\mathbf{9.5 \pm 0.2}$ | $62.7 \pm 0.3$ | $13.6 \pm 0.1$ |

## K.3 CAMELYON17 RESULTS

Table 14 reports the accuracy and ECE results for CAMELYON17. With up to 50 target examples-per-class: (i) our methods reduce the error rate by approximately 20% compared to the next best method; (ii) only our methods meaningfully improve upon the simple AdaBN baseline which uses the target-data BN-statistics (i.e. neither PL or SHOT-IM actually work). With up to 500 target examples-per-class, our methods reduce the error rate by approximately 20% compared to the next best method. With over 15,000 examples-per-class, our methods are competitive with existing ones.

Table 14: CAMELYON17 results for different numbers of (unlabelled) examples-per-class in the target domain.

| Model | 5 | | 50 | | 500 | | >15k | |
|---|---|---|---|---|---|---|---|---|
| | ACC ↑ | ECE ↓ | ACC ↑ | ECE ↓ | ACC ↑ | ECE ↓ | ACC ↑ | ECE ↓ |
| Source-only | $55.8 \pm 1.6$ | $40.8 \pm 2.1$ | $55.8 \pm 1.6$ | $40.8 \pm 2.1$ | $55.8 \pm 1.6$ | $40.8 \pm 2.1$ | $55.8 \pm 1.6$ | $40.8 \pm 2.1$ |
| AdaBN | $82.6 \pm 2.2$ | $14.7 \pm 2.1$ | $83.7 \pm 1.0$ | $13.7 \pm 0.8$ | $83.9 \pm 0.8$ | $13.5 \pm 0.7$ | $84.0 \pm 0.5$ | $13.5 \pm 0.5$ |
| PL | $82.5 \pm 2.0$ | $14.2 \pm 1.1$ | $83.6 \pm 1.2$ | $13.8 \pm 1.0$ | $85.0 \pm 0.8$ | $13.0 \pm 0.8$ | $\mathbf{90.6 \pm 0.9}$ | $\mathbf{8.8 \pm 0.9}$ |
| SHOT-IM | $82.6 \pm 2.2$ | $13.8 \pm 1.8$ | $83.7 \pm 1.2$ | $13.8 \pm 1.1$ | $86.4 \pm 0.7$ | $11.9 \pm 0.7$ | $89.9 \pm 0.2$ | $9.7 \pm 0.2$ |
| FR (ours) | $\mathbf{84.6 \pm 0.6}$ | $12.9 \pm 0.5$ | $86.0 \pm 1.1$ | $12.1 \pm 1.1$ | $89.0 \pm 0.6$ | $\mathbf{9.7 \pm 0.6}$ | $89.5 \pm 0.4$ | $9.8 \pm 0.5$ |
| BUFR (ours) | $84.5 \pm 0.8$ | $\mathbf{12.8 \pm 0.8}$ | $\mathbf{87.0 \pm 1.2}$ | $\mathbf{11.1 \pm 1.1}$ | $\mathbf{89.1 \pm 0.8}$ | $\mathbf{9.7 \pm 0.8}$ | $89.7 \pm 0.5$ | $9.6 \pm 0.6$ |

## K.4 MNIST-C FULL RESULTS

Tables 15 and 16 show the accuracy and ECE results for each individual corruption of the MNIST-C dataset. We provide the average performance with and without the translate corruption as the assumptions behind the methods that rely on a fixed classifier $h$ no longer hold. Without the translate corruption (Avg. \translate) we see that all methods achieve high accuracy ($\geq 95\%$).

Table 15: MNIST-C accuracy (%) results. Shown are the mean and 1 standard deviation.

|  | Src-only | AdaBN | PL | SHOT-IM | SHOT | FR | BUFR |
|---|---|---|---|---|---|---|---|
| Brightness | $84.8 \pm 11.4$ | $99.4 \pm 0.0$ | $99.5 \pm 0.0$ | $99.4 \pm 0.1$ | $99.5 \pm 0.1$ | $98.7 \pm 0.1$ | $99.2 \pm 0.1$ |
| Canny Edges | $72.2 \pm 0.8$ | $91.0 \pm 0.7$ | $96.2 \pm 1.0$ | $98.1 \pm 0.1$ | $98.6 \pm 0.1$ | $97.8 \pm 0.1$ | $98.5 \pm 0.0$ |
| Dotted Line | $98.6 \pm 0.2$ | $98.8 \pm 0.2$ | $99.3 \pm 0.1$ | $99.4 \pm 0.1$ | $99.4 \pm 0.1$ | $98.6 \pm 0.1$ | $99.1 \pm 0.0$ |
| Fog | $30.1 \pm 12.5$ | $93.9 \pm 1.9$ | $99.3 \pm 0.0$ | $99.4 \pm 0.1$ | $99.5 \pm 0.0$ | $98.6 \pm 0.1$ | $99.2 \pm 0.0$ |
| Glass Blur | $88.9 \pm 2.3$ | $95.3 \pm 0.3$ | $97.8 \pm 0.1$ | $98.2 \pm 0.1$ | $98.3 \pm 0.0$ | $97.2 \pm 0.1$ | $97.9 \pm 0.0$ |
| Impulse Noise | $95.2 \pm 0.6$ | $97.9 \pm 0.1$ | $98.4 \pm 0.1$ | $98.7 \pm 0.1$ | $98.9 \pm 0.1$ | $97.8 \pm 0.0$ | $98.6 \pm 0.0$ |
| Motion Blur | $85.6 \pm 3.9$ | $97.3 \pm 0.4$ | $98.8 \pm 0.1$ | $99.1 \pm 0.1$ | $99.2 \pm 0.1$ | $98.1 \pm 0.1$ | $98.8 \pm 0.1$ |
| Rotate | $96.7 \pm 0.1$ | $96.7 \pm 0.0$ | $97.7 \pm 0.1$ | $98.4 \pm 0.1$ | $98.8 \pm 0.0$ | $97.5 \pm 0.1$ | $97.9 \pm 0.1$ |
| Scale | $97.2 \pm 0.1$ | $97.2 \pm 0.1$ | $98.7 \pm 0.1$ | $99.1 \pm 0.0$ | $99.2 \pm 0.0$ | $98.0 \pm 0.0$ | $98.7 \pm 0.2$ |
| Shear | $98.9 \pm 0.1$ | $98.9 \pm 0.0$ | $99.0 \pm 0.0$ | $99.1 \pm 0.0$ | $99.2 \pm 0.0$ | $98.3 \pm 0.1$ | $98.8 \pm 0.1$ |
| Shot Noise | $98.6 \pm 0.0$ | $99.0 \pm 0.0$ | $99.2 \pm 0.0$ | $99.2 \pm 0.1$ | $99.2 \pm 0.0$ | $98.3 \pm 0.2$ | $99.0 \pm 0.1$ |
| Spatter | $98.7 \pm 0.1$ | $98.8 \pm 0.1$ | $99.0 \pm 0.1$ | $99.0 \pm 0.0$ | $99.1 \pm 0.1$ | $98.4 \pm 0.1$ | $98.8 \pm 0.0$ |
| Stripe | $91.1 \pm 1.2$ | $90.9 \pm 1.5$ | $97.9 \pm 1.0$ | $99.2 \pm 0.0$ | $99.4 \pm 0.1$ | $98.3 \pm 0.1$ | $99.1 \pm 0.1$ |
| Translate | $64.6 \pm 0.5$ | $64.4 \pm 0.6$ | $69.5 \pm 0.8$ | $75.1 \pm 4.1$ | $78.7 \pm 3.1$ | $76.7 \pm 2.1$ | $64.5 \pm 8.9$ |
| Zigzag | $91.8 \pm 0.6$ | $93.0 \pm 0.2$ | $98.2 \pm 0.2$ | $98.9 \pm 0.1$ | $99.2 \pm 0.1$ | $98.2 \pm 0.1$ | $98.8 \pm 0.1$ |
| Avg. | $86.2 \pm 1.8$ | $94.2 \pm 0.2$ | $96.6 \pm 0.1$ | $97.3 \pm 0.2$ | $97.7 \pm 0.2$ | $96.7 \pm 0.1$ | $96.4 \pm 0.6$ |
| Avg.\translate | $87.7 \pm 1.9$ | $96.3 \pm 0.2$ | $98.5 \pm 0.1$ | $98.9 \pm 0.0$ | $99.1 \pm 0.0$ | $98.1 \pm 0.1$ | $98.7 \pm 0.0$ |

Table 16: MNIST-C ECE (%) results. Shown are the mean and 1 standard deviation.

|  | Src-only | AdaBN | PL | SHOT-IM | SHOT | FR | BUFR |
|---|---|---|---|---|---|---|---|
| Brightness | $2.4 \pm 0.4$ | $0.3 \pm 0.1$ | $0.3 \pm 0.1$ | $0.4 \pm 0.1$ | $0.9 \pm 0.1$ | $0.9 \pm 0.1$ | $0.5 \pm 0.1$ |
| Canny Edges | $22.2 \pm 0.7$ | $6.1 \pm 0.7$ | $4.0 \pm 2.5$ | $1.5 \pm 0.1$ | $0.6 \pm 0.1$ | $1.6 \pm 0.1$ | $1.0 \pm 0.1$ |
| Dotted Line | $0.7 \pm 0.1$ | $0.7 \pm 0.1$ | $0.5 \pm 0.1$ | $0.4 \pm 0.1$ | $0.9 \pm 0.0$ | $1.0 \pm 0.1$ | $0.6 \pm 0.1$ |
| Fog | $26.4 \pm 18.0$ | $3.5 \pm 1.4$ | $0.5 \pm 0.0$ | $0.4 \pm 0.0$ | $0.9 \pm 0.1$ | $1.0 \pm 0.1$ | $0.5 \pm 0.1$ |
| Glass Blur | $5.9 \pm 1.6$ | $3.1 \pm 0.3$ | $1.8 \pm 0.1$ | $1.4 \pm 0.1$ | $0.4 \pm 0.2$ | $2.1 \pm 0.1$ | $1.5 \pm 0.1$ |
| Impulse Noise | $1.2 \pm 0.2$ | $1.2 \pm 0.1$ | $1.2 \pm 0.1$ | $0.9 \pm 0.1$ | $0.7 \pm 0.1$ | $1.5 \pm 0.1$ | $1.0 \pm 0.1$ |
| Motion Blur | $9.1 \pm 3.4$ | $1.4 \pm 0.2$ | $1.0 \pm 0.2$ | $0.7 \pm 0.1$ | $0.8 \pm 0.0$ | $1.4 \pm 0.1$ | $0.8 \pm 0.1$ |
| Rotate | $2.0 \pm 0.1$ | $2.2 \pm 0.1$ | $1.9 \pm 0.1$ | $1.2 \pm 0.1$ | $0.6 \pm 0.1$ | $1.9 \pm 0.1$ | $1.5 \pm 0.1$ |
| Scale | $1.0 \pm 0.1$ | $1.7 \pm 0.1$ | $1.0 \pm 0.1$ | $0.7 \pm 0.1$ | $0.8 \pm 0.1$ | $1.5 \pm 0.0$ | $0.8 \pm 0.1$ |
| Shear | $0.7 \pm 0.1$ | $0.7 \pm 0.0$ | $0.8 \pm 0.1$ | $0.7 \pm 0.1$ | $0.8 \pm 0.1$ | $1.2 \pm 0.1$ | $0.9 \pm 0.1$ |
| Shot Noise | $0.7 \pm 0.1$ | $0.6 \pm 0.1$ | $0.5 \pm 0.1$ | $0.5 \pm 0.1$ | $0.8 \pm 0.1$ | $1.2 \pm 0.1$ | $0.7 \pm 0.1$ |
| Spatter | $0.6 \pm 0.1$ | $0.7 \pm 0.1$ | $0.7 \pm 0.1$ | $0.7 \pm 0.1$ | $0.9 \pm 0.1$ | $1.2 \pm 0.1$ | $0.8 \pm 0.0$ |
| Stripe | $4.3 \pm 1.0$ | $6.5 \pm 1.4$ | $2.8 \pm 3.1$ | $0.5 \pm 0.1$ | $0.9 \pm 0.1$ | $1.2 \pm 0.2$ | $0.6 \pm 0.0$ |
| Translate | $25.2 \pm 0.3$ | $28.6 \pm 0.6$ | $29.0 \pm 0.7$ | $24.2 \pm 4.1$ | $19.2 \pm 3.0$ | $18.8 \pm 2.7$ | $33.7 \pm 8.9$ |
| Zigzag | $5.4 \pm 0.5$ | $4.9 \pm 0.3$ | $1.3 \pm 0.2$ | $0.8 \pm 0.0$ | $0.8 \pm 0.0$ | $1.4 \pm 0.1$ | $0.8 \pm 0.1$ |
| Avg. | $7.2 \pm 1.4$ | $4.1 \pm 0.1$ | $3.1 \pm 0.4$ | $2.3 \pm 0.2$ | $2.0 \pm 0.2$ | $2.5 \pm 0.2$ | $3.0 \pm 0.6$ |
| Avg.\translate | $5.9 \pm 1.5$ | $2.4 \pm 0.2$ | $1.3 \pm 0.4$ | $0.8 \pm 0.0$ | $0.8 \pm 0.0$ | $1.4 \pm 0.0$ | $0.8 \pm 0.0$ |

### K.5 EMNIST-DA FULL RESULTS

Tables 17 and 18 show the accuracy and ECE results for each individual shift of EMNIST-DA. We provide the average performance with and without the 'background shifts' (bgs), where the background and digit change colour, as these are often the more severe shifts.

By inspecting Table 17, we see that the sky shift resulted in the lowest AdaBN accuracy, while the shot-noise shift resulted in the highest AdaBN accuracy. Thus, we deem these to be the most and least severe EMNIST-DA shifts, i.e. the "severe" and "mild" shifts. We find AdaBN to be a better indicator of shift severity than source-only as some shifts with poor source-only performance can be well-resolved by simply updating the BN-statistics (no parameter updates), e.g. the fog shift.

Table 17: EMNIST-DA accuracy (%) results. Shown are the mean and 1 standard deviation.

| | Src-only | AdaBN | Marg. Gauss. | Full Gauss. | PL | BNM-IM | SHOT-IM | SHOT | FR | BUFR |
|---|---|---|---|---|---|---|---|---|---|---|
| Bricks | $4.2 \pm 0.5$ | $5.9 \pm 1.0$ | $9.1 \pm 1.2$ | $22.6 \pm 2.1$ | $6.8 \pm 1.2$ | $14.2 \pm 1.4$ | $20.5 \pm 4.8$ | $76.0 \pm 0.2$ | $32.4 \pm 4.8$ | $83.8 \pm 0.3$ |
| Crystals | $19.7 \pm 3.1$ | $42.1 \pm 1.9$ | $50.0 \pm 0.7$ | $60.0 \pm 1.0$ | $47.4 \pm 1.6$ | $61.2 \pm 2.5$ | $71.5 \pm 3.8$ | $80.1 \pm 0.2$ | $76.8 \pm 0.4$ | $82.6 \pm 0.3$ |
| Dotted Line | $76.2 \pm 0.7$ | $80.8 \pm 0.4$ | $82.3 \pm 0.4$ | $82.6 \pm 0.5$ | $80.7 \pm 0.5$ | $86.5 \pm 0.4$ | $87.1 \pm 0.1$ | $87.5 \pm 0.1$ | $87.6 \pm 0.1$ | $88.3 \pm 0.0$ |
| Fog | $4.5 \pm 0.9$ | $69.0 \pm 2.6$ | $77.1 \pm 0.6$ | $85.1 \pm 0.6$ | $77.4 \pm 3.3$ | $86.3 \pm 0.3$ | $86.2 \pm 0.1$ | $87.0 \pm 0.1$ | $87.0 \pm 0.1$ | $88.3 \pm 0.1$ |
| Gaussian Blur | $45.1 \pm 3.2$ | $65.2 \pm 1.6$ | $77.1 \pm 0.8$ | $82.3 \pm 0.2$ | $78.8 \pm 0.8$ | $83.7 \pm 0.3$ | $83.7 \pm 0.2$ | $83.9 \pm 0.2$ | $83.0 \pm 0.1$ | $86.0 \pm 0.1$ |
| Grass | $2.3 \pm 0.1$ | $6.1 \pm 0.4$ | $5.9 \pm 1.1$ | $52.7 \pm 3.5$ | $6.7 \pm 1.8$ | $14.6 \pm 6.1$ | $42.4 \pm 40.9$ | $61.8 \pm 36.5$ | $79.2 \pm 0.3$ | $84.5 \pm 0.2$ |
| Impulse Noise | $36.8 \pm 1.6$ | $76.7 \pm 0.8$ | $81.4 \pm 0.3$ | $82.6 \pm 0.1$ | $79.9 \pm 0.6$ | $84.2 \pm 0.3$ | $84.4 \pm 0.2$ | $84.4 \pm 0.2$ | $84.8 \pm 0.2$ | $86.0 \pm 0.1$ |
| Inverse | $5.6 \pm 0.5$ | $8.1 \pm 2.1$ | $14.1 \pm 6.2$ | $64.4 \pm 4.3$ | $11.3 \pm 2.0$ | $60.4 \pm 23.4$ | $83.2 \pm 0.4$ | $85.1 \pm 0.2$ | $83.1 \pm 0.7$ | $88.3 \pm 0.1$ |
| Oranges | $26.5 \pm 2.7$ | $40.7 \pm 2.3$ | $49.3 \pm 1.0$ | $77.1 \pm 0.6$ | $43.0 \pm 3.1$ | $79.9 \pm 0.6$ | $80.5 \pm 0.3$ | $82.4 \pm 0.2$ | $82.3 \pm 0.3$ | $84.8 \pm 0.3$ |
| Shot Noise | $78.5 \pm 0.7$ | $84.9 \pm 0.2$ | $85.8 \pm 0.3$ | $85.7 \pm 0.2$ | $85.0 \pm 0.3$ | $86.5 \pm 0.1$ | $86.3 \pm 0.1$ | $86.1 \pm 0.1$ | $86.4 \pm 0.1$ | $87.0 \pm 0.2$ |
| Sky | $3.8 \pm 0.4$ | $3.7 \pm 0.7$ | $4.8 \pm 0.4$ | $29.8 \pm 9.8$ | $3.3 \pm 0.6$ | $8.3 \pm 1.3$ | $24.0 \pm 7.5$ | $55.1 \pm 23.5$ | $84.6 \pm 0.2$ | $85.5 \pm 0.2$ |
| Stripe | $15.4 \pm 1.1$ | $46.9 \pm 4.9$ | $63.0 \pm 5.6$ | $82.3 \pm 0.8$ | $63.8 \pm 3.7$ | $82.8 \pm 0.5$ | $83.9 \pm 0.3$ | $85.1 \pm 0.2$ | $84.5 \pm 0.4$ | $87.1 \pm 0.1$ |
| Zigzag | $65.0 \pm 0.2$ | $71.3 \pm 0.2$ | $73.8 \pm 0.1$ | $76.1 \pm 0.3$ | $72.3 \pm 0.2$ | $79.7 \pm 0.5$ | $81.0 \pm 0.5$ | $85.8 \pm 0.2$ | $85.7 \pm 0.2$ | $87.5 \pm 0.2$ |
| Avg. | $29.5 \pm 0.5$ | $46.2 \pm 1.1$ | $51.8 \pm 1.1$ | $67.9 \pm 0.7$ | $50.5 \pm 0.6$ | $63.7 \pm 2.2$ | $70.3 \pm 3.7$ | $80.0 \pm 4.4$ | $74.4 \pm 0.8$ | $86.1 \pm 0.1$ |
| Avg.\bgs | $40.9 \pm 0.4$ | $62.8 \pm 1.1$ | $69.3 \pm 1.4$ | $80.1 \pm 0.5$ | $68.6 \pm 0.5$ | $81.2 \pm 3.1$ | $84.5 \pm 0.1$ | $85.6 \pm 0.1$ | $85.2 \pm 0.1$ | $87.3 \pm 0.0$ |

Table 18: EMNIST-DA ECE (%) results. Shown are the mean and 1 standard deviation.

| | Src-only | AdaBN | Marg. Gauss. | Full Gauss. | PL | BNM-IM | SHOT-IM | SHOT | FR | BUFR |
|---|---|---|---|---|---|---|---|---|---|---|
| Bricks | $54.6 \pm 5.0$ | $64.4 \pm 1.1$ | $62.0 \pm 0.9$ | $52.4 \pm 2.0$ | $93.2 \pm 1.2$ | $84.9 \pm 1.4$ | $79.4 \pm 4.8$ | $22.5 \pm 0.3$ | $44.1 \pm 4.2$ | $6.2 \pm 0.4$ |
| Crystals | $27.0 \pm 3.0$ | $29.0 \pm 1.2$ | $24.0 \pm 0.3$ | $22.0 \pm 0.6$ | $52.7 \pm 1.7$ | $38.1 \pm 2.5$ | $28.5 \pm 3.8$ | $19.3 \pm 0.1$ | $10.5 \pm 0.3$ | $6.9 \pm 0.3$ |
| Dotted Line | $11.4 \pm 0.6$ | $9.2 \pm 0.5$ | $7.9 \pm 0.4$ | $7.5 \pm 0.4$ | $19.6 \pm 0.7$ | $13.0 \pm 0.4$ | $12.9 \pm 0.1$ | $12.9 \pm 0.1$ | $3.8 \pm 0.2$ | $3.4 \pm 0.2$ |
| Fog | $19.2 \pm 6.5$ | $13.8 \pm 1.5$ | $8.8 \pm 0.8$ | $4.8 \pm 0.4$ | $24.1 \pm 4.0$ | $13.3 \pm 0.4$ | $13.8 \pm 0.1$ | $13.5 \pm 0.1$ | $3.9 \pm 0.2$ | $3.3 \pm 0.1$ |
| Gaussian Blur | $15.0 \pm 3.9$ | $15.3 \pm 0.9$ | $8.7 \pm 0.5$ | $6.8 \pm 0.3$ | $21.2 \pm 0.8$ | $15.8 \pm 0.4$ | $16.4 \pm 0.2$ | $16.0 \pm 0.3$ | $6.4 \pm 0.7$ | $4.9 \pm 0.1$ |
| Grass | $21.6 \pm 5.4$ | $61.3 \pm 0.8$ | $61.0 \pm 1.0$ | $27.2 \pm 2.7$ | $93.6 \pm 1.5$ | $84.6 \pm 6.2$ | $57.5 \pm 40.9$ | $37.5 \pm 36.1$ | $8.7 \pm 0.6$ | $5.7 \pm 0.2$ |
| Impulse Noise | $32.0 \pm 1.8$ | $9.9 \pm 0.6$ | $7.1 \pm 0.3$ | $6.6 \pm 0.1$ | $20.1 \pm 0.6$ | $15.3 \pm 0.3$ | $15.6 \pm 0.2$ | $15.8 \pm 0.2$ | $5.3 \pm 0.1$ | $4.7 \pm 0.1$ |
| Inverse | $65.1 \pm 5.8$ | $60.8 \pm 2.2$ | $54.9 \pm 5.7$ | $18.1 \pm 3.0$ | $89.3 \pm 2.1$ | $39.0 \pm 23.3$ | $16.9 \pm 0.5$ | $14.7 \pm 0.1$ | $5.6 \pm 0.5$ | $3.3 \pm 0.1$ |
| Oranges | $23.8 \pm 2.7$ | $25.3 \pm 2.0$ | $22.5 \pm 2.3$ | $10.1 \pm 0.6$ | $57.6 \pm 2.7$ | $19.6 \pm 0.6$ | $19.6 \pm 0.4$ | $17.4 \pm 0.2$ | $6.9 \pm 0.5$ | $5.5 \pm 0.3$ |
| Shot Noise | $4.8 \pm 0.5$ | $4.9 \pm 0.3$ | $4.5 \pm 0.3$ | $4.9 \pm 0.2$ | $16.4 \pm 0.1$ | $13.0 \pm 0.1$ | $13.7 \pm 0.1$ | $14.8 \pm 0.1$ | $4.6 \pm 0.3$ | $4.2 \pm 0.2$ |
| Sky | $42.6 \pm 3.5$ | $52.4 \pm 4.9$ | $51.6 \pm 6.4$ | $45.8 \pm 8.4$ | $97.2 \pm 0.4$ | $90.2 \pm 1.1$ | $76.0 \pm 7.5$ | $42.7 \pm 23.0$ | $58.0 \pm 6.8$ | $5.6 \pm 0.3$ |
| Stripe | $63.8 \pm 3.0$ | $31.6 \pm 4.4$ | $20.2 \pm 4.8$ | $6.8 \pm 0.4$ | $36.2 \pm 3.8$ | $16.8 \pm 0.5$ | $16.1 \pm 0.3$ | $15.0 \pm 0.3$ | $5.4 \pm 0.2$ | $4.1 \pm 0.1$ |
| Zigzag | $19.9 \pm 0.3$ | $16.7 \pm 0.2$ | $14.6 \pm 0.1$ | $12.7 \pm 0.3$ | $27.6 \pm 0.2$ | $19.7 \pm 0.5$ | $19.0 \pm 0.5$ | $14.4 \pm 0.1$ | $4.9 \pm 0.1$ | $3.8 \pm 0.2$ |
| Avg. | $30.8 \pm 1.6$ | $30.3 \pm 1.1$ | $26.7 \pm 1.1$ | $17.4 \pm 0.7$ | $49.9 \pm 0.6$ | $35.6 \pm 2.2$ | $29.6 \pm 3.7$ | $19.7 \pm 4.4$ | $12.9 \pm 0.9$ | $4.7 \pm 0.2$ |
| Avg.\bgs | $28.9 \pm 1.3$ | $20.3 \pm 0.8$ | $15.8 \pm 1.1$ | $8.5 \pm 0.4$ | $31.8 \pm 0.5$ | $18.2 \pm 3.1$ | $15.6 \pm 0.1$ | $14.6 \pm 0.1$ | $5.0 \pm 0.2$ | $4.0 \pm 0.1$ |

### K.6 CIFAR-10-C FULL RESULTS

Tables 19 and 20 show the accuracy and ECE results for each individual corruption of CIFAR-10-C. It is worth noting that BUFR achieves the biggest wins on the more severe shifts, i.e. those on which AdaBN (Li et al., 2017) performs poorly.

Table 19: CIFAR-10-C accuracy (%) results. Shown are the mean and 1 standard deviation.

| | Src-only | AdaBN | PL | SHOT-IM | TENT | FR | BUFR |
|---|---|---|---|---|---|---|---|
| Brightness | $91.4 \pm 0.4$ | $91.5 \pm 0.3$ | $91.9 \pm 0.2$ | $92.7 \pm 0.3$ | $93.2 \pm 0.3$ | $93 \pm 0.4$ | $93.3 \pm 0.3$ |
| Contrast | $32.3 \pm 1.3$ | $87.1 \pm 0.3$ | $86.6 \pm 3.2$ | $90.8 \pm 0.9$ | $91.3 \pm 1.7$ | $90.9 \pm 0.9$ | $92.9 \pm 0.7$ |
| Defocus blr | $53.1 \pm 6.4$ | $88.8 \pm 0.4$ | $89.3 \pm 0.5$ | $90.5 \pm 0.4$ | $90.9 \pm 0.5$ | $90.9 \pm 0.3$ | $91.5 \pm 0.5$ |
| Elastic | $77.6 \pm 0.6$ | $78.2 \pm 0.4$ | $79.2 \pm 0.8$ | $81.4 \pm 0.5$ | $82.7 \pm 0.5$ | $82.7 \pm 0.4$ | $84.2 \pm 0.3$ |
| Fog | $72.9 \pm 2.6$ | $85.9 \pm 0.9$ | $86.5 \pm 0.8$ | $88.7 \pm 0.4$ | $89.5 \pm 0.4$ | $89.5 \pm 0.5$ | $91.5 \pm 0.5$ |
| Frost | $64.4 \pm 2.4$ | $80.7 \pm 0.7$ | $82.4 \pm 1.2$ | $85.4 \pm 0.6$ | $86.8 \pm 0.7$ | $87 \pm 0.6$ | $89.1 \pm 0.9$ |
| Gauss. blr | $35.9 \pm 8$ | $88.2 \pm 0.6$ | $89 \pm 0.7$ | $90.5 \pm 0.5$ | $91 \pm 0.6$ | $91.2 \pm 0.6$ | $92.3 \pm 0.4$ |
| Gauss. nse | $27.7 \pm 5.1$ | $69.2 \pm 1$ | $74.6 \pm 0.6$ | $79.2 \pm 0.9$ | $81.3 \pm 0.5$ | $81.9 \pm 0.1$ | $85.9 \pm 0.4$ |
| Glass blr | $51.3 \pm 1.8$ | $66.7 \pm 0.4$ | $69 \pm 0.3$ | $73.7 \pm 1$ | $74.7 \pm 0.8$ | $76.8 \pm 0.8$ | $80.3 \pm 0.5$ |
| Impulse nse | $25.9 \pm 3.8$ | $62.1 \pm 1$ | $67.2 \pm 0.5$ | $73.2 \pm 0.8$ | $75.3 \pm 0.8$ | $76.6 \pm 0.4$ | $89.3 \pm 1.4$ |
| Jpeg compr. | $74.9 \pm 1$ | $74.1 \pm 1$ | $77.3 \pm 0.6$ | $81 \pm 0.3$ | $82.9 \pm 0.5$ | $83.4 \pm 0.5$ | $85.8 \pm 0.6$ |
| Motion blr | $66.1 \pm 1.7$ | $87.2 \pm 0.2$ | $87.8 \pm 0.2$ | $89.1 \pm 0.2$ | $90 \pm 0.3$ | $89.8 \pm 0.2$ | $90.8 \pm 0.2$ |
| Pixelate | $48.2 \pm 2.2$ | $80.4 \pm 0.5$ | $82 \pm 0.4$ | $85.5 \pm 0.7$ | $87.6 \pm 0.9$ | $87.5 \pm 0.8$ | $89.9 \pm 0.6$ |
| Saturate | $89.9 \pm 0.4$ | $92 \pm 0.1$ | $92.5 \pm 0.3$ | $93.1 \pm 0.1$ | $93.3 \pm 0.1$ | $93.4 \pm 0.4$ | $93.5 \pm 0.3$ |
| Shot nse | $34.4 \pm 4.9$ | $71.2 \pm 1.2$ | $77.1 \pm 0.9$ | $81.6 \pm 0.7$ | $83.5 \pm 0.6$ | $85.6 \pm 1.9$ | $87 \pm 0.2$ |
| Snow | $76.6 \pm 1.1$ | $82.4 \pm 0.6$ | $83.8 \pm 1.1$ | $86.4 \pm 0.6$ | $87.8 \pm 0.7$ | $88.4 \pm 1.7$ | $89.7 \pm 0.5$ |
| Spatter | $75 \pm 0.8$ | $83.3 \pm 0.5$ | $85.5 \pm 0.3$ | $88 \pm 0.2$ | $88.5 \pm 0.3$ | $91 \pm 2.4$ | $92.6 \pm 0.5$ |
| Speckle nse | $40.7 \pm 3.7$ | $70.4 \pm 0.8$ | $76.1 \pm 1.2$ | $81.3 \pm 1$ | $83.2 \pm 0.9$ | $85.8 \pm 1.7$ | $87.4 \pm 0.4$ |
| Zoom blr | $60.5 \pm 5.1$ | $88.1 \pm 0.3$ | $89 \pm 0.4$ | $90.6 \pm 0.2$ | $91.3 \pm 0.3$ | $91.2 \pm 0.7$ | $91.6 \pm 0.2$ |
| Avg. | $57.8 \pm 0.7$ | $80.4 \pm 0.1$ | $82.5 \pm 0.3$ | $85.4 \pm 0.2$ | $86.6 \pm 0.3$ | $87.2 \pm 0.7$ | $89.4 \pm 0.2$ |

Table 20: CIFAR-10-C ECE (%) results. Shown are the mean and 1 standard deviation.

| | Src-only | AdaBN | PL | SHOT-IM | TENT | FR | BUFR |
|---|---|---|---|---|---|---|---|
| Brightness | $4.7 \pm 0.2$ | $4 \pm 0.1$ | $8.1 \pm 0.2$ | $7.2 \pm 0.4$ | $6.4 \pm 0.3$ | $5.9 \pm 0.3$ | $6.2 \pm 0.2$ |
| Contrast | $43.5 \pm 2.8$ | $5.7 \pm 0.4$ | $13.2 \pm 3.1$ | $9.8 \pm 0.9$ | $8.4 \pm 1.6$ | $6.6 \pm 0.4$ | $6.6 \pm 0.7$ |
| Defocus blr | $28.2 \pm 4$ | $6.1 \pm 0.4$ | $10.7 \pm 0.5$ | $9.4 \pm 0.4$ | $8.6 \pm 0.5$ | $7.8 \pm 0.3$ | $7.9 \pm 0.4$ |
| Elastic | $12.4 \pm 0.7$ | $12.6 \pm 0.4$ | $20.8 \pm 0.8$ | $18.6 \pm 0.5$ | $16.5 \pm 0.5$ | $15.1 \pm 0.5$ | $15.2 \pm 0.3$ |
| Fog | $17.4 \pm 2.1$ | $7.5 \pm 0.7$ | $13.5 \pm 0.8$ | $11.3 \pm 0.3$ | $10.1 \pm 0.4$ | $8.9 \pm 0.3$ | $8 \pm 0.5$ |
| Frost | $22.7 \pm 1.7$ | $10.4 \pm 0.6$ | $17.5 \pm 1.2$ | $14.6 \pm 0.6$ | $12.7 \pm 0.6$ | $10.7 \pm 0.8$ | $10.3 \pm 0.9$ |
| Gauss. blr | $40.7 \pm 6.2$ | $6.1 \pm 0.4$ | $11 \pm 0.7$ | $9.5 \pm 0.4$ | $8.5 \pm 0.5$ | $7.5 \pm 0.4$ | $7.3 \pm 0.4$ |
| Gauss. nse | $57.6 \pm 6.7$ | $18.5 \pm 0.7$ | $25.3 \pm 0.6$ | $20.9 \pm 0.9$ | $18 \pm 0.4$ | $15.9 \pm 0.3$ | $13.3 \pm 0.4$ |
| Glass blr | $31.2 \pm 1.3$ | $20.8 \pm 0.4$ | $30.9 \pm 0.3$ | $26.3 \pm 1$ | $24.2 \pm 0.8$ | $20.9 \pm 0.7$ | $18.9 \pm 0.5$ |
| Impulse nse | $51.2 \pm 4$ | $23.3 \pm 0.8$ | $32.7 \pm 0.5$ | $26.8 \pm 0.9$ | $23.7 \pm 0.8$ | $20.6 \pm 0.5$ | $10.2 \pm 1.3$ |
| Jpeg compr. | $14.6 \pm 0.8$ | $15.5 \pm 0.7$ | $22.6 \pm 0.6$ | $18.9 \pm 0.4$ | $16.4 \pm 0.5$ | $14.5 \pm 0.4$ | $13.6 \pm 0.7$ |
| Motion blr | $21.1 \pm 1.3$ | $6.8 \pm 0.3$ | $12.1 \pm 0.2$ | $10.9 \pm 0.2$ | $9.5 \pm 0.3$ | $8.7 \pm 0.3$ | $8.6 \pm 0.2$ |
| Pixelate | $36.9 \pm 2.5$ | $11.1 \pm 0.4$ | $17.9 \pm 0.4$ | $14.5 \pm 0.7$ | $11.9 \pm 0.9$ | $10.7 \pm 0.7$ | $9.5 \pm 0.6$ |
| Saturate | $5.5 \pm 0.3$ | $4.2 \pm 0.1$ | $7.4 \pm 0.3$ | $6.9 \pm 0.1$ | $6.4 \pm 0.1$ | $5.9 \pm 0.2$ | $6 \pm 0.3$ |
| Shot nse | $50.2 \pm 5.9$ | $17 \pm 0.9$ | $22.8 \pm 0.9$ | $18.4 \pm 0.7$ | $15.9 \pm 0.6$ | $14 \pm 0.3$ | $12.3 \pm 0.2$ |
| Snow | $14.3 \pm 0.5$ | $9.8 \pm 0.4$ | $16.1 \pm 1.1$ | $13.6 \pm 0.6$ | $11.6 \pm 0.7$ | $10.5 \pm 0.7$ | $9.7 \pm 0.4$ |
| Spatter | $16.9 \pm 0.8$ | $9.3 \pm 0.3$ | $14.5 \pm 0.3$ | $12 \pm 0.2$ | $11 \pm 0.2$ | $9.3 \pm 0.2$ | $7 \pm 0.6$ |
| Speckle nse | $43.2 \pm 4.5$ | $17.9 \pm 0.5$ | $23.8 \pm 1.1$ | $18.7 \pm 1$ | $16.1 \pm 0.9$ | $13.9 \pm 0.7$ | $11.9 \pm 0.4$ |
| Zoom blr | $24.4 \pm 3.5$ | $6.2 \pm 0.1$ | $11 \pm 0.4$ | $9.4 \pm 0.2$ | $8.3 \pm 0.3$ | $7.6 \pm 0.5$ | $7.9 \pm 0.2$ |
| Avg. | $28.2 \pm 0.4$ | $11.2 \pm 0.1$ | $17.5 \pm 0.3$ | $14.6 \pm 0.2$ | $12.8 \pm 0.3$ | $11.3 \pm 0.3$ | $10 \pm 0.2$ |

### K.7 CIFAR-100-C FULL RESULTS

Tables 21 and 22 show the accuracy and ECE results for each individual corruption of CIFAR-100-C. It is worth noting that BUFR achieves the biggest wins on the more severe shifts, i.e. those on which AdaBN (Li et al., 2017) performs poorly.

Table 21: CIFAR-100-C accuracy (%) results. Shown are the mean and 1 standard deviation.

|  | Src-only | AdaBN | PL | SHOT-IM | TENT | FR | BUFR |
|---|---|---|---|---|---|---|---|
| Brightness | $63.2 \pm 1.1$ | $66.1 \pm 0.5$ | $69.6 \pm 0.7$ | $72.6 \pm 0.6$ | $72.2 \pm 0.5$ | $71.8 \pm 0.5$ | $73.6 \pm 0.2$ |
| Contrast | $13.9 \pm 0.6$ | $61.4 \pm 0.4$ | $59.2 \pm 3.5$ | $70.1 \pm 0.4$ | $64 \pm 3.1$ | $68 \pm 0.5$ | $72.2 \pm 0.5$ |
| Defocus blr | $35.9 \pm 0.7$ | $65.6 \pm 0.1$ | $69.3 \pm 0.1$ | $71.8 \pm 0.3$ | $71 \pm 0.5$ | $71.2 \pm 0.1$ | $72.2 \pm 0.2$ |
| Elastic | $58.5 \pm 0.7$ | $60.4 \pm 0.2$ | $63.9 \pm 0.4$ | $66.9 \pm 0.2$ | $65.5 \pm 0.2$ | $65.9 \pm 0.4$ | $67.1 \pm 0.5$ |
| Fog | $36.9 \pm 0.5$ | $55.4 \pm 0.6$ | $60.4 \pm 0.6$ | $66.5 \pm 0.6$ | $67.1 \pm 0.6$ | $64.9 \pm 0.4$ | $70.1 \pm 0.5$ |
| Frost | $41.1 \pm 0.9$ | $55.3 \pm 0.6$ | $60.1 \pm 0.8$ | $65.2 \pm 0.4$ | $65.3 \pm 0.9$ | $63 \pm 0.5$ | $67.5 \pm 0.7$ |
| Gauss. blr | $28.2 \pm 1$ | $64.3 \pm 0.3$ | $68.9 \pm 0.1$ | $71.7 \pm 0.2$ | $71 \pm 0.3$ | $70.9 \pm 0.3$ | $72.9 \pm 0.6$ |
| Gauss. nse | $11.9 \pm 1.2$ | $43.8 \pm 0.6$ | $53.1 \pm 0.7$ | $60.3 \pm 0.4$ | $59.5 \pm 0.6$ | $57.7 \pm 0.4$ | $63 \pm 0.3$ |
| Glass blr | $45.1 \pm 0.9$ | $53.3 \pm 0.6$ | $57.3 \pm 0.7$ | $62.4 \pm 0.3$ | $61.4 \pm 0.5$ | $60.5 \pm 0.3$ | $63.2 \pm 0.4$ |
| Impulse nse | $7.2 \pm 0.8$ | $40.8 \pm 0.4$ | $50.6 \pm 0.5$ | $58.4 \pm 0.6$ | $56.3 \pm 0.7$ | $55.2 \pm 0.9$ | $66.9 \pm 0.6$ |
| Jpeg compr. | $48.6 \pm 0.9$ | $49.8 \pm 0.7$ | $55.8 \pm 0.3$ | $61.2 \pm 0.5$ | $60.8 \pm 0.1$ | $59.3 \pm 0.5$ | $62.6 \pm 0.4$ |
| Motion blr | $45.1 \pm 0.5$ | $63.4 \pm 0.2$ | $66.3 \pm 0.6$ | $69.7 \pm 0.2$ | $69 \pm 0.5$ | $68.6 \pm 0.4$ | $70.8 \pm 0.2$ |
| Pixelate | $22.3 \pm 0.4$ | $59.4 \pm 0.6$ | $64.9 \pm 0.6$ | $69.7 \pm 0.4$ | $69.8 \pm 0.4$ | $68.1 \pm 0.3$ | $71.4 \pm 0.5$ |
| Saturate | $55.8 \pm 0.4$ | $65.7 \pm 0.4$ | $70.2 \pm 0.8$ | $72.6 \pm 0.2$ | $71.4 \pm 0.7$ | $72.2 \pm 0.5$ | $72.4 \pm 0.6$ |
| Shot nse | $14.1 \pm 1.2$ | $44.6 \pm 0.9$ | $56.1 \pm 0.8$ | $61.9 \pm 0.6$ | $60.3 \pm 0.4$ | $59.8 \pm 0.3$ | $62.1 \pm 2.8$ |
| Snow | $49.4 \pm 0.8$ | $53.5 \pm 0.4$ | $59.8 \pm 0.9$ | $65 \pm 0.6$ | $65.6 \pm 0.4$ | $63.8 \pm 0.6$ | $65.9 \pm 2.2$ |
| Spatter | $54.8 \pm 1.1$ | $64.9 \pm 0.6$ | $72.1 \pm 0.3$ | $73.8 \pm 0.4$ | $72.9 \pm 0.5$ | $73.8 \pm 0.5$ | $74.3 \pm 0.2$ |
| Speckle nse | $15.6 \pm 1.3$ | $42.3 \pm 1$ | $54.2 \pm 1.5$ | $62.1 \pm 0.6$ | $59.8 \pm 0.3$ | $59.6 \pm 0.8$ | $62.1 \pm 2.7$ |
| Zoom blr | $45.1 \pm 0.7$ | $65.9 \pm 0.3$ | $69.1 \pm 0.5$ | $71.9 \pm 0.3$ | $71.1 \pm 0.8$ | $71 \pm 0.6$ | $71.2 \pm 0.4$ |
| Avg. | $36.4 \pm 0.5$ | $56.6 \pm 0.3$ | $62.1 \pm 0.2$ | $67 \pm 0.2$ | $66 \pm 0.4$ | $65.5 \pm 0.2$ | $68.5 \pm 0.2$ |

Table 22: CIFAR-100-C ECE (%) results. Shown are the mean and 1 standard deviation.

|  | Src-only | AdaBN | PL | SHOT-IM | TENT | FR | BUFR |
|---|---|---|---|---|---|---|---|
| Brightness | $6.3 \pm 0.3$ | $9.4 \pm 0.3$ | $30.2 \pm 0.7$ | $27.4 \pm 0.4$ | $20.7 \pm 0.4$ | $12.4 \pm 0.3$ | $12 \pm 0.6$ |
| Contrast | $37.8 \pm 2.2$ | $11.4 \pm 0.3$ | $40.5 \pm 3.4$ | $29.6 \pm 0.8$ | $29.5 \pm 3.5$ | $14 \pm 0.2$ | $12.8 \pm 0.5$ |
| Defocus blr | $16 \pm 0.8$ | $9.7 \pm 0.3$ | $30.6 \pm 0.2$ | $28.2 \pm 0.4$ | $21.6 \pm 0.3$ | $13.4 \pm 0.3$ | $12.7 \pm 0.2$ |
| Elastic | $8 \pm 0.1$ | $10.8 \pm 0.2$ | $35.9 \pm 0.4$ | $33 \pm 0.3$ | $25.8 \pm 0.1$ | $15.2 \pm 0.2$ | $15.3 \pm 0.3$ |
| Fog | $21 \pm 0.6$ | $12.2 \pm 0.3$ | $39.5 \pm 0.6$ | $33.3 \pm 0.7$ | $24.8 \pm 0.5$ | $15.9 \pm 0.3$ | $14 \pm 0.6$ |
| Frost | $14.1 \pm 1.1$ | $13.3 \pm 0.4$ | $39.7 \pm 0.8$ | $34.8 \pm 0.4$ | $26.1 \pm 0.7$ | $16.3 \pm 0.3$ | $15.3 \pm 0.2$ |
| Gauss. blr | $20.5 \pm 1.4$ | $10 \pm 0.4$ | $31 \pm 0.1$ | $28.4 \pm 0.2$ | $21.7 \pm 0.2$ | $13.5 \pm 0.2$ | $12.5 \pm 0.3$ |
| Gauss. nse | $39.3 \pm 5.5$ | $16.7 \pm 0.2$ | $46.8 \pm 0.6$ | $39.8 \pm 0.5$ | $30.8 \pm 0.7$ | $19.4 \pm 0.5$ | $17.5 \pm 0.5$ |
| Glass blr | $15.7 \pm 1.1$ | $13.4 \pm 0.1$ | $42.5 \pm 0.7$ | $37.6 \pm 0.3$ | $29.1 \pm 0.4$ | $17.9 \pm 0.4$ | $17.6 \pm 0.6$ |
| Impulse nse | $35.1 \pm 2.6$ | $17.4 \pm 0.2$ | $49.3 \pm 0.6$ | $41.5 \pm 0.7$ | $33.7 \pm 0.8$ | $20.5 \pm 0.3$ | $15.2 \pm 0.2$ |
| Jpeg compr. | $8.6 \pm 0.2$ | $15 \pm 0.4$ | $44.1 \pm 0.4$ | $38.8 \pm 0.5$ | $29.6 \pm 0.2$ | $19.1 \pm 0.2$ | $18.2 \pm 0.5$ |
| Motion blr | $12.2 \pm 0.2$ | $10.4 \pm 0.3$ | $33.6 \pm 0.6$ | $30.3 \pm 0.2$ | $23.2 \pm 0.4$ | $14.3 \pm 0.3$ | $13.6 \pm 0.3$ |
| Pixelate | $27.5 \pm 1$ | $11.6 \pm 0.4$ | $35 \pm 0.6$ | $30.3 \pm 0.4$ | $22.5 \pm 0.3$ | $14.2 \pm 0.4$ | $13.6 \pm 0.4$ |
| Saturate | $8.8 \pm 0.2$ | $9.5 \pm 0.3$ | $29.6 \pm 0.8$ | $27.4 \pm 0.3$ | $21.2 \pm 0.6$ | $12.7 \pm 0.2$ | $12.3 \pm 0.7$ |
| Shot nse | $37.2 \pm 5.9$ | $16 \pm 0.2$ | $43.7 \pm 0.8$ | $38.1 \pm 0.5$ | $30.2 \pm 0.8$ | $18.6 \pm 0.4$ | $17 \pm 0.6$ |
| Snow | $8.5 \pm 0.3$ | $14.4 \pm 0.2$ | $40.1 \pm 0.9$ | $34.9 \pm 0.7$ | $25.7 \pm 0.5$ | $17 \pm 0.5$ | $14.9 \pm 0.1$ |
| Spatter | $6.7 \pm 0.3$ | $9.3 \pm 0.1$ | $27.8 \pm 0.3$ | $26.2 \pm 0.4$ | $20 \pm 0.6$ | $12 \pm 0.3$ | $11 \pm 0.4$ |
| Speckle nse | $34.5 \pm 5.4$ | $17.2 \pm 0.3$ | $45.7 \pm 1.6$ | $37.9 \pm 0.6$ | $30.6 \pm 0.2$ | $18.7 \pm 0.6$ | $16.8 \pm 0.8$ |
| Zoom blr | $10.5 \pm 0.3$ | $9.1 \pm 0.2$ | $30.8 \pm 0.5$ | $28.2 \pm 0.4$ | $21.5 \pm 0.7$ | $13.1 \pm 0.5$ | $13.2 \pm 0.7$ |
| Avg. | $19.4 \pm 0.9$ | $12.5 \pm 0.1$ | $37.7 \pm 0.2$ | $32.9 \pm 0.2$ | $25.7 \pm 0.4$ | $15.7 \pm 0.1$ | $14.5 \pm 0.3$ |

### K.8   CIFAR-10-C FULL ONLINE RESULTS

Tables 23 and 24 show the accuracy and ECE results for each individual corruption of CIFAR-10-C when adapting in an *online* fashion (see Appendix K.2). It is worth noting that FR achieves the biggest wins on the more severe shifts, i.e. those on which AdaBN (Li et al., 2017) performs poorly.

Table 23: CIFAR-10-C *online* accuracy (%) results. Shown are the mean and 1 standard deviation.

|  | Src-only | AdaBN | SHOT-IM | TENT | FR |
|---|---|---|---|---|---|
| Brightness | 91.4 ± 0.4 | 91.6 ± 0.2 | 92.2 ± 0.4 | 91.8 ± 0.3 | 92.8 ± 0.3 |
| Contrast | 32.3 ± 1.3 | 87.1 ± 0.4 | 87.8 ± 0.5 | 87.8 ± 0.6 | 89.8 ± 0.6 |
| Defocus blr | 53.1 ± 6.4 | 88.7 ± 0.5 | 89.7 ± 0.5 | 89.1 ± 0.5 | 90.6 ± 0.5 |
| Elastic | 77.6 ± 0.6 | 78 ± 0.3 | 80.3 ± 0.6 | 79.2 ± 0.5 | 82 ± 0.4 |
| Fog | 72.9 ± 2.6 | 85.9 ± 1.1 | 87.2 ± 0.5 | 86.5 ± 0.8 | 89 ± 0.8 |
| Frost | 64.4 ± 2.4 | 80.7 ± 0.8 | 83 ± 0.8 | 81.8 ± 0.8 | 85.9 ± 0.7 |
| Gauss. blr | 35.9 ± 8 | 88.3 ± 0.7 | 89.5 ± 0.6 | 88.8 ± 0.5 | 90.8 ± 0.6 |
| Gauss. nse | 27.7 ± 5.1 | 68.8 ± 0.9 | 75.4 ± 0.8 | 72.3 ± 0.7 | 80.6 ± 0.6 |
| Glass blr | 51.3 ± 1.8 | 66.7 ± 0.5 | 70.6 ± 1 | 68.3 ± 0.6 | 74.7 ± 0.9 |
| Impulse nse | 25.9 ± 3.8 | 62 ± 1.2 | 68.8 ± 0.8 | 65.5 ± 0.7 | 74.5 ± 0.4 |
| Jpeg compr. | 74.9 ± 1 | 73.9 ± 1.2 | 78.4 ± 0.9 | 76.2 ± 0.9 | 82.2 ± 0.5 |
| Motion blr | 66.1 ± 1.7 | 87 ± 0.1 | 88.2 ± 0.3 | 87.6 ± 0.3 | 89.5 ± 0.2 |
| Pixelate | 48.2 ± 2.2 | 80.5 ± 0.4 | 83.2 ± 0.7 | 81.7 ± 0.5 | 86.7 ± 0.7 |
| Saturate | 89.9 ± 0.4 | 91.9 ± 0.1 | 92.4 ± 0.1 | 92.3 ± 0.2 | 92.8 ± 0.2 |
| Shot nse | 34.4 ± 4.9 | 70.9 ± 1.2 | 77.7 ± 1.6 | 74.6 ± 1.3 | 82.2 ± 0.6 |
| Snow | 76.6 ± 1.1 | 82.6 ± 0.7 | 84.5 ± 0.9 | 83.4 ± 0.9 | 86.8 ± 0.6 |
| Spatter | 75 ± 0.8 | 83.2 ± 0.5 | 86 ± 0.2 | 84.6 ± 0.2 | 88.6 ± 0.2 |
| Speckle nse | 40.7 ± 3.7 | 70.2 ± 0.7 | 77.2 ± 0.6 | 74.2 ± 0.6 | 82.4 ± 0.2 |
| Zoom blr | 60.5 ± 5.1 | 88 ± 0.4 | 89.4 ± 0.2 | 88.6 ± 0.3 | 90.7 ± 0.2 |
| Avg. | 57.8 ± 0.7 | 80.3 ± 0 | 83.2 ± 0.2 | 81.8 ± 0.2 | 85.9 ± 0.3 |

Table 24: CIFAR-10-C *online* ECE (%) results. Shown are the mean and 1 standard deviation.

|  | Src-only | AdaBN | SHOT-IM | TENT | FR |
|---|---|---|---|---|---|
| Brightness | 4.7 ± 0.2 | 5.4 ± 0.2 | 5.1 ± 0.3 | 5.1 ± 0.2 | 4.9 ± 0.3 |
| Contrast | 43.5 ± 2.8 | 6.8 ± 0.4 | 8.7 ± 0.5 | 7.6 ± 0.5 | 6.1 ± 0.4 |
| Defocus blr | 28.2 ± 4 | 7.1 ± 0.4 | 6.7 ± 0.3 | 7 ± 0.3 | 6.4 ± 0.3 |
| Elastic | 12.4 ± 0.7 | 13.5 ± 0.3 | 12.7 ± 0.4 | 12.9 ± 0.5 | 12.3 ± 0.4 |
| Fog | 17.4 ± 2.1 | 8.4 ± 0.6 | 8.3 ± 0.3 | 8.3 ± 0.4 | 7.3 ± 0.5 |
| Frost | 22.7 ± 1.7 | 11.2 ± 0.7 | 10.9 ± 0.5 | 11 ± 0.5 | 9 ± 0.6 |
| Gauss. blr | 40.7 ± 6.2 | 7.3 ± 0.4 | 6.8 ± 0.4 | 7 ± 0.3 | 6.3 ± 0.4 |
| Gauss. nse | 57.6 ± 6.7 | 19.2 ± 0.7 | 16 ± 0.6 | 17.6 ± 0.4 | 13.2 ± 0.6 |
| Glass blr | 31.2 ± 1.3 | 21.4 ± 0.6 | 19.6 ± 0.7 | 20.7 ± 0.5 | 17.9 ± 0.7 |
| Impulse nse | 51.2 ± 4 | 23.8 ± 0.9 | 20.6 ± 0.8 | 22.2 ± 0.4 | 17.9 ± 0.4 |
| Jpeg compr. | 14.6 ± 0.8 | 16.3 ± 0.9 | 14 ± 0.5 | 15.1 ± 0.6 | 12.2 ± 0.4 |
| Motion blr | 21.1 ± 1.3 | 7.8 ± 0.1 | 7.6 ± 0.3 | 7.7 ± 0.2 | 7 ± 0.2 |
| Pixelate | 36.9 ± 2.5 | 12 ± 0.4 | 10.8 ± 0.6 | 11.5 ± 0.5 | 8.9 ± 0.5 |
| Saturate | 5.5 ± 0.3 | 5.1 ± 0.1 | 5 ± 0.1 | 5.1 ± 0.1 | 4.9 ± 0.2 |
| Shot nse | 50.2 ± 5.9 | 17.9 ± 0.9 | 14.3 ± 1.1 | 16 ± 0.8 | 12 ± 0.5 |
| Snow | 14.3 ± 0.5 | 10.7 ± 0.3 | 9.9 ± 0.6 | 10.4 ± 0.6 | 8.9 ± 0.5 |
| Spatter | 16.9 ± 0.8 | 10.2 ± 0.4 | 9.1 ± 0.2 | 9.6 ± 0.2 | 7.6 ± 0.2 |
| Speckle nse | 43.2 ± 4.5 | 18.8 ± 0.5 | 14.8 ± 0.5 | 16.3 ± 0.5 | 11.9 ± 0.2 |
| Zoom blr | 24.4 ± 3.5 | 7.3 ± 0.3 | 6.8 ± 0.2 | 7.1 ± 0.2 | 6.3 ± 0.1 |
| Avg. | 28.2 ± 0.4 | 12.1 ± 0 | 10.9 ± 0.1 | 11.5 ± 0.1 | 9.5 ± 0.2 |

### K.9 CIFAR-100-C FULL ONLINE RESULTS

Tables 25 and 26 show the accuracy and ECE results for each individual corruption of CIFAR-100-C when adapting in an *online* fashion (see Appendix K.2). It is worth noting that FR achieves the biggest wins on the more severe shifts, i.e. those on which AdaBN (Li et al., 2017) performs poorly.

Table 25: CIFAR-100-C *online* accuracy (%) results. Shown are the mean and 1 standard deviation.

|  | Src-only | AdaBN | SHOT-IM | TENT | FR |
|---|---|---|---|---|---|
| Brightness | $63.2 \pm 1.1$ | $66.1 \pm 0.4$ | $69.3 \pm 0.9$ | $69.9 \pm 0.7$ | $69.4 \pm 0.4$ |
| Contrast | $13.9 \pm 0.6$ | $61.4 \pm 0.5$ | $64.8 \pm 0.5$ | $66.6 \pm 1.1$ | $64.5 \pm 0.3$ |
| Defocus blr | $35.9 \pm 0.7$ | $65.6 \pm 0.1$ | $69 \pm 0.1$ | $69.4 \pm 0.3$ | $68.6 \pm 0.2$ |
| Elastic | $58.5 \pm 0.7$ | $60.4 \pm 0.2$ | $63.3 \pm 0.5$ | $63.7 \pm 0.1$ | $63.4 \pm 0.3$ |
| Fog | $36.9 \pm 0.5$ | $55.4 \pm 0.6$ | $61 \pm 0.5$ | $62.5 \pm 0.7$ | $61.7 \pm 0.5$ |
| Frost | $41.1 \pm 0.9$ | $55.3 \pm 0.6$ | $60.5 \pm 1$ | $61.8 \pm 0.6$ | $60.8 \pm 0.8$ |
| Gauss. blr | $28.2 \pm 1$ | $64.3 \pm 0.3$ | $68.6 \pm 0.2$ | $69 \pm 0.6$ | $68.4 \pm 0.5$ |
| Gauss. nse | $11.9 \pm 1.2$ | $43.8 \pm 0.6$ | $53.5 \pm 0.2$ | $55.1 \pm 0.5$ | $54.7 \pm 0.3$ |
| Glass blr | $45.1 \pm 0.9$ | $53.3 \pm 0.6$ | $57.8 \pm 0.4$ | $58.2 \pm 0.5$ | $57.9 \pm 0.5$ |
| Impulse nse | $7.2 \pm 0.8$ | $40.8 \pm 0.5$ | $50.2 \pm 0.4$ | $50.9 \pm 0.7$ | $51.7 \pm 0.8$ |
| Jpeg compr. | $48.6 \pm 0.9$ | $49.8 \pm 0.7$ | $56 \pm 0.2$ | $57.2 \pm 0.2$ | $56.6 \pm 0.6$ |
| Motion blr | $45.1 \pm 0.5$ | $63.4 \pm 0.2$ | $66.4 \pm 0.4$ | $66.7 \pm 0.6$ | $66 \pm 0.3$ |
| Pixelate | $22.3 \pm 0.4$ | $59.4 \pm 0.6$ | $65.1 \pm 0.7$ | $67.1 \pm 0.4$ | $65.6 \pm 0.6$ |
| Saturate | $55.8 \pm 0.4$ | $65.7 \pm 0.4$ | $69.5 \pm 0.6$ | $69.5 \pm 0.6$ | $69.3 \pm 0.4$ |
| Shot nse | $14.1 \pm 1.2$ | $44.6 \pm 0.9$ | $54.9 \pm 0.1$ | $55.5 \pm 0.4$ | $56.4 \pm 0.3$ |
| Snow | $49.4 \pm 0.8$ | $53.5 \pm 0.4$ | $59.7 \pm 1.1$ | $61.6 \pm 0.8$ | $60.5 \pm 0.5$ |
| Spatter | $54.8 \pm 1.1$ | $64.9 \pm 0.6$ | $71.3 \pm 0.5$ | $70.6 \pm 0.6$ | $71.6 \pm 0.7$ |
| Speckle nse | $15.6 \pm 1.3$ | $42.3 \pm 1$ | $54.2 \pm 0.3$ | $54.9 \pm 0.3$ | $55.8 \pm 0.6$ |
| Zoom blr | $45.1 \pm 0.7$ | $65.9 \pm 0.3$ | $68.9 \pm 0.6$ | $69 \pm 0.4$ | $68.8 \pm 0.3$ |
| Avg. | $36.4 \pm 0.5$ | $56.6 \pm 0.3$ | $62.3 \pm 0.3$ | $63.1 \pm 0.3$ | $62.7 \pm 0.3$ |

Table 26: CIFAR-100-C *online* ECE (%) results. Shown are the mean and 1 standard deviation.

|  | Src-only | AdaBN | SHOT-IM | TENT | FR |
|---|---|---|---|---|---|
| Brightness | $6.3 \pm 0.3$ | $11.4 \pm 0.1$ | $11.6 \pm 0.4$ | $11.9 \pm 0.2$ | $10.9 \pm 0.2$ |
| Contrast | $37.8 \pm 2.2$ | $12.5 \pm 0.2$ | $14.6 \pm 0.3$ | $14.1 \pm 0.6$ | $12.5 \pm 0.2$ |
| Defocus blr | $16 \pm 0.8$ | $11.4 \pm 0.2$ | $11.3 \pm 0.2$ | $11.8 \pm 0.2$ | $11.4 \pm 0.3$ |
| Elastic | $8 \pm 0.1$ | $12.7 \pm 0.2$ | $12.9 \pm 0.2$ | $13.4 \pm 0.3$ | $13 \pm 0.2$ |
| Fog | $21 \pm 0.6$ | $13.8 \pm 0.2$ | $13.9 \pm 0.2$ | $14.4 \pm 0.2$ | $13.6 \pm 0.3$ |
| Frost | $14.1 \pm 1.1$ | $14.5 \pm 0.2$ | $14.5 \pm 0.6$ | $14.8 \pm 0.3$ | $13.9 \pm 0.4$ |
| Gauss. blr | $20.5 \pm 1.4$ | $11.9 \pm 0.3$ | $11.6 \pm 0.4$ | $11.9 \pm 0.2$ | $11.9 \pm 0.4$ |
| Gauss. nse | $39.3 \pm 5.5$ | $17.7 \pm 0.3$ | $16.7 \pm 0.3$ | $17.2 \pm 0.7$ | $16.5 \pm 0.3$ |
| Glass blr | $15.7 \pm 1.1$ | $15 \pm 0.1$ | $15.2 \pm 0.5$ | $16.1 \pm 0.3$ | $15.1 \pm 0.1$ |
| Impulse nse | $35.1 \pm 2.6$ | $18.4 \pm 0.2$ | $18.1 \pm 0.2$ | $19.4 \pm 0.5$ | $17.9 \pm 0.3$ |
| Jpeg compr. | $8.6 \pm 0.2$ | $16.2 \pm 0.3$ | $15.9 \pm 0.1$ | $16.4 \pm 0.4$ | $16.2 \pm 0.2$ |
| Motion blr | $12.2 \pm 0.2$ | $12.2 \pm 0.2$ | $12.2 \pm 0.3$ | $12.9 \pm 0.2$ | $12.5 \pm 0.2$ |
| Pixelate | $27.5 \pm 1$ | $13 \pm 0.3$ | $12.5 \pm 0.3$ | $12.4 \pm 0.1$ | $12.3 \pm 0.2$ |
| Saturate | $8.8 \pm 0.2$ | $11.4 \pm 0.1$ | $11.3 \pm 0.3$ | $11.7 \pm 0.4$ | $11.3 \pm 0.4$ |
| Shot nse | $37.2 \pm 5.9$ | $17.2 \pm 0.3$ | $16.4 \pm 0.2$ | $17.5 \pm 0.7$ | $16.2 \pm 0.3$ |
| Snow | $8.5 \pm 0.3$ | $15.6 \pm 0.2$ | $15 \pm 0.4$ | $14.7 \pm 0.3$ | $14.8 \pm 0.1$ |
| Spatter | $6.7 \pm 0.3$ | $11.4 \pm 0.2$ | $10.7 \pm 0.2$ | $11.3 \pm 0.3$ | $10.7 \pm 0.3$ |
| Speckle nse | $34.5 \pm 5.4$ | $18.2 \pm 0.4$ | $16.5 \pm 0.4$ | $17.5 \pm 0.3$ | $16.1 \pm 0.4$ |
| Zoom blr | $10.5 \pm 0.3$ | $11.2 \pm 0.2$ | $11.2 \pm 0.3$ | $11.6 \pm 0.3$ | $11.4 \pm 0.1$ |
| Avg. | $19.4 \pm 0.9$ | $14 \pm 0.1$ | $13.8 \pm 0.1$ | $14.3 \pm 0.1$ | $13.6 \pm 0.1$ |

## L  NOTATIONS

Table 27 summarizes the notations used in the paper.

Table 27: Notations.

| | Symbol | Description |
|---|---|---|
| Distributions | $p_{\mathbf{z}}$ | Source feature distribution |
| | $q_{\mathbf{z}}$ | Target feature distribution |
| | $p_{z_d}$ | Source $d$-th marginal feature distribution |
| | $q_{z_d}$ | Target $d$-th marginal feature distribution |
| | $\pi_{\mathbf{z}}^s$ | Source approx. marginal feature distributions |
| | $\pi_{\mathbf{z}}^t$ | Target approx. marginal feature distributions |
| | $\pi_{z_d}^s$ | Source $d$-th approx. marginal feature distribution |
| | $\pi_{z_d}^t$ | Target $d$-th approx. marginal feature distribution |
| | $\pi_{\mathbf{a}}^s$ | Source approx. marginal logit distributions |
| | $\pi_{\mathbf{a}}^t$ | Target approx. marginal logit distributions |
| | $\pi_{a_k}^s$ | Source $k$-th approx. marginal logit distribution |
| | $\pi_{a_k}^t$ | Target $k$-th approx. marginal logit distribution |
| Sets | $\mathcal{D}_s$ | Labelled source dataset |
| | $\mathcal{D}_t$ | Unlabelled target dataset |
| | $\mathcal{X}_s$ | Input-set of the source domain |
| | $\mathcal{X}_t$ | Input-set of the target domain |
| | $\mathcal{Y}_s$ | Label-set of the target domain |
| | $\mathcal{Y}_t$ | Label-set of the target domain |
| Network | $f_s$ | Source model, $f_s = h(g_s(\cdot))$ |
| | $f_t$ | Target model, $f_t = h(g_t(\cdot))$ |
| | $g_s$ | Source feature-extractor |
| | $g_t$ | Target feature-extractor |
| | $h$ | Classifier (or regressor) |
| Other | $\mathbf{u}$ | Soft-binning function |
| | $z_d^{min}$ | Minimum value of feature $d$ (on the source data) |
| | $z_d^{max}$ | Maximum value of feature $d$ (on the source data) |
| | $\tau$ | Temperature parameter for soft binning |

