# OpenReview forum: "Source-Free Adaptation to Measurement Shift via Bottom-Up Feature Restoration"
_ICLR.cc/2022/Conference — ICLR 2022 Spotlight_

### Official Review · Reviewer_gJxM · 2021-10-21

**Correctness:** 4
**Technical Novelty And Significance:** 3
**Empirical Novelty And Significance:** 3
**Recommendation:** 8
**Confidence:** 4

**Main Review:**

The strengths of the papers:
a) The paper is very well written, well-motivated, well compared with existing literature and it was a pleasure to read.
b) proposed method makes sense
c) extensive results show that the method obtains good results on multiple datasets, outperforming existing source-free DA methods.
d) the efficiency of proposed bottom-up feature restoration is ablated (and shows large gains for some experiments).

The weaknesses:
a) The proposed method is very simple (both the feature alignment and the bottom up feature restoration). However, I do not consider this a large weakness: results show that these first steps already provide large improvements.
b) the results on the real application (Table 6) are maybe less convincing (especially the impact of bottom-up feature restoration is small) and it would be great if another real application would be found. The authors argue this scenario is not optimal for their method. Maybe they could extend a bit more on potential applciations.

Minor:
the sentence 'Unlike most existing SFDA methods, we make no modifications ' would be nice to see references here to what SFDA methods the authors refer.

I was wondering if the method could work for extending the number of channels of the input data (For exeample source only red-green channels and target red-green-blue).


**Summary Of The Paper:**

Paper proposes a new scenario for source-free domain adaptation where the domains undergo a measurement shift (characterized by a change in the measurement system). Other than the typical source-free DA scenario, in this case, you would not need new features but you would like to update the lower layers of the network. The paper proposes a method that is based on two ideas i) aligning the marginal feature distribution (with KL loss) and ii) performing updating in a bottom-up way (allowing lower layers to first adapt to the target domain). Results show that the method outperforms existing methods on this setting.

**Summary Of The Review:**

The authors propose a relevant new scenario for SFDA and propose a new method to address this setting. Results show that the existing methods struggle for this setting. The proposed method outperforms existing methods. I think the proposed setting is relevant and I would therefore recommend acceptance. The novelty of the proposed solution is not that large though.

---

> ### Author Response · Authors · 2021-11-18
> **Response to Reviewer gJxM**
>
> Dear Reviewer gJxM,
>
> Thank you for your review and helpful feedback.
>
> With regard to your points under weaknesses:
>  - We believe that simplicity can be seen as a strength rather than a weakness, given the large improvements.
>  - We agree that including another real-world dataset in our evaluation would be ideal, particularly one that is “suboptimal” for entropy-minimization methods---i.e. a dataset with more than two classes and results in poor initial performance in the target domain. We did find many such applications in the biomedical domain:
>     - image segmentation across [different MRI scanners](https://arxiv.org/pdf/2001.09313.pdf) [1, Figure 1];
>     - segmentation, attribute detection and disease classification for [skin lesions across different imaging systems](https://arxiv.org/pdf/1902.03368.pdf) [2, Figure 1]; and
>     - nuclei detection across [different staining techniques](https://arxiv.org/pdf/1907.04681.pdf) [3, Figure 1].
>  - However, these biomedical datasets have all been kept private. While this has limited the width of our real-world evaluation, it has also provided further motivation for *source-free* methods.
>
>
> With regard to your minor points:
>  - *References for the sentence: “Unlike most existing SFDA methods, we make no modifications [...]”*
>    - Before this sentence, in the second paragraph of Section 2, we do reference several SFDA methods and detail each of their modifications to the training process.
>    - However, we agree that it would be good to reference these SFDA methods again after this sentence, and will do so in the next update of the paper.
> - *Extending the number of channels of the input data*
>   - Yes, our method would work for extending the number of channels of the input data. Different numbers of channels can be seen as different input representations, i.e. different measurement systems. This could be red-green → red-green-blue (as you suggest), or it could be greyscale → colour.
>   - In fact, the setting of black-and-white EMNIST→ coloured EMNIST is already quite close to this setting (depicted in Figure 7). Here, when training in the source domain, we duplicate the greyscale values across the 3 RGB channels to avoid architectural changes in the target domain.
>   - However, we could instead train directly on the single-channel greyscale data in the source domain. Then, in the target domain, we could randomly initialize new weights in the first convolutional layer connecting to the new channels. We speculate that BUFR would work just as well for this setup.
>
> We hope that our above responses have addressed your concerns. If not, please do let us know so that we can try to address any remaining concerns.
>
> Best wishes,
>
> Authors
>
>
> [1] Li, Hongwei, et al. "Domain Adaptive Medical Image Segmentation via Adversarial Learning of Disease-Specific Spatial Patterns." *arXiv:2001.09313* (2020).
>
> [2] Codella, Noel, et al. "Skin lesion analysis toward melanoma detection 2018: A challenge hosted by the international skin imaging collaboration (isic)." *arXiv:1902.03368* (2019).
>
> [3] Brieu, Nicolas, et al. "Domain adaptation-based augmentation for weakly supervised nuclei detection." *arXiv:1907.04681* (2019).

---

> > ### Comment · Reviewer_gJxM · 2021-11-26
> > **Rebuttal**
> >
> > I thank the reviewers for their feedback and it addressed my few remaining doubts. I remain with my recommendation and hope to be able to see this paper at ICLR 2022.

---

### Official Review · Reviewer_uMHx · 2021-11-03

**Correctness:** 4
**Technical Novelty And Significance:** 3
**Empirical Novelty And Significance:** 3
**Recommendation:** 8
**Confidence:** 4

**Main Review:**

### *Strengths:*
- The paper is well structured and easy to follow. It is well written and the method is easy to understand.
- The experimental analysis is very extensive. Most of the claims and arguments made in the earlier sections are validated with experimental analysis. Also, the proposed method for the most part is novel and quite interesting.
- In my opinion, the major strength of the paper comes from its experimental analysis section. Specifically, extensive comparisons are provided with the most of the methods available in the literature. The paper considers many benchmarking datasets such as cifar-10/100, camelyon17 and also develops a new benchmark with character recognition dataset emnist. However, I would suggest omniglot [1] would be a more challenging dataset than emnist and could be considered in future updates of the work.
- The paper also provides nice analysis of the activation statistics under the *measurement shift*, ablation analysis on which components are useful and discussions on the observations. Overall it provides great insights on the affects of *measurement shift* on activation statistics.


### *Concerns:*
- Though the paper has reasonable novelty and an extensive experimental analysis, there are some aspects of the work which are either not clearly explained or is not validated through experiments.
- In the earlier sections, it is argued that methods such as [2] [3] [4] etc are not comparable to the proposed method as they " are still classification-specific and rely on good initial feature-space class-separation for entropy minimization". However, it is not a strong argument to avoid comparison with these methods which have previously addressed the problem of source-free adaptation. Furthermore, they are very closely related to the proposed method idea of activation distribution matching as compared to their approach of feature-distribution matching. Comparing with these methods would provide key insights into which type of feature-matching is better for the cases considered in the paper. Also, the argument that these methods are classification-specific is not properly evaluated, since the proposed method is also evaluated for only the task of classification.
- The considered *measurement shift* case is widely established as performance under *common corruptions* in the literature, which is what the paper uses for performance evaluation. It is not entirely clear if they are one of the same, which it seems to be. Then why is there a need to define it as *measurement shift*. It would be helpful to get more clarification on the differences between *common corruptions* vs proposed *measurement shift*.
- Also, there are methods like [5], [6] which consider generalization by improving training on source/clean data and without the need for any target/corrupted data. Adding those comparisons (I am assuming in most cases proposed FR/BURF would be better) would also be helpful for benchmarking results. An interesting addition in the experiment would be to consider these improved models [5] [6] as initialization for SFDA and comparing relevant methods.
- I understand comparison with [6] would not have been possible as its a very recent work. However, some of the performances in [6] matches the SFDA method performance, which leads to a more general question about the problem, i.e., Is corrupted data required for model to generalize to those conditions? Should we consider common corruptions as a DA problem at all or is it better modeled as a generalization issue and the focus should be to improve source/clean data training?






[1] https://github.com/brendenlake/omniglot

[2] Li, Rui, et al. "Model adaptation: Unsupervised domain adaptation without source data." Proceedings of the IEEE/CVF Conference on Computer Vision and Pattern Recognition. 2020.

[3] Morerio, Pietro, et al. "Generative pseudo-label refinement for unsupervised domain adaptation." Proceedings of the IEEE/CVF Winter Conference on Applications of Computer Vision. 2020.

[4] Kurmi, Vinod K., Venkatesh K. Subramanian, and Vinay P. Namboodiri. "Domain Impression: A Source Data Free Domain Adaptation Method." Proceedings of the IEEE/CVF Winter Conference on Applications of Computer Vision. 2021.

[5] Hendrycks, Dan, et al. "AugMix: A Simple Data Processing Method to Improve Robustness and Uncertainty." International Conference on Learning Representations. 2019.

[6] Wang, Haotao, et al. "AugMax: Adversarial Composition of Random Augmentations for Robust Training." arXiv e-prints (2021): arXiv-2110.




**Summary Of The Paper:**

The paper tackles a variation of domain adaptation problem where the model is pre-trained on the source data and then deployed in the target domain for adaptation. The major challenge being the absence of source domain data during the adaptation process and hence generally used unsupervised domain adaptation methods are not useful in the case. Furthermore, the paper tackles a sub-set within the ''Source-Free Domain Adaptation'' setting, termed in the paper as *measurement shift*. The proposed method addresses this by aligning the target domain activation statistics with stored statistics of activations measured during development stage (i.e. source domain training). The experimental analysis provided in the paper and supplementary material shows that proposed method is able to outperform all existing methods for the *measurement shift* case.

**Summary Of The Review:**

Overall, I think the paper has some interesting ideas and has a compelling experimental analysis. However, there still remain few concerns/confusions which require further explanation.

---

> ### Author Response · Authors · 2021-11-18
> **Response to Reviewer uMHx (1)**
>
> Dear Reviewer uMHx,
>
> Thank you for your detailed review and helpful feedback. We agree that Omniglot would be even more challenging than EMNIST and thus further expose the weaknesses of existing DA methods for addressing measurement shift. We appreciate this suggestion, and will consider it in future updates of the work. With regard to your concerns:
> - **Why don’t we compare to source-free feature-matching methods like [2], [3] and [4]?**
>   - To clarify, we say that these methods are “still classification-specific and rely on good initial feature-space class-separation” in order to point out their weaknesses rather than to imply that they are not comparable.
>   - We do not compare to these methods as: (i) code has not been released; and (ii) their implementations are complicated, e.g. involving GANs and custom optimization procedures, and would thus require an unreasonable amount of time to replicate.
>   - While we would compare to these methods in an ideal world, we believe our current comparisons to be sufficient as:
>     - We already compare to three *purely* feature-matching approaches: (i) the batch-norm batching method of [5] (“Marg. Gaus”); (ii) an approach that stores and aligns Full Gaussian distributions in feature space (“Full Gauss”); and (iii) the UDA method DANN [6] (on CIFAR-10/100-C). As these do not involve any entropy minimization, they yield a more controlled comparison to our purely feature-matching approach.
>     - Methods [2], [3] and [4] all involve some form of entropy minimization alongside the feature-matching, thus will likely suffer from some of the issues with entropy-minimization that we expose.
> - **“The argument that these methods are classification-specific is not properly evaluated, since the proposed method is also evaluated for only the task of classification”**
>   - We take the reviewer’s point that we do not evaluate on any regression problems, yet emphasize the classification-specificity of other methods.
>   - However, these methods are indeed classification-specific---*it is not possible* to apply them to continuous outputs. In contrast, *it is possible* to apply our methods to continuous outputs.
>   - Exactly how well our method performs on regression problems is a separate matter, although we anticipate that it would do quite well---it only needs to outperform AdaBN as no other SFDA method can be used.
> - **Common corruptions vs. measurement shifts**
>   - We thank the reviewer for raising this point and agree that it would be helpful to clarify the difference between common corruptions and measurement shifts. We will do so in the next update of the paper, using the material below.
>   - *Measurement shifts* are introduced in Section 2, where we:
>     - Define measurement shifts to be domain shifts for which restoring the source features in the target domain is sufficient to restore performance—we do not need to learn *new* features in order to discriminate well between the classes in the target domain.
>     - Justify the name “measurement shifts” by pointing out that these shifts generally arise from a change in measurement system.
>     - Depict some examples of measurement shifts in Figure 1c.
>   - *Common corruptions*:
>     - Are simple transformations of the input, such as Gaussian noise (common in low-lighting conditions), defocus blur (camera is not properly focused or calibrated), brightness (variations in daylight intensity), and impulse noise (colour analogue of salt-and-pepper noise, caused by bit errors) [7].
>     - Can be seen as a change in measurement system.
>     - Can be resolved by restoring the source features in the target domain, e.g. by extracting the same features from blurrier or brighter images.
>     - Are one particular type of measurement shift. E.g., the left column of Figure 1c depicts an impulse-noise shift which is both a common corruption *and* a measurement shift.
>   - *Some measurement shifts are not common corruptions*:
>     - The right column of Figure 1c depicts tissue slides from different hospitals. Here, the shift has arisen from changes in slide-staining procedures, patient populations and image acquisition (e.g. different sensing equipment) [8]. This measurement shift cannot be described in terms of simple input transformations like Gaussian noise or blurring, and thus is not included in common corruptions.
>     - EMNIST-DA shifts like bricks, crystals, grass, sky and oranges *use knowledge of the object type* (i.e. a digit) to change the background and foreground separately (see Figure 7). We do not see these shifts as falling under common corruptions [7], since common corruptions rarely have knowledge of the image content---e.g. blurring all pixels or adding noise randomly.
>   - In summary, measurement shifts are a superset of common corruptions and thus warrant their own definition.

---

> > ### Author Response · Authors · 2021-11-18
> > **Response to Reviewer uMHx (2)**
> >
> > - **Comparing to (and pre-training with) domain generalization (DG) methods like [5] and [6]**
> >   - We agree that combining ideas from domain generalization and domain adaptation is likely to lead to strong results. However:
> >     - In proposing a new method for domain adaptation, we found it best to focus on the question of adaptation. Furthermore, as we discuss in Appendix E and point 5 below, we believe that this focus is warranted since domain adaptation will always be needed---there can always be unseen shifts at deployment time.
> >     - While useful, thoroughly comparing and combining DG and DA approaches is a substantial piece of work in and of itself. We thus leave this interesting avenue of investigation to future work.
> > - **[6] matches the performance of some SFDA methods---do we even need to adapt the source model on the target data with DA methods, or should we instead just focus on DG methods that improve training on the source data?**
> >   - We thank the reviewer for this great question. By applying suitable data augmentations, DG methods like [6] can indeed do surprisingly well in the target domain *without updating the source model*. This naturally raises the question posed by the reviewer.
> >   - However, once we view the augmentation process as specifying the shifts that a model should be robust to a priori, then it becomes clear why we need methods to adapt on the target data. In particular: (i) we generally do not know what shift will occur upon deployment—there will always be unseen shifts; and (ii) the condition that our augmented development process be sufficiently diverse is untestable—with the worst-case error still being arbitrarily high [9]. Adapting in the target domain is one reasonable solution to these problems.
> >   - The above point is discussed in more detail in Appendix E. To clarify this in the main paper, we will move material from Appendix E to the related work section.
> >
> > We hope that our above responses have addressed your concerns/confusions. If so, we hope that you consider raising your score. If not, please do let us know so that we can try to address any remaining concerns/confusions.
> >
> > Best wishes,
> >
> > Authors
> >
> > [1] https://github.com/brendenlake/omniglot
> >
> > [2] Li, Rui, et al. "Model adaptation: Unsupervised domain adaptation without source data." *Proceedings of the IEEE/CVF Conference on Computer Vision and Pattern Recognition*. 2020.
> >
> > [3] Morerio, Pietro, et al. "Generative pseudo-label refinement for unsupervised domain adaptation." *Proceedings of the IEEE/CVF Winter Conference on Applications of Computer Vision*. 2020.
> >
> > [4] Kurmi, Vinod K., Venkatesh K. Subramanian, and Vinay P. Namboodiri. "Domain Impression: A Source Data Free Domain Adaptation Method." *Proceedings of the IEEE/CVF Winter Conference on Applications of Computer Vision*. 2021.
> >
> > [5] Hendrycks, Dan, et al. "AugMix: A Simple Data Processing Method to Improve Robustness and Uncertainty." International Conference on Learning Representations. 2019.
> >
> > [6] Wang, Haotao, et al. "AugMax: Adversarial Composition of Random Augmentations for Robust Training." *arXiv-2110*. 2021.
> >
> > [7] Hendrycks, Dan, and Dietterich, Thomas. “Benchmarking Neural Network Robustness to Common Corruptions and Perturbations”. *International Conference on Learning Representations*. 2018.
> >
> > [8] Bandi, Peter, et al. "From detection of individual metastases to classification of lymph node status at the patient level: the Camelyon17 challenge." *IEEE Transactions on Medical Imaging*. 2018.
> >
> > [9] Ben-David, Shai, et al. "A theory of learning from different domains." *Machine learning*. 2010.

---

> > > ### Comment · Reviewer_uMHx · 2021-11-24
> > > **Response to Authors**
> > >
> > > The response addresses all the concerns raised in my review. The lack of comparative analysis with [2], [3], [4] is justified in the response. Furthermore, the differences between *measurement shift* and *common corruption* is also clear and adding similar discussions in the paper would be beneficial for reader's understanding.
> > > I agree that proposed method does not specifically assume classification unlike existing works and It does have potential to work on other tasks. However, the lack of analysis on it does not match the claim made in the introduction. I would suggest to put it as a potential future work rather than contribution of the paper.,
> > > In conclusion, I do feel the paper has many interesting analysis and contribution for the field and would like to raise my score and recommend to "Accept" this work.

---

### Official Review · Reviewer_Fzmz · 2021-11-03

**Correctness:** 2
**Technical Novelty And Significance:** 3
**Empirical Novelty And Significance:** 3
**Recommendation:** 6
**Confidence:** 4

**Main Review:**

Strengths:
- This paper addresses the interesting novel problem of measurement shifts, a subset of general domain shifts that occur due to changes in the measurement systems.
- To approximate the source feature distributions, this work uses a softly-binned histogram and provides a novel differentiable implementation that is used in a loss function to adapt the model to the target domain.
- The improvements of BUFR are demonstrated against prior SFDA works on the proposed EMNIST-DA dataset, CIFAR-10-C/100-C and Camelyon17 datasets. Extensive ablation studies are performed on EMNIST-DA to study the proposed components.

Weaknesses:
- This work attempts to illustrate the importance of measurement shifts (a subset of domain shifts) and to propose a method specific to tackling measurement shifts. Most of the empirical justifications rely on evaluations on the proposed EMNIST-DA dataset. However, there are several issues with the arguments, which are detailed as follows.
	- Considering Table 1, the authors do not mention which type of shifts are actually included in the severe category, except that the severe shifts cause a large drop in performance for the AdaBN prior art. Some of the shifts indicated in Fig. 7 like zig-zag, stripe and dotted line do not seem like natural measurement shifts as they occur only locally and may change the class label as well (like in the zig-zag example of Fig. 7, N looks like P). Conversely, the Camelyon17 dataset, with real-world measurement shifts, shows that measurement shifts tend to have more global effects on the images (in Fig. 9). The authors should clearly mention the criteria used for selecting shifts to represent measurement shifts. This is also important because the EMNIST-DA dataset is one of the contributions of this work. Hence, it is not clear whether these severe shifts are a good representation of measurement shifts and thus, whether evaluation on severe shifts is a good indicator of a method successfully tackling measurement shifts.
	- For the Camelyon17 dataset, representing real-world measurement shifts, the performance of all prior arts is competitive w.r.t. the proposed FR and BUFR methods, even when a very small number of target images are available for training. For example, when only 5 target images per class are available, all prior arts (AdaBN, PL, SHOT-IM) get ~82.6% accuracy while FR and BUFR get ~84.6% accuracy. This implies that the advantage of having a method specifically catered to measurement shifts is small. As the number of available target samples increases, the performance gap reduces and prior arts perform the same as the proposed methods. Further, in most practical scenarios, a large number of unlabeled target samples are usually available or easily obtainable.
	- For the CIFAR10-C/100-C benchmarks as well, the expected calibration errors (ECE) of prior arts (AdaBN and TENT) are on par with FR and BUFR (Table 13). Significant improvements to ECE are shown only for the synthetic EMNIST-DA dataset while the ECE metrics for Camelyon17 dataset are not reported. Thus, we cannot conclude whether the improvements w.r.t. usual DA methods are significant.
	- The previous three points illustrate that the significant improvements of this approach are on the synthetic EMNIST-DA dataset and not on the real-world measurement shift based Camelyon17 dataset or the standard CIFAR10-C/100-C benchmarks. Thus, it is not clear whether this method will give significant improvements over simpler and more generic DA techniques like PL (pseudo-labeling) or AdaBN on realistic measurement shift datasets.
	- Further, the ablations are performed only for the EMNIST-DA dataset. Specifically, the ablation studying the bottom-up hypothesis i.e. effect of number of blocks unfrozen for training (in Fig. 4b) is shown for the EMNIST-DA dataset only. In Table 6, it is observed that the FR performance is sometimes better than or very close to BUFR. While the bottom-up hypothesis is intuitive, these observations from Table 6 cast a doubt on it. Thus, ablations on the Camelyon-17 dataset will help clear these doubts and make the argument stronger as they represent realistic measurement shifts.

Minor issues:
- Sec. 2 (Feature restoration): $f_s = h(g_s(X_s))$. Here, $f_s$ is defined as the output of the network rather than the network itself. A possible correction could be $f_s = h \circ g_s$.
- A table of contents should be added for the Appendices, with hyperlinks to the various tables, figures and subsections. This will improve the readability of the paper, given its length.


**Summary Of The Paper:**

This paper addresses the problem of source-free domain adaptation (SFDA) under measurement shifts. The measurement shifts are a subset of general domain shifts which arise from a change in measurement system. The proposed method aims to resolve this problem by restoring the target features to the source feature distribution. Towards this, the source distribution is approximated using a lightweight and flexible approximation, namely softly-binned histograms. For the target adaptation, the feature extractor is trained to re-align the target feature distribution (of a batch) with the saved source distribution. This method is termed as Feature Restoration (FR) and is performed for the feature activations and the pre-softmax activations. An extension of this method, called Bottom-Up FR (BUFR) is introduced which performs the feature restoration in a block-wise manner starting from the early layers of the network. The improvements of BUFR and FR are demonstrated on a new proposed EMNIST-DA dataset for simulating measurement shifts as well as on standard CIFAR10-C, CIFAR100-C (common corruptions) benchmarks and the Camelyon17 dataset that contains realistic measurement shifts. They perform several ablation studies on the EMNIST-DA dataset to highlight the effectiveness of the proposed components.

**Summary Of The Review:**

This work focuses on specifically tackling measurement shifts (subset of general domain shifts). Most of the justifications to highlight the importance of measurement shifts are based on evaluations on the proposed synthetic EMNIST-DA dataset. While the improvements are significant on EMNIST-DA, the simpler and more generic DA techniques obtain competitive performance on the real-world measurement shift dataset (Camelyon17) which casts doubt on the requirement of specific methods to tackle measurement shifts. See the main review for detailed comments. While the paper addresses an interesting problem with good results, I am not convinced by the paper’s argument that such a specific method is better suited for realistic measurement shift DA problems than simpler, generic methods like pseudo-labelling. Thus, I give the rating of “5: marginally below the acceptance threshold”.

**Pose-rebuttal comments**

I thank the authors for their efforts in the rebuttal period. Most of our concerns have been addressed by the authors. Further, having read the other reviews and considering that this work is the first that addresses measurement shift DA, I increase my score to 6: marginally above the acceptance threshold. I hope that the authors will include all the discussions from the rebuttal in their revised draft.

---

> ### Author Response · Authors · 2021-11-18
> **Response to Reviewer Fzmz (1)**
>
> Dear Reviewer Fzmz,
>
> Thank you for your very detailed review and feedback. With regard to your minor issues: we thank you for these corrections and helpful suggestions, and will include them in the updated version of the paper. With regard to your points under weaknesses:
> - **Severe shift in Table 1**
>   - The severe shift was the EMNIST-DA shift that resulted in the lowest AdaBN accuracy. As shown in Table 17, this was the sky shift.
>   - Similarly, the mild shift was the EMNIST-DA shift that resulted in the highest AdaBN accuracy. This was the shot-noise shift.
>   - We agree that this could be made clearer in the text of Section 5.2, and so will clarify this in the next update of the paper.
> - **Natural measurement shifts tend to have more global effects on images**
>   - We believe that naturally-occuring measurement shifts can indeed cause local changes in images. As a simple example, imagine that the sensor developed a small crack, or had some dirt or hair stuck to it---this would be quite similar to the zigzag shift. As a more realistic example, signal intensities may vary across different tissue types for certain medical scans. As the area(s) of interest form only part of the image, such effects can be local [1], [2].
>   - Thus, we believe that EMNIST-DA is indeed a good representation of measurement shifts having 12 “global” shifts and 2 “local” ones---dotted line and zigzag.
> - **Criteria used for selecting shifts to represent measurement shifts**
>   - In Section 2, we define measurement shifts to be domain shifts for which restoring the source features in the target domain is sufficient to do well—we do not need to learn *new* features.
>   - We use this definition to both select real measurement shifts and create synthetic ones:
>     - *EMNIST-DA:* We created a range of shifts, mild to severe, that satisfy our definition and highlight the problems that arise when using entropy minimization techniques to resolve measurement shifts.
>     - *CIFAR-10/100-C:* These corrupted datasets were chosen as they satisfy our definition, i.e. they are particular types of measurement shift.
>     - *Camelyon17:* This medical dataset was again chosen as it satisfies our definition---the shifts arise from changes in measurement systems, namely the staining procedures and image acquisition (e.g. different microscopes). This can be confirmed by inspecting the images from different hospitals---it is reasonable to expect that performance can be restored by simply restoring the source features in the target domain.
>     - *VisDA-C and Office31:* these common UDA datasets were *not* chosen as they do not satisfy our definition---by looking at images in the source and target domains, it is clear that *new* features are needed in the target domain to do well (as discussed in detail in the final paragraph of Section 2).
> - **Results on Camelyon17 are not sufficient to justify a specific method for measurement shift**
>   - *“The advantage of having a method specifically catered to measurement shifts is small”*
>     - With up to 50 target examples per class:
>       - *Accuracy:* our methods reduce the error rate by ~20% compared to the next best method---this improvement does not seem small.
>       - *ECE:* We show the ECE results in the table below. They follow the pattern of the accuracies with our methods outperforming the baselines (with 50 examples per class). We reduce the ECE by ~20% compared to the next best method.
>       - Only our methods meaningfully improve upon the simple AdaBN baseline which uses the target-data BN-statistics. That is, neither of the more generic entropy-minimization methods (PL, SHOT-IM) methods actually work here.
>     - With up to 500 target examples per class:
>       - *Accuracy:* our methods reduce the error rate by ~20% compared to the next best method---this improvement does not seem small.
>       - *ECE:* Our methods continue to outperform baseline methods, reducing the ECE by ~20% compared to the next best method.
>
> **Camelyon17 ECEs:**
>
> |    Method   	|        5       	|       10       	|       50       	|       500      	|   All (15k+)   	|
> |:-----------	|:--------------:	|:--------------:	|:--------------:	|:--------------:	|:--------------:	|
> | Source-only 	|        -       	|        -       	|        -       	|        -       	| $40.8\pm2.1$ 	|
> | AdaBN       	| $14.7\pm2.1$      	| $14\pm2.1$      	| $13.7\pm0.8$       	| $13.5\pm0.7$       	| $13.5\pm0.5$ 	|
> | PL          	| $14.2\pm1.1$ 	| $12.8\pm0.6$ 	|  $13.8\pm1$  	|  $13\pm0.8$  	|  $\mathbf{8.8\pm0.9}$ 	|
> | IM          	| $13.8\pm1.8$ 	| $12.8\pm1.5$ 	| $13.8\pm1.1$ 	| $11.9\pm0.7$ 	|  $9.7\pm0.2$ 	|
> | FR          	| $12.9\pm0.5$ 	| $11.8\pm0.5$ 	| $12.1\pm1.1$ 	|  $9.7\pm0.6$ 	|  $9.8\pm0.5$ 	|
> | BUFR        	| $\mathbf{12.8\pm0.8}$ 	| $\mathbf{11.6\pm0.1}$ 	| $\mathbf{11.1\pm1.1}$ 	|  $\mathbf{9.6\pm0.8}$ 	|  $9.6\pm0.6$ 	|

---

> > ### Author Response · Authors · 2021-11-18
> > **Response to Reviewer Fzmz (2)**
> >
> > - **Results on Camelyon17 are not sufficient to justify a specific method for measurement shift (ctd)**
> >   - *“As the number of available target samples increases, the performance gap reduces and prior arts perform the same as the proposed methods”*
> >     - As highlighted in the above point, with 500 examples-per-class, our methods still significantly improve upon prior methods.
> >     - In our experiments, it is only with over 15,000 examples-per-class that prior methods achieve the same performance.
> >   - *“In most practical scenarios, a large number of unlabeled target samples are usually available or easily obtainable”*
> >     - We disagree with the above statement as changes in measurement systems often arise from the deployment or use of a *new* measurement system. For example, a hospital gets a new scanner, a new slide-staining technique is introduced, the latest iPhone employs a new camera, the fleet of self-driving cars get a sensor upgrade (e.g. cameras or LiDAR).
> >     - In such scenarios, we do not wish to wait until we have over 30,000 examples before we can make a prediction (or even longer for multi-class classification).
> >   - Finally, as mentioned in Section 5.4, it is worth noting that:
> >     - Camelyon17 is an ideal candidate for entropy-minimization techniques due to: (i) only having 2 classes---random pseudo-labels have a 50% chance of being correct; and (ii) AdaBN achieving a high accuracy---most pseudo-labels are correct since updating only the BN-statistics gives around 84%.
> >     - Thus, we expect that our margin of improvement would significantly improve on real-world measurement shifts that: (i) have more than 2 classes; and (ii) are more severe, i.e. cannot be so well resolved by simply updating the BN-statistics (no parameter updates). Examples include:
> >       - image segmentation across [different MRI scanners](https://arxiv.org/pdf/2001.09313.pdf) [2, Figure 1];
> >       - segmentation, attribute detection and disease classification for [skin lesions across different imaging systems](https://arxiv.org/pdf/1902.03368.pdf) [3, Figure 1]; and
> >       - nuclei detection across [different staining techniques](https://arxiv.org/pdf/1907.04681.pdf) [4, Figure 1].
> >     - Unfortunately, these medical datasets have not been made public, limiting the width of our evaluation but also further motivating the need for source-free methods.
> > - **Results on CIFAR-10-C/100-C**
> >   - *“ECE of prior arts (AdaBN and TENT) are on par with FR and BUFR (Table 13)”*
> >     - We kindly ask the reviewer to recheck the ECEs in Table 13, and to read them in tandem with the accuracies in Table 2. Considering the goal is to be both accurate *and* well-calibrated, BUFR is far superior to prior methods on both CIFAR-10-C and CIFAR-100-C. To illustrate this point, we post the ECE and accuracy results side-by-side below, for the prior methods mentioned by the reviewer.
> >     - On CIFAR-10-C, BUFR achieves both the highest accuracy and lowest ECE, reducing the error margin of TENT by 20.9%.
> >     - On CIFAR-100-C, BUFR achieves the highest accuracy and the ECE is close to that of AdaBN. While AdaBN achieves a lower ECE, note that its accuracy is 12% lower. Also note that the ECE of TENT is almost double that of BUFR.
> >     - This illustrates that significant ECE improvements are *not* just shown for EMNIST-DA, but also for CIFAR-10-C, CIFAR-100-C and Camelyon17 (in the low-data regime).
> >
> > **CIFAR-10-C/100-C accuracy and ECE results:**
> >
> > | Method 	| CIFAR-10-C         	|                      	| CIFAR-100-C        	|                      	|
> > |--------	|--------------------	|----------------------	|--------------------	|----------------------	|
> > |        	| Acc ($\uparrow$) 	| ECE ($\downarrow$) 	| Acc ($\uparrow$) 	| ECE ($\downarrow$) 	|
> > | AdaBN  	| $80.4 \pm 0.1 $       	| $11.2 \pm 0.1$          	| $56.6 \pm 0.3$        	| $\mathbf{12.5 \pm 0.1}$          	|
> > | TENT   	| $86.6 \pm 0.3$        	| $12.8 \pm 0.3$          	| $66.0 \pm 0.4$        	| $25.7 \pm 0.4$          	|
> > | BUFR   	| $\mathbf{89.4 \pm 0.2}$        	| $\mathbf{10.0 \pm 0.2}$          	| $\mathbf{68.5 \pm 0.2}$        	| $14.5 \pm 0.3$          	|

---

> > > ### Author Response · Authors · 2021-11-18
> > > **Response to Reviewer Fzmz (3)**
> > >
> > > - **Significant improvements are on EMNIST-DA and not Camelyon17 or CIFAR10-C/100-C, making it unclear if the proposed method will give significant improvements over simpler and more generic DA techniques like PL (pseudo-labeling) or AdaBN on realistic measurement shifts**
> > >   - As argued in the above responses:
> > >     - We do in fact show significant improvements on the standard CIFAR-10-C/100-C benchmarks;
> > >     - We do show significant improvements on Camelyon17 with 500 or less examples-per-class (something we believe to be a realistic constraint for new measurement systems);
> > >     - We would expect these improvements to be even more significant on real-world measurement shifts that are severe and have more than 2 classes.
> > >   - Thus, for both synthetic and real measurement shifts, we believe that the advantages of our method over more generic DA ones (like pseudo-labelling) are clear.
> > >   - In addition, some of these “more generic” DA techniques like PL and SHOT-IM can only be applied to classification problems (discrete outputs), while our method can be applied to regression problems (continuous outputs).
> > >
> > > We hope that our responses have convinced you that measurement shifts are indeed better handled by a specific method (like ours), rather than a more generic entropy-minimization method (like pseudo-labelling). If they did, we hope that you consider raising your score. If they did not, please let us know why so that we can try to address any further concerns.
> > >
> > > Best wishes,
> > >
> > > Authors
> > >
> > > [1] Jager, Florian, and Joachim Hornegger. "Nonrigid registration of joint histograms for intensity standardization in magnetic resonance imaging." *IEEE Transactions on Medical Imaging* (2008).
> > >
> > > [2] Li, Hongwei, et al. "Domain Adaptive Medical Image Segmentation via Adversarial Learning of Disease-Specific Spatial Patterns." *arXiv:2001.09313* (2020).
> > >
> > > [3] Codella, Noel, et al. "Skin lesion analysis toward melanoma detection 2018: A challenge hosted by the international skin imaging collaboration (isic)." *arXiv:1902.03368* (2019).
> > >
> > > [4] Brieu, Nicolas, et al. "Domain adaptation-based augmentation for weakly supervised nuclei detection." *arXiv:1907.04681* (2019).

---

### Official Review · Reviewer_V9Lg · 2021-11-03

**Correctness:** 4
**Technical Novelty And Significance:** 3
**Empirical Novelty And Significance:** 4
**Recommendation:** 8
**Confidence:** 4

**Main Review:**

Strengths
+ The detailed framework is well explained to be easily understood, and the figures were very helpful for understanding.
+ The state-of-the-art performance was validated well by using various types of datasets and the compared algorithms were recent and reasonable.
+ The empirical analysis was helpful to understand the overall workflow and solve the questions about the framework.

Weaknesses
- The analysis for the intuition and assumption of the proposed algorithm is not presented. Even though the empirical analysis for the algorithm is well described, the presence of measurement shift and its importance cannot be understood just through this paper. I agree that the measurement shift can be occurred by the measurement system, but it is hard to know how the shift looks like and which part of the model is affected by the measurement shift. With the description of the weak intuition, the design of the framework was hard to be followed. Furthermore, the bottom-up feature restoration assumes that the low-level features are affected by the measurement shift, but there is no related analysis showing the effect of measurement shift for the respective layers at all. Thus, I want to see the analysis of the measurement shift to explain the reason for the proposed architecture.
- Some sentences are repeated in the abstract and the introduction. With the limited pages of paper, repeated sentences should be avoided to explain the proposed framework as much as possible with the various aspects. I recommend the rewriting of the abstract to improve the readability of the paper. The current abstract contains too much detailed information, which can let the readers confused before understanding the intuition of this paper.

**Summary Of The Paper:**

This paper focuses on the improvement of the source-free domain adaptation problem. While the previous studies destroy the model calibration to improve the domain generalization or update the source model to fit the feature distribution from the target domain, the proposed algorithm reduces the domain gap by restoring the distribution of source features at the adaptation phase. This feature restoration step is based on the assumption of the measurement shift that comes from the difference between the data-grabbing system and environments. Based on the assumption, the authors proposed the method of feature restoration (FR) and bottom-up feature restoration (BUFR) that is its extended version. The state-of-the-art performance of the proposed algorithm was validated by numerous datasets including the simple digit datasets and the real-world complicate datasets.

**Summary Of The Review:**

Even though the intuition of the algorithm is not given yet, the designed framework showed the state-of-the-art performance, which means that the measurement shift would be one of the hidden keys to improve the domain generalization. When the authors present adequately the additional analysis for its intuition and revise some repeated parts, I am preparing to increase my score.

---

> ### Author Response · Authors · 2021-11-18
> **Response to Reviewer V9Lg**
>
> Dear Reviewer V9Lg,
>
> Thank you for your thorough review and helpful feedback. With regard to your points under weaknesses:
> - **Analysis and intuition**
>   - *Building intuition*
>     - In Section 2, we discuss how measurement shifts (shown in Figure 1c) can be resolved by simply *restoring the same features* rather than learning new ones, while non-measurement shifts (shown in Figure 1d) cannot as they require learning *new* features. In particular, we use these examples in Figures 1c and 1d to motivate the use of FR (and BUFR) for measurement shifts.
>     - In Section 3.3, we build intuition for bottom-up training:
>       - *“A simple gradient-based adaptation of g_t would adapt the weights of all layers at the same time. Intuitively, however, we expect that many measurement shifts like brightness or blurring can be resolved by only updating the weights of early layers. If the early layers can learn to extract the same features from the target data as they did from the source (e.g. the same edges from brighter or blurrier images of digits), then the subsequent layers shouldn’t need to update. Building on this intuition, we argue that adapting all layers simultaneously unnecessarily destroys learnt structure in the later layers of a network, and propose a bottom-up training strategy to alleviate the issue.”*
>     - Through Figure 4a, we build strong intuition as to why our method works---it successfully extracts features with a similar semantic meaning in the target domain. To aid intuition earlier in the paper, we will add a forward pointer to this figure in Section 1 or 3.
>   - *Illustrating which parts of a model are most affected by a measurement shift / the effect of measurement shift on different layers*
>     - In Figure 4b, we show that many measurement shifts can indeed be resolved by simply updating the first one or two blocks of the network---confirming the intuition provided in Section 3.3. This illustrates that the early blocks or layers of a network are indeed most affected by measurement shifts, and explains the efficacy of bottom-up training for resolving measurement shifts.
>     - In Figure 17, we further analyse the effect of measurement shift on different layers. Again we find that measurement shifts mostly affect the early blocks or layers as they can be well resolved by only updating the early blocks or layers. In particular, Figure 17 shows that BUFR outperforms FR by primarily updating just the first two blocks.
>     - We now provide some further intuition through an additional analysis of which layers are most affected by a measurement shift, through Figures S1 and S2 which we have added to the supplementary material. For 10 sample units in each layer, Figure S1 shows the (symmetric) KL divergence between the unit-level activation distributions under the source (EMNIST) and target (EMNIST-DA crystals) data. This shows that, before adapting, the unit-activation distributions in all layers of the network have changed significantly, as indicated by the large KL divergences. Figure S2 shows the KL divergences after updating *just the first layer*. As shown, updating only the first layer restores “normality” in all subsequent layers, with the unit-level activation distributions on the target data realigning with those saved on the source (very low KL divergences). This indicates that measurement shifts primarily affect the first layer (or block)---since they can be mostly resolved by updating the first layer (or block)---and also further motivates bottom-up training for measurement shifts. In the next update of the paper, we will include this analysis in the Appendix.
>   - *Further examples on “what do measurement shifts look like?”*
>     - To complement the examples shown in Figure 7, we now link some further illustrations of real-world measurement shift:
>       - [Different MRI scanners](https://arxiv.org/pdf/2001.09313.pdf) (Figure 1);
>       - [Different imaging techniques for skin lesion analysis](https://arxiv.org/pdf/1902.03368.pdf) (Figure 1);
>       - [Different staining techniques for the detection of nuclei](https://arxiv.org/pdf/1907.04681.pdf) (Figure 1); and
>       - [Different staining techniques in histopathology more generally](https://arxiv.org/pdf/2012.12413.pdf) (Figure 2).
>     - Unfortunately, many such biomedical datasets are not publicly available for us to analyse our methods on.
> - **Shortening the abstract**
>   - Upon reflection, we tend to agree with the reviewer that the abstract could be shortened to avoid one or two points being repeated in the introduction. As a result, we will shorten the abstract in next update of the paper.
>
> We hope that our above responses are adequate to: (i) provide further analysis for the intuition behind our method; and (ii) avoid repeated sentences in the abstract and introduction. If not, please let us know why so that we can try to address any remaining concerns.
>
> Best wishes,
>
> Authors

---

> > ### Author Response · Authors · 2021-11-18
> > **Response to Reviewer V9Lg (ctd)**
> >
> > In particular, we will use the following shortened abstract in the next updated version of the paper:
> > - Source-free domain adaptation (SFDA) aims to adapt a model trained on labelled data in a source domain to unlabelled data in a target domain *without access to the source-domain data during adaptation*. Existing methods for SFDA leverage entropy-minimization techniques which: (i) apply only to classification; (ii) destroy model calibration; and (iii) rely on the source model achieving a good level of feature-space class-separation in the target domain. We address these issues for a particularly pervasive type of domain shift called *measurement shift* which can be resolved by *restoring* the source features rather than extracting new ones. In particular, we propose *Feature Restoration* (FR) wherein we: (i) store a lightweight and flexible approximation of the feature distribution under the source data; and (ii) adapt the feature-extractor such that the approximate feature distribution under the target data realigns with that saved on the source. We additionally propose a bottom-up training scheme which boosts performance, which we call *Bottom-Up Feature Restoration* (BUFR). On real and synthetic data, we demonstrate that BUFR outperforms existing SFDA methods in terms of accuracy, calibration, and data efficiency, while being less reliant on the performance of the source model in the target domain.

---

> > > ### Comment · Reviewer_V9Lg · 2021-11-20
> > > **I appreciate for your kind response**
> > >
> > > I appreciate your kind response. The revision to show and validate the importance of measurement shifts would be helpful for the readers to understand the study. If possible, I want the analysis for the measurement shift by using domains other than the medical images. However, the importance of the measurement shifts was well validated by the experimental results and the authors said the repeated sentences would be revised, so I increase my score to "Accept".
> > >
> > > Thank you.

---

### Official Review · Reviewer_HPoa · 2021-11-06

**Correctness:** 4
**Technical Novelty And Significance:** 3
**Empirical Novelty And Significance:** 3
**Recommendation:** 8
**Confidence:** 5

**Main Review:**

Strengths
1. The paper is very well-written and easy to follow. It targets the SFDA setting but with a different type of domain shift to the common setting. The measurement shift is indeed an important piece of domain shift on many applications.
2. The proposed method is simple and effective. At the first glance, it might be doubtful that they use source features for training. The authors give a memory storage comparison for the estimated source distributions, proving it is very lightweight.
3. Considering the simplicity of this method, it achieves promising results on several types of datasets in terms of accuracy and calibration compared to the SOTA. The ablation and analysis are sufficient to support the claim of this paper.

Weakness:
1. The main comparison of this paper is SHOT, which is an entropy-based method and thus has calibration problems and strict assumptions. I think it is also preferable to see how this method compares to other feature restoration type SFDA methods such as [1] and [2].
2. 2nd point is not weakness. But it will be also helpful to include performance comparison on Office-31 or home as a baseline.

[1]Li, Rui, et al. "Model adaptation: Unsupervised domain adaptation without source data." Proceedings of the IEEE/CVF Conference on Computer Vision and Pattern Recognition. 2020.
[2] Liu, Yuang, Wei Zhang, and Jun Wang. "Source-free domain adaptation for semantic segmentation." Proceedings of the IEEE/CVF Conference on Computer Vision and Pattern Recognition. 2021.

**Summary Of The Paper:**

This paper tackles the specific domain shift, named measurement shift on source-free adaptation setting. They analyze the drawbacks of existing entropy-based SFDA methods and instead propose a method to retore the source feature with a lightweight approximation. The paper emphasizes the difference of measurement shift with common concept shift, and their method can achieve significant performance on measurement shift setting compared to SOTA.

**Summary Of The Review:**

This paper deals with the measurement shift in the source-free DA problem. The authors propose a simple but effective method to approximate source distribution in a lightweight manner. I recommend to accept this paper.

---

> ### Author Response · Authors · 2021-11-18
> **Response to Reviewer HPoa**
>
> Dear Reviewer HPoa,
>
> Thank you for your valuable review and helpful suggestions. With regard to your points under weaknesses:
> - **Comparing to more feature-restoration based SFDA methods like [1] and [2]**
>   - We agree that it is important to compare our method against restoration-based SDFA methods, not only entropy-based ones. For this reason, we compare against:
>     - The batch-norm matching method of [3] (called “Marginal Gaussian” in the tables); and
>     - A “Full Gaussian” method which actually stores more information about the source data-distribution than our method.
>   - While we would have liked to also compare against more restoration-based SDFA methods like [1] and [2], it was not possible as these works:
>     - Have not released their code; and
>     - Involve complicated GAN-based frameworks which are notoriously difficult to implement and tune.
>   - Thus, while we agree that it would be preferable to include these additional methods, we feel that:
>     - Implementing them would take an unreasonable amount of time. We hope that the authors of these works release code in the future to facilitate comparisons.
>     - Our current restoration-based SFDA baselines, mentioned above, are sufficient to illustrate the efficacy of our particular restoration method (softly-binned histograms and bottom-up training).
> - **Helpful to also performance comparison on Office-31 or Office-Home**
>   - As discussed in Section 2, common UDA benchmarks like Office-31, Office-Home and VisDA-C do not contain measurement shift. Thus, they are not suitable for evaluating our methods for addressing measurement shift.
>   - Nonetheless, in Appendix D, we report and analyse results on VisDA-C to evaluate our methods on domain shifts that require learning *new* features in the target domain (i.e. *non* measurement shifts).
>   - For the final version of the paper, we will add Office-31 results to our analysis in Appendix D.
>
> We hope that our above responses have addressed your concerns. If not, please do let us know so that we can try to address any remaining concerns.
>
> Best wishes,
>
> Authors
>
> [1] Li, Rui, et al. "Model adaptation: Unsupervised domain adaptation without source data." *Proceedings of the IEEE/CVF Conference on Computer Vision and Pattern Recognition* (2020).
>
> [2] Liu, Yuang, Wei Zhang, and Jun Wang. "Source-free domain adaptation for semantic segmentation." *Proceedings of the IEEE/CVF Conference on Computer Vision and Pattern Recognition* (2021).
>
> [3] Ishii, Masato, and Masashi Sugiyama. "Source-free Domain Adaptation via Distributional Alignment by Matching Batch Normalization Statistics." *arXiv:2101.10842* (2021).

---

### Public Comment · ~Yuejiang_Liu1 · 2022-02-21
**Questions about feature distribution approximation**

Dear Authors,

Thanks for the great paper! The idea of soft-binning for distribution approximation looks very interesting!
I have a couple of questions and would love to seek your opinion.

1. Binning Parameter \
It’s mentioned in Appendix A that the binning parameter is fixed as `8 bins per feature` for all datasets, such as MNIST, CIFAR100, and VisDA. \
It looks impressive that 8 bins are sufficient for CIFAR-100, given that there are likely 100 clusters in high dimensions. Do you have any intuition why it worked well?

2. Space Complexity \
It's emphasized in your Related Work section that the `storage is linear in the number of features` for binning, but `at least quadratic in the number of features` for previous methods, including CMD [1]. \
Please correct me if I’m wrong, but isn’t the space complexity of CMD also linear to the feature dimension ([cf. page 5](https://openreview.net/pdf?id=SkB-_mcel))?

3. Soft Binning vs. Moment Matching \
In our recent paper TTT++ [2], we also explored lightweight approximations of feature distributions, and found online moment matching quite effective for source-free test-time adaptation. \
I'm quite curious about how well the binning-based method performs compared with moment matching. What do you think about the pros/cons of soft binning vs. second-order moments or high-order central moments?

[1] [Central Moment Discrepancy (CMD) for Domain-Invariant Representation Learning, ICLR'17](https://openreview.net/pdf?id=SkB-_mcel) \
[2] [TTT++: When Does Self-Supervised Test-Time Training Fail or Thrive?, NeurIPS'21](https://proceedings.neurips.cc/paper/2021/file/b618c3210e934362ac261db280128c22-Paper.pdf)

---

> ### Public Comment · ~Cian_Eastwood1 · 2022-03-02
> **Response**
>
> Dear Yuejiang,
>
> Thank you for your interest and great questions! Below we response to each point:
> 1. *Binning Parameter:* For CIFAR-100, there are likely 100 clusters in the 256-D feature space, *not* in each of the 1-D marginal feature distributions. As an extreme example, if each feature was only active for 1 class, then 1 bin per feature would suffice (assuming at least 100 features). Thus, we believe that 8 bins is often sufficient as individual features tend to "fire" for only a small subset of the classes. We verified this empirically by plotting the marginal feature distributions for CIFAR-100, and they looked very similar to those on CIFAR-10 (see Figure 14). However, some datasets may of course require more bins. We speculate that the ratio n_classes/feature_dim is one of many factors determining the number of bins required.
>
> 2. *Space complexity:* Yes you are correct, thank you for pointing this out. It will be reflected in the camera-ready version. To clarify, the quadratic space-complexity comes from modelling the joint feature distribution (i.e. the cross-terms) rather than just the marginals. We had thought that CMD was aligning the full joint distribution, but now see that only the moments of the marginals are aligned (making its storage linear in the number of features).
>
> 3. *Soft Binning vs. Moment Matching:* We found soft-binning to work better than moment matching on our datasets. With 2 moments, Table 1 shows that soft binning (FR) worked better than aligning the moments of the marginals (Marg. Gauss.) and aligning the moments of the joint (Full Gauss.). We saw similar results with more than 2 moments, but did not conduct a rigorous study. However, it would be very interesting to see such a rigorous study or comparison. We actually considered a follow-up study in this direction, comparing the performance and storage complexity of different alignment methods. Happy to discuss offline if of interest.

---

> > ### Public Comment · ~Yuejiang_Liu1 · 2022-04-27
> > **Thanks for your answer**
> >
> > Thank you Cian for the detailed answer!

---

### Decision · Program_Chairs · 2022-01-20

**Decision:**

Accept (Spotlight)

**Comment:**

The paper aims to solve the source-free domain adaptation, specifically on measurement shift. The proposed method lowers the domain gap via restoring the source feature distribution with a lightweight approximation. The effectiveness and performance are well validated by extensive experiments on various datasets compared with other recent methods, and the ablation analysis supports the claim of the paper well. The paper is well written with clear logic to follow.